# CAUSAL REPRESENTATION LEARNING FROM MULTI-MODAL BIOMEDICAL OBSERVATIONS

**Yuewen Sun**[1,2][*]**, Lingjing Kong**[2][*]**, Guangyi Chen**[1,2]**, Loka Li**[1]**, Gongxu Luo**[1]**, Zijian Li**[1]**,
Yixuan Zhang**[1]**, Yujia Zheng**[2]**, Mengyue Yang**[3]**, Petar Stojanov**[4]**, Eran Segal**[1]**,
Eric P. Xing**[1,2]**, Kun Zhang**[1,2]
[1]Mohamed bin Zayed University of Artificial Intelligence, [2]Carnegie Mellon University,
[3]University of Bristol, [4]Broad Institute of MIT and Harvard

## ABSTRACT

Prevalent in biomedical applications (e.g., human phenotype research), multimodal datasets can provide valuable insights into the underlying physiological mechanisms. However, current machine learning (ML) models designed to analyze these datasets often lack interpretability and identifiability guarantees, which are essential for biomedical research. Recent advances in causal representation learning have shown promise in identifying interpretable latent causal variables with formal theoretical guarantees. Unfortunately, most current work on multimodal distributions either relies on restrictive parametric assumptions or yields only coarse identification results, limiting their applicability to biomedical research that favors a detailed understanding of the mechanisms.

In this work, we aim to develop flexible identification conditions for multimodal data and principled methods to facilitate the understanding of biomedical datasets. Theoretically, we consider a nonparametric latent distribution (c.f., parametric assumptions in previous work) that allows for causal relationships across potentially different modalities. We establish identifiability guarantees for each latent component, extending the subspace identification results from previous work. Our key theoretical contribution is the structural sparsity of causal connections between modalities, which, as we will discuss, is natural for a large collection of biomedical systems. Empirically, we present a practical framework to instantiate our theoretical insights. We demonstrate the effectiveness of our approach through extensive experiments on both numerical and synthetic datasets. Results on a real-world human phenotype dataset are consistent with established biomedical research, validating our theoretical and methodological framework.

## 1 INTRODUCTION

Multimodal datasets provide rich and comprehensive insights into complex biomedical systems, offering the potential to provide a deeper understanding of physiological mechanisms. For example, the human phenotype dataset (Levine et al., 2024) contains measurements from multiple modalities, including anthropometric data, sleep monitoring, and genetic information. Proper analysis of such data can potentially uncover the underlying mechanisms that drive phenotypic diversity and disease susceptibility, leading to the discovery of novel molecular markers and the development of predictive models for disease. Recent advances in large-scale models have made it possible to exploit large biomedical datasets for various tasks such as protein structure prediction (Jumper et al., 2021; Lin et al., 2023), gene-disease association identification (Diaz Gonzalez et al., 2023; Zagirova et al., 2023), and novel drug candidate discovery (Pal et al., 2023; Zheng et al., 2024b).

Despite the impressive performance of these models, their trustworthiness remains a contentious issue (Zheng et al., 2023). A major concern lies in their lack of interpretability, which poses significant challenges in biomedical research and hinders the safe and ethical application of these models. For example, in clinical decision-making (Hager et al., 2024), if the model recommends a specific

---

[*]Equal contribution.

treatment plan for a patient, it is important for clinicians to understand the rationale behind the recommendation. Without such transparency, it is difficult to trust the model's output or integrate these systems into critical decision-making workflows. Although several explainable models have been developed for multimodal datasets (Tang et al., 2023), this area remains largely underexplored.

Fortunately, recent advances in causal representation learning (CRL) (Schölkopf et al., 2021) have shown promise in identifying latent causal structures from raw observations, making it well-suited for biomedical applications. For example, a plethora of CRL studies (Hyvarinen et al., 2019; Khemakhem et al., 2020a; Zhang et al., 2024b; Buchholz et al., 2024; von Kügelgen et al., 2023; Zhang et al., 2024a; Li et al., 2024c; Ahuja et al., 2023) effectively utilize temporal information or domain indices to identify latent causal models and apply them in fMRI data. Recently, a growing body of CRL research has investigated multimodal distributions (Yao et al., 2023; Morioka & Hyvarinen, 2023; 2024; Daunhawer et al., 2023; Sturma et al., 2023; Gresele et al., 2020). These works leverage shared information across modalities to establish identifiability guarantees for latent variables (Yao et al., 2023; Morioka & Hyvarinen, 2024; Daunhawer et al., 2023). Despite these advancements, some aspects of these works are still limited. For instance, Von Kügelgen et al. (2021); Daunhawer et al. (2023); Yao et al. (2023) only focus on identifying latent subspaces that are directly shared by multiple modalities. In practice, however, many informative latent variables may influence multiple modalities indirectly through intermediate latent mechanisms. Moreover, such subspace identifiability loses track of the intricate causal influences among individual components, leading to a limited view of the underlying latent mechanism. Morioka & Hyvarinen (2024); Gresele et al. (2020); Morioka & Hyvarinen (2023) rely on specific assumptions about latent variable distributions (e.g., independence or exponential family). These constraints significantly limit their applicability for biomedical datasets that involve complex interactions among latent factors.

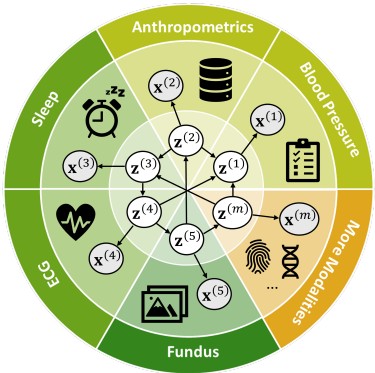

Figure 1: Multimodal data with causal latent variables.

In this work, we aim to develop identification theory with *multimodal biomedical* datasets in mind, and design *principled and interpretable* models to facilitate analyzing such datasets. We assume that observations $\mathbf{x}^{(m)}$ in each modality $m$ are generated by a specific set of latent components $\{z_i^{(m)}\}_i$, and allow for flexible causal relationships among latent components from potentially different modalities, such that $z_i^{(m)} \rightarrow z_j^{(n)}$ for $m \neq n$, $i \neq j$, as shown in Figure 1. **Theoretically**, we provide identifiability guarantees for each latent component $z_i^{(m)}$, thus generalizing the subspace identification results in Yao et al. (2023); Daunhawer et al. (2023) while avoiding independence or parametric restrictions on the latent distribution $p(\{\mathbf{z}^{(m)}\}_m)$ as in Morioka & Hyvarinen (2023; 2024); Gresele et al. (2020). In particular, we first show that any latent subspace $\mathbf{z}^{(m)}$ can be identified as long as $\mathbf{z}^{(m)}$ exerts sufficient influences on other modalities, which is weaker than assuming that $\mathbf{z}^{(m)}$ is directly shared across multiple modalities as in Daunhawer et al. (2023); Yao et al. (2023). Based on this subspace identification, we leverage the sparsity of the causal connections between modalities to further identify each latent component $\{z_i^{(m)}\}_i$. This notion of causal sparsity has been explored in recent work (Lachapelle et al., 2023; Xu et al., 2024; Zheng et al., 2022) in other causal identification settings and has been shown realistic in many biomedical systems (Busiello et al., 2017; Milo et al., 2002; Babu et al., 2004; West et al., 2002; Banavar et al., 1999), as we will discuss further in Section 4.

**Empirically**, we develop a theoretically grounded estimation framework to recover the latent components in each modality. Our model implements our theoretical conditions (in particular, conditional independence and sparsity constraints) on top of normalizing flow (Huang et al., 2018; Kobyzev et al., 2020) within the encoder-decoder framework. Extensive experiments on both numerical and synthetic datasets demonstrate its effectiveness. Most notably, our framework enables the identification of latent causal variables that capture complex biomedical interactions and facilitates the analysis of potential causal mechanisms across modalities, which are important for clinical decision-making. The evaluation results on a real-world human phenotype dataset provide novel insights into the relationships between modalities, and the discovered causal relationships align with the findings from biomedical research, highlighting our contributions to the biomedical domain.

## 2 RELATED WORK

**ML models for biomedical research.** For biomedical applications, ML models are developed to extract informative representations to facilitate downstream tasks, including DNA sequence modeling (Zhou et al., 2024; Nguyen et al., 2023; Dalla-Torre et al., 2023), protein structure prediction (Jumper et al., 2021; Lin et al., 2023), and disease detection (Zhou et al., 2023; Jang et al., 2024). The success of large language models (LLMs) has significantly advanced sequence modeling for DNA, RNA, and proteins (Celaj et al., 2023; Shulgina et al., 2024; Nguyen et al., 2024; Li et al., 2023; Chen et al., 2023; Lin et al., 2023), yet these methods primarily operate on a single modality, limiting their applicability to the multimodal datasets, which are commonly encountered in biomedical research. Although several studies have explored integrating multimodal biomedical data (Garau-Luis et al., 2024; Pei et al., 2024; Taylor et al., 2022), these approaches often lack theoretical guarantees, raising concerns about the reliability of their results. In this paper, we leverage causal principles to develop theoretically sound ML models for multimodal biomedical data, aiming to provide reliable and interpretable insights into complex biomedical systems.

**Multimodal representation learning.** Multimodal representation learning (Zhang et al., 2020; Manzoor et al., 2023) refers to the process of learning representations from multiple data modalities (e.g., text, image, audio) for specific tasks. Recent advances have leveraged contrastive learning techniques to improve the alignment of latent spaces across different modalities (Daunhawer et al., 2023; Wang et al., 2022; Radford et al., 2021; Khosla et al., 2020). Methods like CLIP (Radford et al., 2021) and Contrastive Predictive Coding (Oord et al., 2018) have demonstrated the ability to recover shared latent factors across modalities by (implicitly) maximizing mutual information between representations. However, challenges remain in achieving finding modality-specific representations, which requires novel approaches that preserve the unique characteristics of each modality.

**Identifiable CRL.** CRL aims to identify high-level causal variables from low-level observations, integrating principles from both machine learning and causality (Schölkopf et al., 2021), and can be viewed as an extension of causal discovery (Spirtes et al., 2001; Li et al., 2024a; Luo et al., 2025; Li et al., 2024b; Ziu et al., 2024). CRL methods with identifiability guarantees can be classified based on the assumptions they impose, including sparsity constraints (Xu et al., 2024; Zheng et al., 2022; Zheng & Zhang, 2023; Lachapelle et al., 2024), interventional/multi-distribution settings (Hyvarinen et al., 2019; Khemakhem et al., 2020a; Zhang et al., 2024b; Kong et al., 2023; Buchholz et al., 2024; von Kügelgen et al., 2023; Zhang et al., 2024a; Li et al., 2024c; Varici et al., 2023; Ahuja et al., 2023; Jiang & Aragam, 2023), and of particular relevance to our work, multimodality (Yao et al., 2023; Morioka & Hyvarinen, 2023; 2024; Daunhawer et al., 2023; Sturma et al., 2023; Gresele et al., 2020). To provide a clearer comparison, Table 1 summarizes representative works in the multimodality category and highlights their differences from our work.

**Empirical CRL for multimodal applications.** In contrast to the previously discussed works that emphasize identifiability, another line of multimodal CRL research prioritizes practical applications in various domains, without addressing theoretical identifiability. Mao et al. (2022) assume independent latent variables and introduce a two-module amortized variational algorithm to learn representations from medical images and biomedical data. Zheng et al. (2024a) develop a contrastive learning-based approach to extract modality-specific and modality-invariant representations from time-series tabular and textual data for root cause analysis. Rawls et al. (2021) leverage behavioral and psychiatric phenotyping alongside high-resolution neuroimaging data from the Human Connectome Project (Van Essen et al., 2013), and perform greedy fast causal inference (Ogarrio et al., 2016) to investigate causal relations in alcohol use disorder. In contrast, our work establishes formal identification theory and integrates the theoretical insights into our estimation model.

Table 1: **Related work on multimodal causal representation learning.** This table considers whether a method can accommodate more than two modalities, whether the latent variable distribution is nonparametric, whether it allows dependency among latent variables, and whether identifiability is component-wise.

| Related work | > 2 Modalities | Nonparam. Dist. | Latent Dependency | Component-wise Iden. |
|---|---|---|---|---|
| Gresele et al. (2020) | ✓ | ✗ | ✗ | ✓ |
| Von Kügelgen et al. (2021) | ✗ | ✓ | ✓ | ✗ |
| Daunhawer et al. (2023) | ✗ | ✓ | ✓ | ✗ |
| Yao et al. (2023) | ✓ | ✓ | ✓ | ✗ |
| Morioka & Hyvarinen (2024) | ✓ | ✗ | ✓ | ✓ |
| **Our work** | ✓ | ✓ | ✓ | ✓ |

## 3 LATENT MULTIMODAL CAUSAL MODELS

Real-world biomedical datasets often integrate multiple modalities, each characterizing a unique yet interrelated aspect of the subject. For instance, the human phenotype dataset (Shilo et al., 2021) consists of tabular data, time series, images, and text, capturing diverse biomedical measurements such as anthropometrics, sleep monitoring, and genetic information. Understanding the latent factors behind each modality and their interplay can provide valuable insights into underlying biomedical mechanisms, ultimately facilitating the advancement of biomedical technologies. With this goal in mind, we formalize the multimodal data-generating processes as follows.

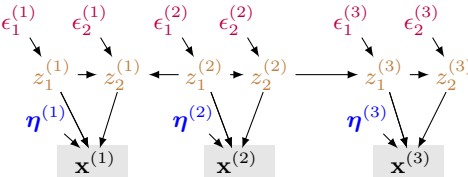

Figure 2: An illustrative example of the hypothesis space underlying the biomedical system.

**Data-generating processes.** Let $\mathbf{x} := [\mathbf{x}^{(1)}, \ldots, \mathbf{x}^{(M)}]$ be a set of observations/measurements from $M$ modalities, where $\mathbf{x}^{(m)} \in \mathbb{R}^{d(\mathbf{x}^{(m)})}$ represents the observation from modality $m$ with dimensionality $d(\mathbf{x}^{(m)})$. Let $\mathbf{z} = [\mathbf{z}^{(1)}, \ldots, \mathbf{z}^{(M)}]$ be the set of causally related latent variables underlying $M$ modalities. Specifically, the data generation process (Figure 2) can be formulated as

$$z_i^{(m)} := g_{z_i^{(m)}}(\text{Pa}(z_i^{(m)}), \epsilon_i^{(m)}), \quad \text{(latent causal relations)} \tag{1}$$

$$\mathbf{x}^{(m)} := g_{\mathbf{x}^{(m)}}(\mathbf{z}^{(m)}, \boldsymbol{\eta}^{(m)}), \qquad \text{(generating functions)} \tag{2}$$

where $\text{Pa}(\cdot)$ denotes the parents of a variable. Since we allow for causal relations to exist within and across modalities, $\text{Pa}(\cdot)$ potentially includes latent variables across multiple modalities. The differentiable function $g_{\mathbf{z}}$ encodes the latent causal graph connecting the latent components, and its Jacobian matrix $\boldsymbol{J}_{g_{\mathbf{z}}}$ can be permuted into a strictly triangular matrix. We denote $\epsilon_i^{(m)}$ as the exogenous variable for $z_i^{(m)}$, where all exogenous variables are mutually independent. $\boldsymbol{\eta}^{(m)}$ represents domain-specific information independent of other components.

**Example.** In healthcare, different modalities capture complementary physiological aspects. A chest X-ray $\mathbf{x}^{(m)}$ may reflect latent factors such as lung density, cardiac silhouette, and ribcage structure, represented by $\mathbf{z}^{(m)}$. These latent factors can causally influence those in other modalities, $\mathbf{z}^{(n)}$, such as pulmonary function parameters (e.g., forced vital capacity) and cardiovascular biomarkers (e.g., left ventricular mass). These, in turn, may affect electrical activity recorded in an ECG, represented by $\mathbf{x}^{(n)}$, by modulating heart rate variability and conduction patterns.

**Goal.** As outlined previously, we aim to learn the latent variables underlying each modality and their causal relations. Formally, consider two specifications of the data-generating process in Eq. (1) and Eq. (2): $\boldsymbol{\theta} := \{g_{\mathbf{x}^{(m)}}, g_{\mathbf{z}^{(m)}}, p(\boldsymbol{\epsilon}^{(m)})\}_{m=1}^M$ and $\hat{\boldsymbol{\theta}} := \{\hat{g}_{\mathbf{x}^{(m)}}, \hat{g}_{\mathbf{z}^{(m)}}, \hat{p}(\boldsymbol{\epsilon}^{(m)})\}_{m=1}^M$, both of which fit the marginal distribution $p(\mathbf{x})$. Our objective is to show that, *given the same value of* $\mathbf{x}$, each estimated latent component $\hat{z}_i^{(m)}$ is equivalent to its true counterpart $z_i^{(m)}$ up to an invertible transformation $h_i^{(m)}$, i.e., $\hat{z}_i^{(m)} = h_i^{(m)}(z_i^{(m)})$. [1] This *component-wise identifiability* ensures that latent components (e.g., gene types, nutrient levels) are disentangled from the observed measurements $\mathbf{x}$ while preserving their original information. Once component-wise identifiability is achieved, one can readily apply standard causal discovery algorithms (e.g., PC (Spirtes et al., 2001)) to the identified components $\hat{z}_i^{(m)}$ to infer the graphical structures. The choice of structure learning algorithms can be tailored to the assumed graph class (e.g., potentially non-DAGs), and this step is orthogonal to our contribution. These structures characterize the interactions between all latent components across modalities, which is particularly desirable for biomedical applications.

## 4 IDENTIFICATION THEORY

As motivated in Section 3, we address the component-wise identifiability of latent components $z_i^{(m)}$.

---

[1]Please see Appendix A for details on the notion of identifiability.

**Remarks on the problem.** Identification for multimodal distributions often leverages the structure among the available modalities. However, component-wise identification, especially in the general nonparametric setting, is challenging. Daunhawer et al. (2023); Von Kügelgen et al. (2021); Yao et al. (2023) require certain information redundancy: the information of the latent variables should be fully shared and preserved by the observations of at least two modalities – that is, we can express $\mathbf{z}^{(m)}$ as functions of $\mathbf{x}^{(m_1)}$ and $\mathbf{x}^{(m_2)}$ individually. Moreover, the identification can only be achieved up to *subspaces* (i.e., groups of latent components) determined by the sharing pattern. Often, however, the latent components may not be fully shared by multiple modalities. For example, in health monitoring, while sleep monitoring data (e.g., sleep stages or duration) may not fully encode genetic predispositions, genetic factors may still influence sleep disorders, such as insomnia and circadian rhythm disruptions. In this case, the subspace identification may fall short of providing detailed interpretations of biomedical systems and the mechanisms encoded in the graphical structures over individual causal components.

For work that achieves component-wise identifiability, Morioka & Hyvarinen (2023; 2024) assume that the latent distribution $p(\{\mathbf{z}^{(m)}\}_{m=1}^M)$ follows an exponential family form with additive causal influences from multiple parents, which may be restrictive in general cases. For instance, in brain imaging studies, fMRI data and EEG data capture different neural activities, and the interactions between brain regions are often highly nonlinear. Clearly, for general multimodal distributions (Figure 2), we cannot access the information redundancy assumed in Daunhawer et al. (2023); Von Kügelgen et al. (2021); Yao et al. (2023) and the nicely-behaved latent causal models in parametric assumptions (Morioka & Hyvarinen, 2023; 2024).

**Our high-level approach.** We divide the problem into two parts: we first identify latent subspaces $\mathbf{z}^{(m)}$ (Section 4.1) and further disentangle identified subspaces into components $z_i^{(m)}$ (Section 4.2). For the subspace identification, we only assume that the information of the subspace $\mathbf{z}^{(m)}$ is preserved in its corresponding observation $\mathbf{x}^{(m)}$ and exerts sufficient influence on other modalities' observations $\mathbf{x}^{(-m)}$, thus weakening the redundancy assumption in previous work (Daunhawer et al., 2023; Yao et al., 2023). For the component-wise identification, we leverage a natural notion of structural sparsity in the literature (Zheng et al., 2022; Lachapelle et al., 2024) – the dependency among all the modalities should be explained with a minimal number of causal edges among latent subspaces $\{\mathbf{z}^{(m)}\}_{m=1}^M$. This allows us to further disentangle each subspace into components, without resorting to parametric assumptions (Morioka & Hyvarinen, 2023; 2024).

**Notations.** We denote the dimensionality and the component indices of a given argument with $d(\cdot)$ and $I(\cdot)$, respectively. The notation $-m$ represents the complement of modality $m$, while superscripts and subscripts enclosed in parentheses, such as $(m)$, explicitly index modality $m$. We denote sub-matrices using the notation $[\cdot]_{R,C}$, where $R$ and $C$ are index sets corresponding to row and column selections, respectively. In this notation, setting $R$ (or $C$) to : indicates the inclusion of all indices along the corresponding dimension.

## 4.1 IDENTIFYING LATENT SUBSPACES

As previously discussed, we now provide the subspace identifiability. Formally, we would like to show that the estimated latent subspace $\hat{\mathbf{z}}^{(m)}$ for any modality $m$ and its true counterpart $\mathbf{z}^{(m)}$ are equivalent up to an invertible map $h^{(m)}(\cdot)$, i.e., $\hat{\mathbf{z}}^{(m)} = h^{(m)}(\mathbf{z}^{(m)})$.

Given the data-generating process Eq. (2), the task is to remove modality-specific information $\boldsymbol{\eta}^{(m)}$ from the observational data $\mathbf{x}^{(m)}$ while retaining the latent variables $\mathbf{z}^{(m)}$ causally related to other modalities. In light of this, we express the relations between the latent variables $\mathbf{z}^{(m)}$ and the observation of its own modality $\mathbf{x}^{(m)}$ and other modalities $\mathbf{x}^{(-m)}$ as Eq. (3).

$$\mathbf{x}^{(m)} = g_{\mathbf{x}^{(m)}}(\mathbf{z}^{(m)}, \boldsymbol{\eta}^{(m)}), \quad \mathbf{x}^{(-m)} = \tilde{g}_{\mathbf{x}^{(-m)}}(\mathbf{z}^{(m)}, \tilde{\boldsymbol{\eta}}^{(-m)}), \tag{3}$$

where $\tilde{\boldsymbol{\eta}}^{(-m)}$ denotes all the information necessary to generate the complement group $\mathbf{x}^{(-m)}$ beyond $\mathbf{z}^{(m)}$. [2] Consequently, $\tilde{\boldsymbol{\eta}}^{(-m)}$ may admit causal/statistical relations with $\mathbf{z}^{(m)}$. We denote the joint map of $g_{\mathbf{x}^{(m)}}$ and $\tilde{g}_{\mathbf{x}^{(-m)}}$ as $\tilde{g}^{(m)} : (\mathbf{z}^{(m)}, \boldsymbol{\eta}^{(m)}, \tilde{\boldsymbol{\eta}}^{(-m)}) \mapsto \mathbf{x}$.

---

[2] We use $\tilde{\cdot}$ to differentiate $\tilde{\boldsymbol{\eta}}^{(-m)}$ from a collection of modality-specific variables $\boldsymbol{\eta}$ defined in Eq. (2).

**Condition 4.1** (Subspace Identifiability Conditions)**.**

A1 [Smoothness & Invertibility]: The generating functions $g_{\mathbf{x}^{(m)}}$ and $\tilde{g}^{(m)}$ are smooth and have smooth inverse functions.

A2 [Linear Independence]: The generating function $\tilde{g}_{\mathbf{x}^{(-m)}}$ is smooth and its Jacobian columns corresponding to $\mathbf{z}^{(m)}$ (i.e., $[\boldsymbol{J}_{\tilde{g}_{\mathbf{x}^{(-m)}}}]_{:,I(\mathbf{z}^{(m)})}$) are linearly independent almost anywhere.

**Discussion on the conditions.** Condition 4.1-A1 requires that the information of the latent variables $\mathbf{z}^{(m)}$ is preserved in its observation $\mathbf{x}^{(m)}$, so that the identification of latent variables is well-defined (Hyvarinen et al., 2019; Khemakhem et al., 2020a; Von Kügelgen et al., 2021; Kong et al., 2023; Yao et al., 2023; Daunhawer et al., 2023). Since this holds for any modality $m$, the observations $\mathbf{x}^{(-m)}$ should collectively preserve the information of other modalities $\mathbf{z}^{(-m)}$.

Condition 4.1-A2 formalizes the notation of a minimal connectivity over modalities: $\mathbf{z}^{(m)}$ should also exert sufficient influence on other modalities $\mathbf{z}^{(-m)}$, so that the other modality observations $\mathbf{x}^{(-m)}$ could be informative to identify $\mathbf{z}^{(m)}$. This condition excludes degenerate scenarios where the causal influences between modalities are nearly negligible and is equivalent to local invertibility of $\mathbf{z}^{(m)}$, which is strictly weaker than the global invertibility assumption in previous work (Daunhawer et al., 2023; Von Kügelgen et al., 2021; Yao et al., 2023) (e.g., $y = x^2$ is locally invertible but not globally so), as discussed earlier.

**Theorem 4.2** (Subspace Identifiability)**.** *Let $\boldsymbol{\theta} := \{g_{\mathbf{x}^{(m)}}, \tilde{g}_{\mathbf{z}^{(-m)}}, p(\boldsymbol{\epsilon}^{(m)}), p(\tilde{\boldsymbol{\epsilon}}^{(-m)})\}_{m=1}^{M}$ and $\hat{\boldsymbol{\theta}} := \{\hat{g}_{\mathbf{x}^{(m)}}, \hat{\tilde{g}}_{\mathbf{z}^{(-m)}}, p(\hat{\boldsymbol{\epsilon}}^{(m)}), p(\hat{\tilde{\boldsymbol{\epsilon}}}^{(-m)})\}_{m=1}^{M}$ be two specifications of the data-generating process in Eq. (3). Suppose that they generate identical observational distributions (i.e., $p(\mathbf{x}) = \hat{p}(\mathbf{x})$), $\boldsymbol{\theta}$ satisfies Condition 4.1, and $\hat{\boldsymbol{\theta}}$ satisfies Condition 4.1-A1. The latent subspace $\hat{\mathbf{z}}^{(m)}$ for any group $m$ and its counterpart $\mathbf{z}^{(m)}$ are equivalent up to an invertible map $h^{(m)}(\cdot)$, i.e., $\hat{\mathbf{z}}^{(m)} = h^{(m)}(\mathbf{z}^{(m)})$.*

**Interpretation and proof sketch.** Theorem 4.2 states that one can disentangle the modality-specific information $\boldsymbol{\eta}^{(m)}$ and the latent variables $\mathbf{z}^{(m)}$ contained in the observation $\mathbf{x}^{(m)}$ (which is a mixture of both). To achieve this, we leverage the fact that $\boldsymbol{\eta}^{(m)}$ has no influence on other modalities $\mathbf{x}^{(-m)}$, while $\mathbf{z}^{(m)}$ has a non-trivial influence on $\mathbf{x}^{(-m)}$, as characterized in Condition 4.1-A2. This crucial distinction provides sufficient footprints to disentangle these two subspaces for each modality, yielding the intended result.

## 4.2 Identifying Latent Components

Proceeding from the subspace identifiability (Theorem 4.2), we now further disentangle each subspace into individual components $z_i^{(m)}$ as outlined in Section 3. As foreshadowed, our key condition entails the sparsity of the graphical structures between modalities. Such dependency structures are captured in the generating function $g_{\mathbf{z}}$ defined component-wise in Eq. (1), in particular its partial derivatives. We now introduce Condition 4.3, which facilitates component-wise identification.

**Additional notations.** We denote the indices of the non-zero matrix entries by $\text{Supp}(\cdot)$. We denote the collection of partial derivatives among all latent components $\frac{\partial z_i^{(m)}}{\partial z_j^{(n)}}$ as a matrix function $\boldsymbol{G}(\mathbf{z}, \boldsymbol{\epsilon}) \in \mathbb{R}^{d(\mathbf{z}) \times d(\mathbf{z})}$. We adopt $\text{diag}(\cdot)$ to denote matrices consisting of equally-sized square matrices on its diagonal and define $\boldsymbol{T}$ to possess the structure $\boldsymbol{T} = \text{diag}(\boldsymbol{T}_1, \ldots, \boldsymbol{T}_M)$ with invertible $\boldsymbol{T}_m \in \mathbb{R}^{d(\mathbf{z}^{(m)}) \times d(\mathbf{z}^{(m)})}$. We denote the class of generalized permutation matrices of dimensionality $d(\mathbf{z})$ as $\mathcal{P}(d(\mathbf{z}))$.

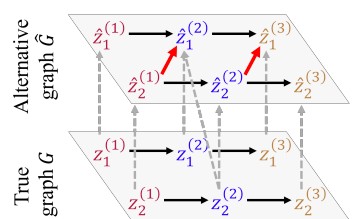

Figure 3: **Denser graph $\hat{G}$.**

**Condition 4.3** (Component Identifiability Conditions)**.** Over the domain of $(\mathbf{z}, \boldsymbol{\epsilon})$, for any modality $m$ and any $\boldsymbol{T} \notin \mathcal{P}(d(\mathbf{z}))$, we have

$$\sum_{m \neq n \in [M]} \left\| \boldsymbol{T}_m^{-1} [\boldsymbol{G}]_{(m),(n)} \boldsymbol{T}_n \right\|_0 > \sum_{m \neq n \in [M]} \left\| [\boldsymbol{G}]_{(m),(n)} \right\|_0. \tag{4}$$

**Discussion on the conditions.** Condition 4.3 stipulates sparse cross-modality causal connections among latent components $\mathbf{z}$. Under this condition, if a latent component $\hat{z}_i^{(m)}$ is a function of two components $\hat{z}_j^{(m)}$ and $\hat{z}_k^{(m)}$ (when component-wise identification breaks down), the cross-modality causal connections in $\boldsymbol{G}$ are guaranteed to be denser than those in $\hat{\boldsymbol{G}}$. We give a simple example to aid intuition in Figure 3: for a causal graph with three modalities, $z_1^{(1)} \to z_1^{(2)} \to z_1^{(3)}$ and $z_2^{(1)} \to z_2^{(2)} \to z_2^{(3)}$, suppose that $\hat{z}_1^{(2)}$ is a non-trivial mixture of $z_1^{(2)}$ and $z_2^{(2)}$ and other components are correctly identified, i.e., $[\hat{z}_1^{(1)}, \hat{z}_2^{(1)}, \hat{z}_1^{(2)}, \hat{z}_2^{(2)}, \hat{z}_1^{(3)}, \hat{z}_2^{(3)}] = [z_1^{(1)}, z_2^{(1)}, h(z_1^{(2)}, z_2^{(2)}), z_2^{(2)}, z_1^{(3)}, z_2^{(3)}]$. As a consequence, the alternative causal graph $\hat{\boldsymbol{G}}$ would include additional edges $\hat{z}_2^{(1)} \to \hat{z}_1^{(2)}$ and $\hat{z}_2^{(2)} \to \hat{z}_1^{(3)}$, giving rise to a strictly denser graph. In Theorem 4.4, we show that this sparse structure could give us the desired component-wise identifiability under a proper sparse regularization constraint. The availability of multiple modalities greatly improves the feasibility of such sparsity conditions, especially with a large number of modalities, because the entanglement is limited within a single modality (owing to Theorem 4.1) and all other modalities can be leveraged to provide space for sparse connections.

Sparsity conditions have been embraced by the causal representation learning community (Lachapelle et al., 2024; Moran et al., 2022; Fumero et al., 2023; Xu et al., 2024). Especially relevant to our work is Zheng et al. (2022). As discussed above, we are obliged to deal with causal structures among all latent variables. In contrast, Zheng et al. (2022) assumes the sparsity of the causal connections between the latent variables and the observed variables – the directions (from the latent to the observed variables) are given and the children are directly observed. Notably, sparse properties manifest in biomedical systems of our interest, including gene regulatory networks (Milo et al., 2002; Babu et al., 2004; Nacher & Akutsu, 2013; Liu et al., 2011), metabolic systems (West et al., 2002; Banavar et al., 1999), and other living systems (Busiello et al., 2017), evidencing the plausibility of Condition 4.3 for biomedical applications.

**Theorem 4.4** (Component-wise Identifiability)**.** *Let* $\boldsymbol{\theta} := (\{g_{\mathbf{x}^{(m)}}, g_{\mathbf{z}^{(m)}}, p(\boldsymbol{\epsilon}^{(m)})\}_{m=1}^M)$ *and* $\hat{\boldsymbol{\theta}} := (\{\hat{g}_{\mathbf{x}^{(m)}}, \hat{g}_{\mathbf{z}^{(m)}}, \hat{p}(\boldsymbol{\epsilon}^{(m)})\}_{m=1}^M)$ *be two specifications of the data-generating process in Eq. (1) and Eq. (2). Suppose that they generate identical observational distributions (i.e., $p(\mathbf{x}) = \hat{p}(\mathbf{x})$) and $\boldsymbol{\theta}$ satisfies Condition 4.1 and Condition 4.3. If $\hat{\boldsymbol{\theta}}$ satisfies the following sparse regularization condition:*

$$\sum_{m \neq n \in [M]} \left\| [\hat{\boldsymbol{G}}]_{(m),(n)} \right\|_0 \leq \sum_{m \neq n \in [M]} \left\| [\boldsymbol{G}]_{(m),(n)} \right\|_0, \tag{5}$$

*each component $z_i^{(m)}$ and its counterpart $\hat{z}_{\pi(i)}^{(m)}$ are equivalent up to an invertible map $h(\cdot)$, i.e., $\hat{z}_{\pi(i)}^{(m)} = h(z_i^{(m)})$ under a permutation $\pi$ over $[d(\mathbf{z}^{(m)})]$.*

**Interpretation and proof sketch.** The key idea of Theorem 4.4 is that for sparse causal graphs (as characterized in Condition 4.3), the mixing of latent components in any modality would introduce unnecessary causal edges connecting the other modalities. As the sparsity regularization Eq. (5) selects alternative models $\hat{\boldsymbol{\theta}}$ that are not denser than the model $\boldsymbol{\theta}$, the mixing within each modality would be excluded. Consequently, each latent component $\hat{z}_i^{(m)}$ is a function of a unique component $z_j^{(m)}$, yielding the desired component-wise identifiability.

**Implications.** In the context of biomedical applications, Theorem 4.4 indicates that under appropriate constraints, each component $\hat{z}_i^{(m)}$ in our estimation uniquely captures the information of an intrinsic biomedical factor behind the medical measurements (e.g., genetic predisposition). Therefore, the learned representation enjoys strong interpretability under theoretical guarantees, which is often lacking in existing biomedical models, as noted in Section 2. Theorem 4.2 and Theorem 4.4 provide insights for practical model design, which we employ in our architecture in Section 5.

**Shared latent variables.** Certain applications may involve latent variables that are shared across modalities. In such scenarios, we can employ contrastive learning objectives and the associated theoretical guarantees in previous work (Yao et al., 2023; Daunhawer et al., 2023; Von Kügelgen et al., 2021) as a pre-processing procedure and treat such shared latent variables as separate modalities in our implementation. Please refer to Appendix C.3, C.4, and E for detailed discussion and results.

## 5 ESTIMATION MODEL ARCHITECTURES

Given the identifiability results, we further propose an estimation framework (shown in Figure 4) that enforces the proposed assumptions as constraints to identify the latent variables in each modality.

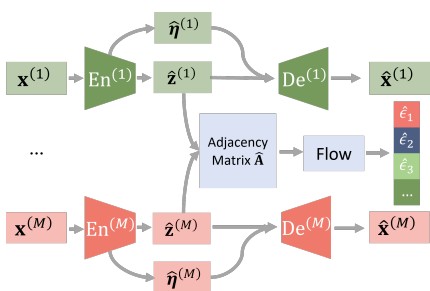

Figure 4: **Estimation framework**. Given multi-modal observations $(\mathbf{x}^{(1)}, \ldots, \mathbf{x}^{(M)})$, the latent variables and domain-specific information in modality $m$ are inferred as $\hat{\mathbf{z}}^{(m)}$ and $\hat{\eta}^{(m)}$ by individual encoders. The observations are then reconstructed with corresponding decoders as $\hat{\mathbf{x}}^{(m)}$. We enforce independence conditions by minimizing the KL divergence term $D_{\text{KL}}\left([\{\hat{\eta}^{(m)}\}_{m=1}^M, \{\hat{\epsilon}_i\}_{i=1}^{d(\mathbf{z})}]||\mathcal{N}(\mathbf{0}, \mathbf{I})\right)$. We enforce the sparsity constraint by minimizing the $\mathcal{L}_1$ norm in the inferred adjacency matrix $\hat{\mathbf{A}}$.

**Encoder and decoder.** Each modality $\mathbf{x}^{(m)}$ is given as an input to the corresponding encoder and outputs the estimated latent $\hat{\mathbf{z}}^{(m)}$ and domain-specific information $\hat{\eta}^{(m)}$. They are then concatenated and passed to the corresponding decoder to reconstruct the observations as $\hat{\mathbf{x}}^{(m)}$. The reconstruction loss is calculated using the mean squared error (MSE) as $\mathcal{L}_{\text{Recon}} = \sum_{m=1}^M ||\mathbf{x}^{(m)} - \hat{\mathbf{x}}^{(m)}||_2^2$.

**Conditional independence constraints.** Given Eq (3), we enforce the conditional independence condition $\mathbf{x}^{(m)} \perp\!\!\!\perp \mathbf{x}^{(n)} \mid \mathbf{z}^{(m)}$ and the independence condition on $\eta^{(m)} \perp\!\!\!\perp \mathbf{z}^{(m)}$ by enforcing independence among components in $\gamma = [\{\hat{\eta}^{(m)}\}_{m=1}^M, \{\hat{\epsilon}_i\}_{i=1}^{d(\mathbf{z})}]$. Such equivalence is shown in Propositions 5.1 and 5.2, and proofs are provided in Appendix B. Specifically, we minimize the KL divergence loss between the posterior and a Gaussian prior distribution: $\mathcal{L}_{\text{Ind}} = D_{\text{KL}}(p(\gamma)||\mathcal{N}(\mathbf{0}, \mathbf{I}))$.

**Proposition 5.1.** *[Conditional Independence Condition] Let $\mathbf{x}^{(m)}$ and $\mathbf{x}^{(n)}$ be two different multi-modal observations. $\mathbf{z}^{(m)} \subset \mathbf{z}$ are the set of block-identifying latent variables, and $\eta^{(m)} \subset \eta$ are domain-specific information in modality $m$. We have $\mathbf{x}^{(m)} \perp\!\!\!\perp \mathbf{x}^{(n)} \mid \mathbf{z}^{(m)} \iff \eta^{(m)} \perp\!\!\!\perp \eta^{(n)}$.*

**Proposition 5.2.** *[Independent Noise Condition] Let $\mathbf{z}$ and $\eta$ be the block-identified latent variables and domain-specific information, respectively, across all modalities. Denote $\epsilon$ as the exogenous variables in the latent causal structure. We have $\eta \perp\!\!\!\perp \mathbf{z} \iff \eta \perp\!\!\!\perp \epsilon$.*

**Sparsity regularization.** We use flow to estimate the exogenous variables $\epsilon$ in Eq. (1) and implement the causal relations through a learnable adjacency matrix $\hat{\mathbf{A}}$. The binary values in $\hat{\mathbf{A}}$ represent the causal generation process between latent variables, e.g. $\hat{A}_{i,j} = 1$ indicates $\hat{z}_j$ is the parent of $\hat{z}_i$, while $\hat{A}_{i,j} = 0$ means $\hat{z}_j$ dose not contribute to the generation of $\hat{z}_i$. For each component $\hat{z}_i$, we select its parents $\text{Pa}(\hat{z}_i)$ based on the adjacency matrix, and apply the flow transformation to get $\hat{\epsilon}_i$.

To encourage sparsity among the latent variables $\hat{\mathbf{z}}$, we impose a regularization term on the learned adjacency matrix. Based on the sparsity assumption, the optimal causal graph should be the minimal one that still allows the model to accurately match the ground truth generative distribution. To achieve this, we reduce the dependencies between different components of $\hat{\mathbf{z}}$ by adding a $\mathcal{L}_1$ penalty on the adjacency matrix, s.t., $\mathcal{L}_{\text{Sp}} = ||\hat{\mathbf{A}}||_1$.

**Optimization.** The model parameters are optimized using the combination objective:

$$\mathcal{L} = \alpha_{\text{Recon}}\mathcal{L}_{\text{Recon}} + \alpha_{\text{Ind}}\mathcal{L}_{\text{Ind}} + \alpha_{\text{Sp}}\mathcal{L}_{\text{Sp}}. \tag{6}$$

## 6 EXPERIMENT RESULTS

To evaluate the efficacy of our proposed method, we conduct extensive experiments on (1) numerical, (2) synthetic and (3) real-world datasets. In terms of the baselines, we compare our method with: (1) BetaVAE (Higgins et al., 2017), which does not consider causal relations in the latent space. (2) CausalVAE (Yang et al., 2020), which considers the causally related latent variables with

a single modality. (3) Multimodal contrastive learning (MCL) (Daunhawer et al., 2023), which recovers the shared latent factors from multiple modalities. Throughout the experiments, we consider the following underlined evaluation metrics: (1) Mean Correlation Coefficient (MCC) measures how well the estimated latent variables match the true ones, with an MCC of 1 indicating perfect identifiability up to component-wise transformations. (2) R2 measures the proportion of variance in the ground truth latent that is explained by the estimated latent, with a value of 1 indicating that all variance is explained. (3) Structural Hamming Distance (SHD) compares graphs by their adjacency matrices, where a lower SHD indicates stronger similarity between graphs.

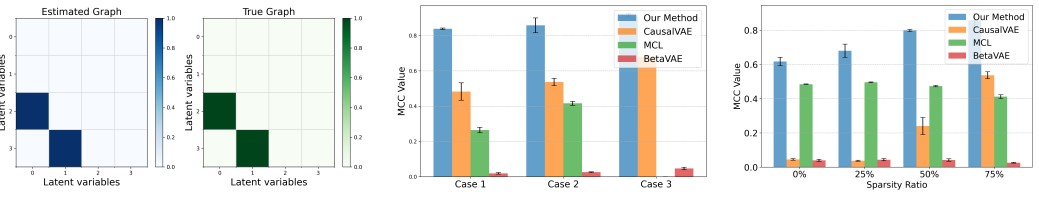

(a) Causal comparison between estimated and true graphs (SHD=0).

(b) Comparison of the identifiability result in different cases.

(c) Identifiability result under different sparsity ratios.

Figure 5: Numerical experiment results. (a) Successful recovery of the inter-modal causal graph. (b) Baseline comparisons in different cases. (c) Result of sparsity ablation study.

## 6.1 NUMERICAL DATASET

**Setup.** In the numerical simulations, we consider three cases with different numbers of modalities and inter-modal causal relations. Case 1: 15-dimensional observations across two modalities, each with two latent and one exogenous variable. Case 2: 20-dimensional observations across two modalities, each with three latent and one exogenous variable. Case 3: 30-dimensional observations across four modalities, each with two latent and one exogenous variable. The nonparametric mixing function is simulated by a random MLP with LeakyReLU units, and the inter-modal latent variables are sparse causally related. The detailed data-generation process is provided in Appendix D.1.

**Results and ablation.** Figure 5 shows the identifiability results in different cases, where the high MCC indicates the successful recovery of the latent variables. The inter-modal causal relations are successfully recovered (SHD=0) and the causal comparison result in case 1 is shown in Figure 5(a). The identifiability comparison results are shown in Figure 5(b) (MCL is not applicable in case 3 due to the two-modality constraint). CausalVAE requires additional supervision signals to establish identifiability, and MCL assumes content invariance and can only block-identify latent variables. In general, these baselines neither account for the multimodal setting nor the modality-specific latent variables, and therefore do not recover the latent variables.

As an ablation study, we further show the consequences of violating the sparsity assumptions to validate our theorem. Based on case 2, we create four types of datasets with different sparsity ratios and report the MCC in each scenario in Figure 5(c). The sparsity ratio represents the ratio of existing causal links to all possible causal links between modality-specific latent variables. A value of 0 indicates that the latent variables between modalities are fully connected, while higher values correspond to sparser connections. The result shows that identifiability can be better achieved with a higher sparsity ratio, and our framework outperforms other baselines in all scenarios.

## 6.2 SYNTHETIC DATASET: VARIANT MNIST

**Setup.** We manually create a variant of the MNIST dataset to encode causal relationships between different modalities, using colored MNIST (Arjovsky et al., 2019) and fashion MNIST (Xiao et al., 2017) as two different modalities. In colored MNIST, the horizontal position of the digit influences the image transparency. This horizontal position further serves as a causal factor for the vertical position of the fashion items in the fashion MNIST, which influences image grayscale. This design ensures a structured causal dependency across modalities while maintaining a non-deterministic mapping. Further data descriptions are provided in the Appendix D.2.

**Results.** Table 2 presents the results of the identifiability comparison, where higher MCC and R2 indicate better performance of our method. BetaVAE does not explicitly model causal relationships among latent variables, leading to suboptimal recovery in our setting.

Table 2: The results of MNIST dataset.

|  | MCL | BetaVAE | CausalVAE | Ours |
|---|---|---|---|---|
| R2 | $0.79 \pm 6e{-}5$ | $0.68 \pm 2e{-}3$ | $0.50 \pm 4e{-}3$ | $\mathbf{0.90} \pm 9e{-}5$ |
| MCC | $0.63 \pm 2e{-}6$ | $0.53 \pm 1e{-}3$ | $0.74 \pm 2e{-}3$ | $\mathbf{0.85} \pm 3e{-}5$ |

CausalVAE, which relies on additional supervision, fails to recover the latent variables effectively.

### 6.3 REAL-WORLD DATASET: HUMAN PHENOTYPE

The human phenotype dataset (Shilo et al., 2021) is a large-scale, longitudinal collection of phenotypic profiles from a diverse global population. It includes comprehensive human health data and provides a comprehensive view of health and disease factors. The dataset contains various types of participant information, categorized into tabular, time series, and image data. Specifically, it includes health information across 30 modalities, such as blood tests, anthropometry, fundus imaging, etc. Detailed data descriptions can be found in the Appendix D.3.

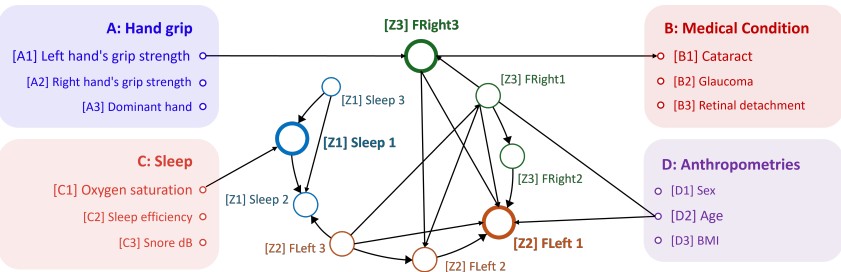

Figure 6: Causal analysis results across different modalities, including hand grip, medical conditions, sleep, and anthropometrics. We ran the causal algorithm on all variables, but for clarity only report the causal relationships that have direct connections to the estimated latent variables.

In this work, we focus on the time-series sleep monitoring dataset (*Sleep*) and the fundus imaging dataset for both left and right eyes (*FLeft* and *FRight*) to estimate the latent factors (Z1, Z2, Z3) underlying each modality. We applied the PC algorithm (Spirtes et al., 2001) to discover causal relationships between the estimated latent variables and other four additional tabular modalities (A, B, C, D), providing an implicit evaluation on the effectiveness. The result with direct causal relations is shown in Figure 6, where variables from the same modality have the same color and different modalities have different colors.

A key finding is that the discovered causal relationships are consistent with findings from medical research. For example, *Sleep 1* shows a direct causal relationship with *Oxygen saturation*, suggesting that sleep conditions may influence blood oxygen levels. This observation is consistent with previous studies (Wali et al., 2020). In addition, the fundus-related latent variables *FRight 1* and *FLeft 1* have a direct causal relationship with *Age*, suggesting that aging plays an important role in changes in retinal health (Ege et al., 2002; Einbock et al., 2005). Interestingly, the fundus image of the right eye has a direct causal relationship with the grip strength of the left hand, as recently demonstrated in biomedical research (Bikbov et al., 2023; Qiu et al., 2020).

## 7 CONCLUSION AND LIMITATIONS

In this work, we develop a theoretically grounded framework for recovering latent causal variables from multi-modal observations. Extensive experimental results on synthetic and real-world datasets demonstrate the practical effectiveness of our approach. **Limitations:** Empirically, our framework assumes prior knowledge of the number of latent variables in each modality, which may be unrealistic in real-world scenarios. Additionally, a detailed evaluation against the quantitative benchmarks used in biomedical models remains an area for future exploration.

## 8    ACKNOWLEDGMENT

We would also like to acknowledge the support from NSF Award No. 2229881, AI Institute for Societal Decision Making (AI-SDM), the National Institutes of Health (NIH) under Contract R01HL159805, and grants from Quris AI, Florin Court Capital, and MBZUAI-WIS Joint Program. The work of L. Kong is supported in part by NSF DMS-2134080 through an award to Y. Chi. P. Stojanov was supported in part by the National Cancer Institute (NCI) grant number: K99CA277583-01, and funding from the Eric and Wendy Schmidt Center at the Broad Institute of MIT and Harvard.

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

# Supplementary Materials for "Causal Representation Learning from Multimodal Biomedical Observations"

Contents

## A    NOTATION AND TERMINOLOGY

We summarize the notations used throughout the paper in Table 3.

| **Index** | |
| --- | --- |
| $m, n$ | Modality index |
| $i, j$ | Variable element index |
| $I(\cdot)$ | Component indices of a given argument |
| $d(\cdot)$ | Dimensionality indices of a given argument |
| **Variable** | |
| $\mathbf{x}^{(m)}$ | Observation/measurement in each modality |
| $\mathbf{z}^{(m)}$ | Causally related latent variables in each modality |
| $\mathbf{x}^{(m)}, \mathbf{x}^{(-m)}$ | One specific observation in modality $m$, and the rest of others |
| $\mathbf{z}^{(m)}, \mathbf{z}^{(-m)}$ | One specific latent variables in modality $m$, and the rest of others |
| $\eta$ | Domain-specific information |
| $\epsilon$ | Mutually independent exogenous variables |
| $\hat{z}_i$ | Estimated latent variables over $z_i$ |
| $\hat{\mathbf{x}}^{(m)}$ | Reconstructed observation in modality $m$ |
| $\mathrm{Pa}(z_i^{(m)})$ | Set of direct cause nodes/parents of variable $z_i^{(m)}$ |
| **Function and Hyperparameter** | |
| $g_{z_i^m}$ | Causal function among latent variables |
| $g_{\mathbf{x}^{(m)}}$ | Nonparametric mixing function in modality $m$ |
| $h$ | Invertible mapping from true latent to the estimated latent |
| $p$ | Distribution function (e.g., $p_{z_i}$ is the distribution of $z_i$) |
| $\alpha_{\mathrm{Recon}}, \alpha_{\mathrm{Ind}}, \alpha_{\mathrm{Sp}}$ | Weights in the combination objective |

Table 3: List of notations.

**Notions of identifiability.**    Following the literature on ICA (Hyvarinen & Morioka, 2016; Hyvarinen et al., 2019; Comon, 1994) and causal representation learning (Yao et al., 2023; Von Kügelgen et al., 2021; Daunhawer et al., 2023), we assume that the generating function $g_{\mathbf{x}^{(m)}}$ (Eq. (2)) is an invertible map from $(\mathbf{z}^{(m)}, \eta^{(m)})$ to $\mathbf{x}^{(m)}$ (Condition 4.1). Under this invertibility assumption, given the value of $\mathbf{x}^{(m)}$, one can perfectly determine the value of $\mathbf{z}^{(m)}$, which is essentially the posterior $p(\mathbf{z}|\mathbf{x})$ (a point mass here). Here, $\mathbf{z}^{(m)}$ is a function of $\mathbf{x}^{(m)}$, and the identifiability of $g$ gives rise to the result that $\mathbf{z}^{(m)}$ values can be identified from $\mathbf{x}^{(m)}$.

In statistics, to show identifiability, we start with equal distributions $p_{\phi_1} = p_{\phi_2}$ to derive the equivalence of the parameters $\phi_1 = \phi_2$. In our case, since the functions $g_{\mathbf{x}^{(m)}}, \hat{g}_{\mathbf{x}^{(m)}}$ are invertible, one can reason about the relation between the two specifications $g_{\mathbf{x}^{(m)}}$ and $\hat{g}_{\mathbf{x}^{(m)}}$ through a composition $h := \hat{g}_{\mathbf{x}^{(m)}}^{-1} \circ g_{\mathbf{x}^{(m)}}$. For instance, $h$ is the identity when the two specifications are identical. Similarly, in this work, we start with equal values of $\mathbf{x}$ to establish the relation between $\mathbf{z}$ and $\hat{\mathbf{z}}$.

## B    CONSTRAINTS IN THE ESTIMATION FRAMEWORK

Here we provide the proofs for the constraints utilized in the estimation framework.

**Proposition B.1.** *[Conditional Independence Condition] Let $\mathbf{x}^{(m)}$ and $\mathbf{x}^{(n)}$ be two different multimodal observations. $\mathbf{z}^{(m)} \subset \mathbf{z}$ are the set of block-identifying latent variables, and $\eta^{(m)} \subset \eta$ are domain-specific information in modality m. We have*

$$\mathbf{x}^{(m)} \perp\!\!\!\perp \mathbf{x}^{(n)} \mid \mathbf{z}^{(m)} \iff \eta^{(m)} \perp\!\!\!\perp \eta^{(n)}. \tag{7}$$

*Proof.* Given the data generation process in Eq. (2), the following assumptions hold true for any $m, n \in [M]$: (1) $\mathbf{z}^{(m)} \perp\!\!\!\perp \eta^{(m)}$; (2) $\mathbf{z}^{(m)} \perp\!\!\!\perp \eta^{(n)}$; (3) $\eta^{(m)} \perp\!\!\!\perp \mathbf{x}^{(n)}$.

**Sufficient condition.** Given LHS of Eq. (7), we have

$$p(\mathbf{x}^{(m)}, \mathbf{x}^{(n)} \mid \mathbf{z}^{(m)}) = p(\mathbf{x}^{(m)} \mid \mathbf{z}^{(m)})p(\mathbf{x}^{(n)} \mid \mathbf{z}^{(m)}).$$

$$\xrightarrow{RHS} p(\mathbf{x}^{(m)}, \mathbf{x}^{(n)} \mid \mathbf{z}^{(m)}) = \frac{p(\mathbf{x}^{(m)}, \mathbf{x}^{(n)}, \mathbf{z}^{(m)})}{p(\mathbf{z}^{(m)})} = \frac{p(\eta^{(m)}, \eta^{(n)}, \mathbf{z}^{(m)})}{p(\mathbf{z}^{(m)})}|\det\frac{\partial\eta^{(m)}}{\partial\mathbf{x}^{(m)}}||\det\frac{\partial\eta^{(n)}}{\partial\mathbf{x}^{(n)}}|$$

$$= p(\eta^{(m)}, \eta^{(n)} \mid \mathbf{z}^{(m)})|\det\frac{\partial\eta^{(m)}}{\partial\mathbf{x}^{(m)}}||\det\frac{\partial\eta^{(n)}}{\partial\mathbf{x}^{(n)}}|$$

$$\xrightarrow{LHS} p(\mathbf{x}^{(m)} \mid \mathbf{z}^{(m)})p(\mathbf{x}^{(n)} \mid \mathbf{z}^{(m)}) = \frac{p(\mathbf{x}^{(m)}, \mathbf{z}^{(m)})}{p(\mathbf{z}^{(m)})}\frac{p(\mathbf{x}^{(n)}, \mathbf{z}^{(m)})}{p(\mathbf{z}^{(m)})}$$

$$= \frac{p(\eta^{(m)}, \mathbf{z}^{(m)})}{p(\mathbf{z}^{(m)})}|\det\frac{\partial\eta^{(m)}}{\partial\mathbf{x}^{(m)}}|\frac{p(\eta^{(n)}, \mathbf{z}^{(m)})}{p(\mathbf{z}^{(m)})}|\det\frac{\partial\eta^{(n)}}{\partial\mathbf{x}^{(n)}}|$$

$$= p(\eta^{(m)} \mid \mathbf{z}^{(m)})p(\eta^{(n)} \mid \mathbf{z}^{(n)})|\det\frac{\partial\eta^{(m)}}{\partial\mathbf{x}^{(m)}}||\det\frac{\partial\eta^{(n)}}{\partial\mathbf{x}^{(n)}}|$$

Thus we have

$$p(\eta^{(m)}, \eta^{(n)}|\mathbf{z}^{(m)}) = p(\eta^{(m)}|\mathbf{z}^{(m)})p(\eta^{(n)}|\mathbf{z}^{(n)}) \Rightarrow p(\eta^{(m)}, \eta^{(n)}) = p(\eta^{(m)})p(\eta^{(n)}) \Rightarrow \eta^{(m)} \perp\!\!\!\perp \eta^{(n)}$$

**Necessary condition.** Given RHS of Eq. (7) and above conclusion, we have

$$p(\mathbf{x}^{(m)} \mid \mathbf{z}^{(m)}) = p(\eta^{(m)})|\det\frac{\partial\eta^{(m)}}{\partial\mathbf{x}^{(m)}}|, \quad p(\mathbf{x}^{(n)} \mid \mathbf{z}^{(m)}) = p(\eta^{(n)})|\det\frac{\partial\eta^{(n)}}{\partial\mathbf{x}^{(n)}}|$$

$$\xrightarrow{Multiplication} p(\mathbf{x}^{(m)} \mid \mathbf{z}^{(m)})p(\mathbf{x}^{(n)} \mid \mathbf{z}^{(n)}) = p(\eta^{(m)})p(\eta^{(n)})|\det\frac{\partial\eta^{(m)}}{\partial\mathbf{x}^{(m)}}||\det\frac{\partial\eta^{(n)}}{\partial\mathbf{x}^{(n)}}|$$

$$= p(\eta^{(m)}, \eta^{(n)})|\det\frac{\partial\eta^{(m)}}{\partial\mathbf{x}^{(m)}}||\det\frac{\partial\eta^{(n)}}{\partial\mathbf{x}^{(n)}}| = \frac{p(\eta^{(m)}, \eta^{(n)}, \mathbf{z}^{(m)})}{p(\mathbf{z}^{(m)})}|\det\frac{\partial\eta^{(m)}}{\partial\mathbf{x}^{(m)}}||\det\frac{\partial\eta^{(n)}}{\partial\mathbf{x}^{(n)}}| = \frac{p(\mathbf{x}^{(m)}, \mathbf{x}^{(n)}, \mathbf{z}^{(m)})}{p(\mathbf{z}^{(m)})}$$

$$\Rightarrow p(\mathbf{x}^{(m)} \mid \mathbf{z}^{(m)})p(\mathbf{x}^{(n)} \mid \mathbf{z}^{(m)}) = p(\mathbf{x}^{(m)}, \mathbf{x}^{(n)} \mid \mathbf{z}^{(m)}) \Rightarrow \mathbf{x}^{(m)} \perp\!\!\!\perp \mathbf{x}^{(n)} \mid \mathbf{z}^{(m)}$$

$$(8)$$

$$\square$$

**Proposition B.2** (Independent Noise Condition). *Let $\mathbf{z}$ and $\eta$ be the block-identified latent variables and domain-specific information, respectively, across all modalities. Denote $\epsilon$ as the exogenous variables in the latent causal structure. We have*

$$\eta \perp\!\!\!\perp \mathbf{z} \Longleftrightarrow \eta \perp\!\!\!\perp \epsilon. \tag{9}$$

*Proof.* Given the causal function in Eq. (1), we have $p(\mathbf{z}) = p(\epsilon)|\det\frac{\partial\epsilon}{\partial\mathbf{z}}|$.

**Sufficient condition.** Suppose $(\mathbf{z}, \eta) = h(\epsilon, \eta)$ and $\eta \perp\!\!\!\perp \mathbf{z}$, we have

$$p(\mathbf{z}, \eta) = p(\epsilon, \eta)|\det\frac{\partial\epsilon}{\partial\mathbf{z}}| \Rightarrow p(\mathbf{z})p(\eta) = p(\epsilon, \eta)|\det\frac{\partial\epsilon}{\partial\mathbf{z}}| \Rightarrow p(\epsilon)p(\eta)|\det\frac{\partial\epsilon}{\partial\mathbf{z}}| = p(\epsilon, \eta)|\det\frac{\partial\epsilon}{\partial\mathbf{z}}|$$

$$\Rightarrow p(\epsilon)p(\eta) = p(\epsilon, \eta) \Rightarrow \eta \perp\!\!\!\perp \epsilon$$

$$(10)$$

**Necessary condition.** Suppose $(\mathbf{z}, \eta) = h(\epsilon, \eta)$ and $\eta \perp\!\!\!\perp \epsilon$, we have

$$p(\mathbf{z}, \eta) = p(\epsilon, \eta)|\det\frac{\partial\epsilon}{\partial\mathbf{z}}| \Rightarrow p(\mathbf{z}, \eta) = p(\epsilon)|\det\frac{\partial\epsilon}{\partial\mathbf{z}}|p(\eta) \Rightarrow p(\mathbf{z}, \eta) = p(\mathbf{z})p(\eta) \Rightarrow \eta \perp\!\!\!\perp \mathbf{z}$$

$$(11)$$

$$\square$$

## C  IDENTIFIABILITY THEORY

### C.1  PROOF FOR THEOREM 4.2

We present the proof for Theorem 4.2. For ease of reference, we duplicate Condition 4.1 and Theorem 4.2 below.

**Condition 4.1** (Subspace Identifiability Conditions).

A1 [Smoothness & Invertibility]: The generating functions $g_{\mathbf{x}^{(m)}}$ and $\tilde{g}^{(m)}$ are smooth and have smooth inverse functions.

A2 [Linear Independence]: The generating function $\tilde{g}_{\mathbf{x}^{(-m)}}$ is smooth and its Jacobian columns corresponding to $\mathbf{z}^{(m)}$ (i.e., $[\boldsymbol{J}_{\tilde{g}_{\mathbf{x}^{(-m)}}}]_{:,I(\mathbf{z}^{(m)})}$) are linearly independent almost anywhere.

**Theorem 4.2** (Subspace Identifiability). *Let* $\boldsymbol{\theta} := \{g_{\mathbf{x}^{(m)}}, \tilde{g}_{\mathbf{z}^{(-m)}}, p(\boldsymbol{\epsilon}^{(m)}), p(\tilde{\boldsymbol{\epsilon}}^{(-m)})\}_{m=1}^{M}$ *and* $\hat{\boldsymbol{\theta}} := \{\hat{g}_{\mathbf{x}^{(m)}}, \hat{\tilde{g}}_{\mathbf{z}^{(-m)}}, p(\hat{\boldsymbol{\epsilon}}^{(m)}), p(\hat{\tilde{\boldsymbol{\epsilon}}}^{(-m)})\}_{m=1}^{M}$ *be two specifications of the data-generating process in Eq. (3). Suppose that they generate identical observational distributions (i.e.,* $p(\mathbf{x}) = \hat{p}(\mathbf{x})$*),* $\boldsymbol{\theta}$ *satisfies Condition 4.1, and* $\hat{\boldsymbol{\theta}}$ *satisfies Condition 4.1-A1. The latent subspace* $\hat{\mathbf{z}}^{(m)}$ *for any group* $m$ *and its counterpart* $\mathbf{z}^{(m)}$ *are equivalent up to an invertible map* $h^{(m)}(\cdot)$*, i.e.,* $\hat{\mathbf{z}}^{(m)} = h^{(m)}(\mathbf{z}^{(m)})$.

*Proof.* Given the generating processes in Eq. (2) and Eq. (1), we can express any observed group $\mathbf{x}^{(m)}$ and its complement $\mathbf{x}^{(-m)} := \mathbf{x} \setminus \mathbf{x}^{(m)}$ as two views of the latent variables of group $m$:

$$\mathbf{x}^{(m)} := g^{(m)}(\mathbf{z}^{(m)}, \boldsymbol{\eta}^{(m)}), \tag{12}$$

$$\mathbf{x}^{(-m)} := g^{(-m)}(\mathbf{z}^{(m)}, \tilde{\boldsymbol{\eta}}^{(-m)}),, \tag{13}$$

where $\boldsymbol{\eta}^{(m)}$ stands for exogenous variables for the group $\mathbf{x}^{(m)}$ and $\tilde{\boldsymbol{\eta}}^{(-m)}$ represents all the information necessary to generate the complement group $\mathbf{x}^{(-m)}$ beyond $\mathbf{z}^{(m)}$.

Following the classic definition of identifiability, we define two specifications $\boldsymbol{\theta} = \{g_{\mathbf{x}^{(m)}}, g_{\mathbf{z}^{(m)}}, p(\boldsymbol{\epsilon}^{(m)})\}_{m=1}^{M}$ and $\hat{\boldsymbol{\theta}} := \{\hat{g}_{\mathbf{x}^{(m)}}, \hat{g}_{\mathbf{z}^{(m)}}, \hat{p}(\boldsymbol{\epsilon}^{(m)})\}_{m=1}^{M}$ that fit the observation distribution $p(\mathbf{x})$. To show the identifiability in terms of the functions in $\boldsymbol{\theta}$ and $\hat{\boldsymbol{\theta}}$, we show that given the same $\mathbf{x}^{(m)}$ value the identifiability between $\mathbf{z}^{(m)}$ and $\hat{\mathbf{z}}^{(m)}$.

Thus, the subspace identification is equivalent to show that for each group $m$, the estimated latent variable $\hat{\mathbf{z}}^{(m)}$ and the true counterpart are related via an invertible map $h$, i.e., $\hat{\mathbf{z}}^{(m)} = h(\mathbf{z}^{(m)})$.

Eq. (12) and the invertibility of the map $(\mathbf{z}, \boldsymbol{\eta}^{(m)}, \tilde{\boldsymbol{\eta}}^{(-m)}) \mapsto (\mathbf{x}^{(m)}, \mathbf{x}^{(-m)})$ (Condition 4.1-A1) give rise to an invertible map $\tilde{h} : (\hat{\mathbf{z}}^{(m)}, \hat{\boldsymbol{\eta}}^{(m)}, \hat{\tilde{\boldsymbol{\eta}}}^{(-m)}) \mapsto (\mathbf{z}^{(m)}, \boldsymbol{\eta}^{m}, \tilde{\boldsymbol{\eta}}^{(-m)})$.

The matched observed distribution between the true and the estimated models for the generating process Eq. (13) yields that

$$g^{(-m)}(\mathbf{z}^{(m)}, \tilde{\boldsymbol{\eta}}^{(-m)}) = \hat{g}^{(-m)}(\hat{\mathbf{z}}^{(m)}, \hat{\tilde{\boldsymbol{\eta}}}^{(-m)}). \tag{14}$$

Plugging in $\tilde{h}$ gives

$$\hat{g}^{(-m)}(\hat{\mathbf{z}}^{(m)}, \hat{\tilde{\boldsymbol{\eta}}}^{(-m)}) = g^{(-m)}\left(\left[\tilde{h}\left(\hat{\mathbf{z}}^{(m)}, \hat{\boldsymbol{\eta}}^{(m)}, \hat{\tilde{\boldsymbol{\eta}}}^{(-m)}\right)\right]_{I(\mathbf{z}^{(m)}), I(\tilde{\boldsymbol{\eta}}^{(-m)})}\right). \tag{15}$$

where we adopt $I(\cdot)$ to indicate the indices of its argument.

For any $i \in [d(\mathbf{x}^{(m)})]$ and $j \in [d(\hat{\boldsymbol{\eta}}^{(m)})]$, we take partial derivative w.r.t. $\hat{\eta}_j^{(m)}$ on both sides of Eq. (15):

$$\underbrace{\frac{\partial[\hat{g}^{(-m)}]_i}{\partial[\hat{\eta}^{(m)}]_j}}_{=0} = \frac{\partial[g^{(-m)}]_i}{\partial[\hat{\eta}^{(m)}]_j}. \tag{16}$$

The left-hand side of Eq. (15) equals to zero because $\hat{g}^{(-m)}$ is not a function of $\hat{\boldsymbol{\eta}}^{(m)}$.

Therefore, expanding the right-hand side of Eq. (15) gives:

$$\sum_{k \in I(\mathbf{z}^{(-m)}) \cup I(\tilde{\boldsymbol{\eta}}^{(-m)})} \frac{\partial [g^{(-m)}]_i}{\partial [\tilde{h}]_k} \cdot \frac{\partial [\tilde{h}]_k}{\partial [\hat{\eta}^{(m)}]_j} = \sum_{k \in I(\mathbf{z}^{(-m)})} \frac{\partial [g^{(-m)}]_i}{\partial [\tilde{h}]_k} \cdot \frac{\partial [\tilde{h}]_k}{\partial [\hat{\eta}^{(m)}]_j} = 0. \qquad (17)$$

The first equality in Eq. (17) is due to the fact that $\tilde{\boldsymbol{\eta}}^{(-m)}$ is a function of $\mathbf{x}^{(-m)}$ and varying $\hat{\eta}^{(m)}$ doesn't vary $\mathbf{x}^{(-m)}$ ($\hat{\eta}^{(m)}$ is a function of $\mathbf{x}^{(m)}$ thanks to the invertibility of $\hat{g}^{(m)}$), i.e., $\frac{\partial [\tilde{\eta}^{(-m)}]_k}{\partial [\hat{\eta}^{(m)}]_j} = 0$.

Condition 4.1-A2 implies that the matrix $\left( \frac{\partial [g^{(-m)}]_i}{\partial [\tilde{h}]_k} \right)_{i,k}$ has a full column rank. Therefore, its null space contains only a zero vector, which, together with Eq. (17), implies that $\frac{\partial [z^{(m)}]_k}{\partial [\hat{\eta}^{(m)}]_j} = 0$. Consequently, given the generating process Eq. (12) and the invertibility of $g^{(m)}$ and $\hat{g}^{(m)}$ (Condition 4.1-A1), the estimated latent variable $\hat{\mathbf{z}}^{(m)}$ and the true latent variable $\mathbf{z}^{(m)}$ are related via an invertible map, as desired.

$\square$

## C.2 Proof for Theorem 4.4

We present the proof for Theorem 4.4. For ease of reference, we duplicate Condition 4.3 and Theorem 4.4.

**Condition 4.3** (Component Identifiability Conditions). Over the domain of $(\mathbf{z}, \boldsymbol{\epsilon})$, for any modality $m$ and any $\boldsymbol{T} \notin \mathcal{P}(d(\mathbf{z}))$, we have

$$\sum_{m \neq n \in [M]} \left\| \boldsymbol{T}_m^{-1} [\boldsymbol{G}]_{(m),(n)} \boldsymbol{T}_n \right\|_0 > \sum_{m \neq n \in [M]} \left\| [\boldsymbol{G}]_{(m),(n)} \right\|_0. \qquad (4)$$

**Theorem 4.4** (Component-wise Identifiability). *Let* $\boldsymbol{\theta} := (\{g_{\mathbf{x}^{(m)}}, g_{\mathbf{z}^{(m)}}, p(\boldsymbol{\epsilon}^{(m)})\}_{m=1}^M)$ *and* $\hat{\boldsymbol{\theta}} := (\{\hat{g}_{\mathbf{x}^{(m)}}, \hat{g}_{\mathbf{z}^{(m)}}, \hat{p}(\boldsymbol{\epsilon}^{(m)})\}_{m=1}^M)$ *be two specifications of the data-generating process in Eq. (1) and Eq. (2). Suppose that they generate identical observational distributions (i.e., $p(\mathbf{x}) = \hat{p}(\mathbf{x})$) and $\boldsymbol{\theta}$ satisfies Condition 4.1 and Condition 4.3. If $\hat{\boldsymbol{\theta}}$ satisfies the following sparse regularization condition:*

$$\sum_{m \neq n \in [M]} \left\| [\hat{\boldsymbol{G}}]_{(m),(n)} \right\|_0 \leq \sum_{m \neq n \in [M]} \left\| [\boldsymbol{G}]_{(m),(n)} \right\|_0, \qquad (5)$$

*each component* $z_i^{(m)}$ *and its counterpart* $\hat{z}_{\pi(i)}^{(m)}$ *are equivalent up to an invertible map* $h(\cdot)$, *i.e.,* $\hat{z}_{\pi(i)}^{(m)} = h(z_i^{(m)})$ *under a permutation* $\pi$ *over* $[d(\mathbf{z}^{(m)})]$.

*Proof.* Given Theorem 4.2, Condition 4.1 implies that the estimated group-wise latent variable $\hat{\mathbf{z}}^{(m)}$ is related to the true variable $\mathbf{z}^{(m)}$ through an invertible transformation $h^{(m)}$, i.e.,

$$\hat{\mathbf{z}}^{(m)} = h^{(m)}(\mathbf{z}^{(m)}). \qquad (18)$$

It follows that the Jacobian matrix $\boldsymbol{T}_{\frac{\partial \hat{\mathbf{z}}}{\partial \mathbf{z}}}$ can be arranged into a block-diagonal matrix, in which diagonal block $m$ corresponds to a Jacobian matrix $\boldsymbol{T}_{\frac{\partial \hat{\mathbf{z}}^{(m)}}{\partial \mathbf{z}^{(m)}}}$. Then, the goal is to prove that these diagonal blocks are actually generalized permutation matrices, whose each column only contains one nonzero entry.

We divide the proof into several steps for the sake of exposition. At step 1, we derive an equivalence relation between the estimation model $(\hat{g}_z, \hat{g}_x)$ and the true model $(g_z, g_x)$. At step 2, we apply Theorem 4.2 to the equivalence to characterize the relation between the true and the estimated graph structure. At step 3, we leverage the sparsity condition (Condition 4.3) to reason about the identifiability of each component $z_i^{(m)}$ for $m \in [M]$ and $i \in [d(z^{(m)})]$.

**Step 1.**  The generating process in Eq. (1) and the subspace identification Eq. (18) imply

$$\hat{g}_z(\hat{\mathbf{z}}, \hat{\boldsymbol{\epsilon}}) = h \circ g_z(\mathbf{z}, \boldsymbol{\epsilon}), \tag{19}$$

where $h$ is defined as the Cartesian product of individual $h^{(m)}$ functions.

Taking partial derivatives w.r.t, $z_i$ of both sides of Eq. (19) yields:

$$\begin{bmatrix} \boldsymbol{G}_{\frac{\partial \hat{\mathbf{z}}}{\partial \hat{\mathbf{z}}}} & \boldsymbol{T}_{\frac{\partial \hat{\mathbf{z}}}{\partial \hat{\boldsymbol{\epsilon}}}} \end{bmatrix} \begin{bmatrix} \boldsymbol{T}_{\frac{\partial \hat{\mathbf{z}}}{\partial \mathbf{z}}} \\ \boldsymbol{T}_{\frac{\partial \hat{\boldsymbol{\epsilon}}}{\partial \mathbf{z}}} \end{bmatrix} = \boldsymbol{T}_{\frac{\partial \hat{\mathbf{z}}}{\partial \mathbf{z}}} \boldsymbol{G}_{\frac{\partial \mathbf{z}}{\partial \mathbf{z}}}. \tag{20}$$

Each $\boldsymbol{T}$ matrix is the Jacobian matrix consisting of the corresponding partial derivatives. We use $\boldsymbol{G}_{\frac{\partial \mathbf{z}}{\partial \mathbf{z}}}$ to denote the derivatives from the function $g_z$ which encodes the dependence structure among $z$ components. The same applies to $\boldsymbol{G}_{\frac{\partial \hat{\mathbf{z}}}{\partial \hat{\mathbf{z}}}}$. As discussed above, the matrix $\boldsymbol{T}_{\frac{\partial \hat{\mathbf{z}}}{\partial \mathbf{z}}}$ has a block-diagonal structure (after proper permutations) with block $m$ corresponding to the Jacobian matrix of $h^{(m)}$. Moreover, the matrix $\boldsymbol{T}_{\frac{\partial \hat{\mathbf{z}}}{\partial \hat{\boldsymbol{\epsilon}}}}$ is strictly diagonal due to the generating function Eq. (1).

**Step 2.**  In this step, we simplify Eq. (20) to derive the relation between the estimated graph structures and true graph structures encoded in $\boldsymbol{G}_{\frac{\partial \hat{\mathbf{z}}}{\partial \hat{\mathbf{z}}}}$ and $\boldsymbol{G}_{\frac{\partial \mathbf{z}}{\partial \mathbf{z}}}$ respectively.

First, we note that the $\boldsymbol{T}_{\frac{\partial \hat{\boldsymbol{\epsilon}}}{\partial \mathbf{z}}}$ is also block-diagonal w.r.t. the groups. To see this, we compute the partial derivatives therein as follows: $\frac{\partial \hat{\epsilon}_i^{(m)}}{\partial z_j^{(n)}} = \frac{\partial \hat{\epsilon}_i^{(m)}}{\partial \hat{z}_i^{(m)}} \frac{\partial \hat{z}_i^{(m)}}{\partial z_j^{(n)}}$, where we denote that output of $\hat{g}_z$ with $\tilde{\hat{z}}$ in the derivative. Due to the equivalent relation $\mathbf{z} = \tilde{\mathbf{z}}$ (Eq. (1)), we have $\frac{\partial \hat{z}_i^{(m)}}{\partial z_j^{(n)}} = \frac{\partial \hat{z}_i^{(m)}}{\partial z_j^{(n)}}$ which is zero for distinct groups $m \neq n$ (Eq. (18)). It follows that

$$\frac{\partial \hat{\epsilon}_i^{(m)}}{\partial z_j^{(n)}} = 0, \ m \neq n. \tag{21}$$

Therefore, we have shown that $\boldsymbol{T}_{\frac{\partial \hat{\boldsymbol{\epsilon}}}{\partial \mathbf{z}}}$ is block-diagonal w.r.t. the groups.

This structure allows us to simplify Eq. (20) to directly characterize the relation between the two graphical structures $\boldsymbol{G}_{\frac{\partial \hat{\mathbf{z}}}{\partial \hat{\mathbf{z}}}}$ and $\boldsymbol{G}_{\frac{\partial \mathbf{z}}{\partial \mathbf{z}}}$. In particular, since $\boldsymbol{T}_{\frac{\partial \hat{\boldsymbol{\epsilon}}}{\partial \mathbf{z}}}$ is block-diagonal and $\boldsymbol{T}_{\frac{\partial \hat{\mathbf{z}}}{\partial \hat{\boldsymbol{\epsilon}}}}$ is diagonal, the off-diagonal blocks on the left-hand side of Eq. (20) are determined by $\boldsymbol{G}_{\frac{\partial \hat{\mathbf{z}}}{\partial \hat{\mathbf{z}}}} \boldsymbol{T}_{\frac{\partial \hat{\mathbf{z}}}{\partial \mathbf{z}}}$. Therefore, it follows from Eq. (20):

$$\begin{bmatrix} \boldsymbol{G}_{\frac{\partial \hat{\mathbf{z}}}{\partial \hat{\mathbf{z}}}} \boldsymbol{T}_{\frac{\partial \hat{\mathbf{z}}}{\partial \mathbf{z}}} \end{bmatrix}_{(m),(n)} = \begin{bmatrix} \boldsymbol{T}_{\frac{\partial \hat{\mathbf{z}}}{\partial \mathbf{z}}} \boldsymbol{G}_{\frac{\partial \mathbf{z}}{\partial \mathbf{z}}} \end{bmatrix}_{(m),(n)}, \ m \neq n, \tag{22}$$

where we adopt subscripts $(m)$ to denote the block for group $m$.

On account of the block-diagonal structure of $\boldsymbol{T}_{\frac{\partial \hat{\mathbf{z}}}{\partial \mathbf{z}}}$, the left-hand side of Eq. (22) can be expressed as follows:

$$\begin{bmatrix} \boldsymbol{G}_{\frac{\partial \hat{\mathbf{z}}}{\partial \hat{\mathbf{z}}}} \boldsymbol{T}_{\frac{\partial \hat{\mathbf{z}}}{\partial \mathbf{z}}} \end{bmatrix}_{(m),(n)} = \begin{bmatrix} \boldsymbol{G}_{\frac{\partial \hat{\mathbf{z}}}{\partial \hat{\mathbf{z}}}} \end{bmatrix}_{(m),:} \begin{bmatrix} \boldsymbol{T}_{\frac{\partial \hat{\mathbf{z}}}{\partial \mathbf{z}}} \end{bmatrix}_{:,(n)} = \begin{bmatrix} \boldsymbol{G}_{\frac{\partial \hat{\mathbf{z}}}{\partial \hat{\mathbf{z}}}} \end{bmatrix}_{(m),(n)} \begin{bmatrix} \boldsymbol{T}_{\frac{\partial \hat{\mathbf{z}}}{\partial \mathbf{z}}} \end{bmatrix}_{(n),(n)}. \tag{23}$$

Analogously, the right-hand side of Eq. (22) can be expressed as:

$$\begin{bmatrix} \boldsymbol{T}_{\frac{\partial \hat{\mathbf{z}}}{\partial \mathbf{z}}} \boldsymbol{G}_{\frac{\partial \mathbf{z}}{\partial \mathbf{z}}} \end{bmatrix}_{(m),(n)} = \begin{bmatrix} \boldsymbol{T}_{\frac{\partial \hat{\mathbf{z}}}{\partial \mathbf{z}}} \end{bmatrix}_{(m),:} \begin{bmatrix} \boldsymbol{G}_{\frac{\partial \mathbf{z}}{\partial \mathbf{z}}} \end{bmatrix}_{:,(n)} = \begin{bmatrix} \boldsymbol{T}_{\frac{\partial \hat{\mathbf{z}}}{\partial \mathbf{z}}} \end{bmatrix}_{(m),(m)} \begin{bmatrix} \boldsymbol{G}_{\frac{\partial \mathbf{z}}{\partial \mathbf{z}}} \end{bmatrix}_{(m),(n)}. \tag{24}$$

It follows from Eq. (22), Eq. (23), and Eq. (24) that

$$\begin{bmatrix} \boldsymbol{G}_{\frac{\partial \hat{\mathbf{z}}}{\partial \hat{\mathbf{z}}}} \end{bmatrix}_{(m),(n)} \begin{bmatrix} \boldsymbol{T}_{\frac{\partial \hat{\mathbf{z}}}{\partial \mathbf{z}}} \end{bmatrix}_{(n),(n)} = \begin{bmatrix} \boldsymbol{T}_{\frac{\partial \hat{\mathbf{z}}}{\partial \mathbf{z}}} \end{bmatrix}_{(m),(m)} \begin{bmatrix} \boldsymbol{G}_{\frac{\partial \mathbf{z}}{\partial \mathbf{z}}} \end{bmatrix}_{(m),(n)}$$

$$\implies$$

$$\begin{bmatrix} \boldsymbol{G}_{\frac{\partial \hat{\mathbf{z}}}{\partial \hat{\mathbf{z}}}} \end{bmatrix}_{(m),(n)} = \begin{bmatrix} \boldsymbol{T}_{\frac{\partial \hat{\mathbf{z}}}{\partial \mathbf{z}}} \end{bmatrix}_{(m),(m)} \begin{bmatrix} \boldsymbol{G}_{\frac{\partial \mathbf{z}}{\partial \mathbf{z}}} \end{bmatrix}_{(m),(n)} \begin{bmatrix} \boldsymbol{T}_{\frac{\partial \mathbf{z}}{\partial \mathbf{z}}} \end{bmatrix}_{(n),(n)}. \tag{25}$$

Eq. (25) relates the true off-diagonal ($m \neq n$) structure $\begin{bmatrix} \boldsymbol{G}_{\frac{\partial \mathbf{z}}{\partial \mathbf{z}}} \end{bmatrix}_{(m),(n)}$ and its estimated counterpart $\begin{bmatrix} \boldsymbol{G}_{\frac{\partial \hat{\mathbf{z}}}{\partial \hat{\mathbf{z}}}} \end{bmatrix}_{(m),(n)}$.

**Step 3.** We now reason about the component-wise identifiability within each modality through the sparsity of the off-diagonal regions.

The component-wise identifiability is equivalent to that each block sub-matrix $\left[\boldsymbol{T}_{\frac{\partial \hat{z}}{\partial z}}\right]_{(m),(m)}$ is a generalized permutation matrix, each row/column of which contains only one nonzero element. Suppose that this was not the case, then it would follow from Eq. (25) and Condition 4.3 that

$$
\sum_{m \neq n \in [M]} \left\|\left[\boldsymbol{G}_{\frac{\partial \hat{z}}{\partial z}}\right]_{(m),(n)}\right\|_0 = \sum_{m \neq n \in [M]} \left\|\left[\boldsymbol{T}_{\frac{\partial \hat{z}}{\partial z}}\right]_{(m),(m)} \left[\boldsymbol{G}_{\frac{\partial z}{\partial z}}\right]_{(m),(n)} \left[\boldsymbol{T}_{\frac{\partial z}{\partial z}}\right]_{(n),(n)}\right\|_0
$$
$$
\underset{\text{Condition 4.3}}{>} \sum_{m \neq n \in [M]} \left\|\left[\boldsymbol{G}_{\frac{\partial z}{\partial z}}\right]_{(m),(n)}\right\|_0,
$$
(26)

which would violate the sparsity constraint Eq. (5).

That is, the component $\hat{z}_{\hat{i}}^{(m)}$ cannot functionally influence components in $U^{(m)}$ other than $\mathbf{z}_i^{(m)}$. Therefore, we have shown that each block sub-matrix $\left[\boldsymbol{T}_{\frac{\partial \hat{z}}{\partial z}}\right]_{(m),(m)}$ is a generalized permutation matrix. Consequently, we have a bijection $\hat{z}_{\hat{i}}^{(m)} = h_{\hat{i}}^{(m)}(z_i^{(m)})$. Since this holds for any group $m$ and any component $i$, we have arrived at the desired conclusion.

$\square$

## C.3 EXTENDED THEOREM 4.2 AND ITS PROOF

We restate Theorem C.5 from Yao et al. (2023), which we invoke in our Theorem C.7. We drop the entropy regularization term in Yao et al. (2023), since we assume the invertibility of estimated functions $\hat{g}^{(m)}$ directly.

**Definition C.1** (View-Specific Encoders). The *view-specific encoders* $R := \{r_k : \mathcal{X}_k \to \mathcal{Z}_{S_k}\}_{k \in V}$ consist of smooth functions mapping from the respective observation spaces to the view-specific latent space, where the dimension of the $k^{\text{th}}$ latent space $|S_k|$ is assumed known for all $k \in V$.

**Definition C.2** (Selection). A selection $\oslash$ operates between two vectors $a \in \{0,1\}^d, b \in \mathbb{R}^d$ s.t.

$$
a \oslash b := [b_j : a_j = 1, j \in [d]]
$$

**Definition C.3** (Content Selectors). The content selectors $\Phi := \{\phi(i,k)\}_{V_i \in \mathcal{V}, k \in V_i}$ with $\phi^{(i,k)} \in \{0,1\}^{|d(\mathbf{z}^{(m)})|}$ perform selection C.2 on the encoded information: for any subset $V_i \subset [M]$ and view $k \in V_i$ we have the selected representation: $\phi(i,m) \oslash \hat{\mathbf{z}}^{(m)}$ with $\left\|\phi^{(i,k)}\right\|_0 = \left\|\phi^{(i,k')}\right\|_0$ for all $V_i \in \mathcal{V}, k, k' \in V_i$.

**Definition C.4** (Information-Sharing Regularizer). The following regularizer penalizes the $\ell_0$-norm $\|\cdot\|_0$ of the content selectors $\Phi$: $\text{Reg}(\Phi) := -\sum_{V_i \in \mathcal{V}} \sum_{k \in V_i} \left\|\phi^{(i,k)}\right\|_0$.

**Theorem C.5** (View-Specific Encoder for Identifiability (Yao et al., 2023)). *Let $R := \{\hat{g}_{(m)}\}_{m=1}^M$ and $\Phi$ respectively be the generating functions and content selectors (Definition C.3) that solve the following constrained optimization problem:*

$$
\min \text{Reg}(\Phi) \qquad \textit{subject to:} \qquad R, \Phi \in \arg\min \mathcal{L}_{alignment}(R, \Phi),
$$
(27)

*where*

$$
\mathcal{L}_{alignment}(R, \Phi) = \sum_{\substack{V_i \in \mathcal{V}}} \sum_{\substack{m_1, m_2 \in V_i \\ k < k'}} \mathbb{E}\left[\left\|\phi(i,m_1) \oslash [\hat{g}^{(m_1)}]^{-1}(\mathbf{x}_k) - \phi(i,m_2) \oslash [\hat{g}^{m_2}]^{-1}(\mathbf{x}_{m_2})\right\|_2\right]
$$
(28)

*Then for any subset of modalities $V_i \subset [M]$ and any modality $m \in V_i$, $\phi(i,m) \oslash [\hat{g}^{(m)}]^{-1}$ identifies the shared subspace $\mathbf{z}^{(\cap_{m \in V_i} m)}$.*

**Definition C.6** (Reconstruction Loss). The following loss penalizes the deviation of the estimate $\hat{x}$ and its corresponding true counterpart $x$ in $\ell_2$ $L_{\text{recons}} := \mathbb{E}_{\mathbf{x}}(\mathbf{x} - \hat{\mathbf{x}})$.

**Additional notations.** We slightly abuse the notation to denote both sets and vectors with bold symbols $\mathbf{z}$. Let $\mathbf{z}^{(m \cap n)}$ be the set of latent components shared by modality $m$ and $n$, i.e., $\mathbf{z}^{(m \cap n)} := \mathbf{z}^{(m)} \cap \mathbf{z}^{(n)}$. Analogously, let $\mathbf{z}^{(m \setminus n)}$ be the set of latent components in modality $m$ that are not shared by $n$, i.e., $\mathbf{z}^{(m \setminus n)} := \mathbf{z}^{(m)} \setminus \mathbf{z}^{(n)}$.

**Theorem C.7** (Generalized Subspace Identifiability). *Let* $\{(g_{\mathbf{x}^{(m)}}, \tilde{g}_{\mathbf{x}^{(-m)}})\}_{m=1}^M$ *and* $\{(\hat{g}_{\mathbf{x}^{(m)}}, \hat{\tilde{g}}_{\mathbf{x}^{(-m)}})\}_{m=1}^M$ *be two specifications of the generating process Eq.* (3) *with potentially shared variables* $\mathbf{z}^{(m \cap n)}$ *over any two modalities* $m$ *and* $n$. *Suppose that they both match the observational distribution* $p(\mathbf{x}) = \hat{p}(\mathbf{x})$ *and satisfy Condition* 4.1. *Then any subspace* $\hat{\mathbf{z}}^{(m)}$, *shared subspace* $\hat{\mathbf{z}}^{(m \cap n)}$ *and their counterparts* $\mathbf{z}^{(m)}$, $\mathbf{z}^{(m \cap n)}$ *are equivalent up to invertible maps.*

*Proof.* We note that the latent model with shared latent variables across modalities can still be cast into Equation (3) and satisfies Condition 4.1. As a consequence, Theorem 4.2 gives us the subspace identification for each modality as in the disjoint case. Moreover, we can identify any blocks among modalities thanks to Theorem C.5. This concludes the proof. □

### C.4 Extended Theorem 4.4 and its Proof

**Additional notations and discussion.** The participation of multiple modalities requires a new definition of the shared blocks in $\mathbf{z}$ since the sharing structure could be nested and various numbers of modalities could share one partition. We partition the entire latent space $\mathbf{z}$ into disjoint blocks $\{\mathbf{z}^{(b)}\}_{b \in B}$, whose components $z$ have exactly the same modality membership $\mathcal{M}(z) := \{m \in [M] : z \in \mathbf{z}^{(m)}\}$. We define the $\mathbf{z}^{H(b)}$ as the smallest (the least components) *identified* block in $\mathbf{z}$ that contains $\mathbf{z}^{(b)}$. In the two-modal case, we have $B = \{(m \cap n), (m \setminus n), (n \setminus m)\}$ and $\mathbf{z}^{H(m \setminus n)} = \mathbf{z}^{(m)}$.

We denote $\mathbf{z}^{(b_1)} \prec \mathbf{z}^{(b_2)}$ if block $\mathbf{z}^{(b_1)}$ is shared by a strict subset of modalities that share $\mathbf{z}^{(b_2)}$, i.e., $\mathcal{M}(\mathbf{z}^{(b_1)}) \subsetneq \mathcal{M}(\mathbf{z}^{(b_2)})$. Therefore, we have either $\mathbf{z}^{H(b)} = \mathbf{z}^{(b)}$ (it is identifiable itself) or $\mathbf{z}^{(b)} \prec \mathbf{z}^{H(b)} \setminus \mathbf{z}^{(b)}$ (it is not identifiable by itself but belongs to an identifiable block $\mathbf{z}^{H(b)}$ together with a more deeply shared block $\mathbf{z}^{H(b)} \setminus \mathbf{z}^{(b)}$). We denote former blocks as $b^+ \in B^+$, i.e., $\mathbf{z}^{H(b^+)} = \mathbf{z}^{(b^+)}$, and the latter blocks as $b^- \in B^- = B \setminus B^+$. In the two-modal case, the modal-specific blocks $\mathbf{z}^{(m \setminus n)}, \mathbf{z}^{(n \setminus m)}$ are not identifiable themselves, and we have $\mathbf{z}^{(m \setminus n)} \prec \mathbf{z}^{(m \cap n)}$ and $\mathbf{z}^{(n \setminus m)} \prec \mathbf{z}^{(m \cap n)}$, and $B^+ = \{(m \cap n)\}$ and $B^- = \{(m \setminus n), (n \setminus m)\}$.

We note that all shared blocks $\mathbf{z}^{(b^+)}$ are identified. Thus their bijective indeterminacies are w.r.t, themselves, i.e., $\mathbf{z}^{(b^+)} \mapsto \hat{\mathbf{z}}^{(b^+)}$, which implies the square shape of their indeterminacy matrices $[\boldsymbol{T}_{\frac{\partial \hat{\mathbf{z}}}{\partial \mathbf{z}}}]_{(b^+),(b^+)}$. In contrast, the unidentifiable blocks $\hat{\mathbf{z}}^{(b^-)}$ can potentially receive the influence from all other blocks in its minimal block $\mathbf{z}^{H(b^-)}$. However, $\mathbf{z}^{(b^-)}$ do not influence the complement block $\hat{\mathbf{z}}^{H(b^-)} \setminus \hat{\mathbf{z}}^{(b^-)}$, since $\mathbf{z}^{(b^-)} \prec \mathbf{z}^{H(b^-)} \setminus \mathbf{z}^{(b^-)}$. For instance, $\hat{\mathbf{z}}^{(m \setminus n)}$ may receive influences from $\mathbf{z}^{(m \cap n)}$ and $\mathbf{z}^{(m \setminus n)}$. Consequently, their associated non-trivial indeterminacy matrices are $[\boldsymbol{T}_{\frac{\partial \hat{\mathbf{z}}}{\partial \mathbf{z}}}]_{(b^-),H(b^-)}$ (e.g., $[\boldsymbol{T}_{\frac{\partial \hat{\mathbf{z}}}{\partial \mathbf{z}}}]_{(m \setminus n),(m)}$). Thus, the indeterminacy matrix $\boldsymbol{T}$ can be expressed as $\boldsymbol{T} := \boldsymbol{T}_{\text{on}} + \boldsymbol{T}_{\text{off}}$, where The matrix $\boldsymbol{T}_{\text{on}}$ contains all the on-diagonal square invertible matrices $\boldsymbol{T}_{\text{on}} := \text{diag}(\boldsymbol{T}_{\text{on}}^1, \ldots, \boldsymbol{T}_{\text{on}}^{(|B|)})$ and $\boldsymbol{T}_{\text{off}}$ contains all the off-diagonal elements potentially nonzero in the regions $(b, H(b) \setminus b)$ for $b \in B$. We denote this class of matrix $\boldsymbol{T}$ as $\mathcal{T}$. We denote a set of blocks $E(b)$ whose memberships are either a strict superset or do not nest with $b$'s membership $E(b) := \{\tilde{b} \in B | \mathbf{z}^{(b)} \prec \mathbf{z}^{(\tilde{b})} \vee \left( \mathcal{M}(\mathbf{z}^{(b)} \not\subset \mathcal{M}(\mathbf{z}^{(\tilde{b})}) \wedge \mathcal{M}(\mathbf{z}^{(\tilde{b})} \not\subset \mathcal{M}(\mathbf{z}^{(b)}) \right)\}$. In the two-modality case, we have $E(m \setminus n) = \{(m \cap n), (n \setminus m)\}$. The regions $\{(E(b), b)\}_{b \in B}$ in the alternative graph $\hat{G}$ reveal identifiability of the latent variables. With these notations, we state the generalized component identification result in Theorem C.9.

**Condition C.8** (Generalized Component Identifiability Conditions). Over the domain of $(\mathbf{z}, \boldsymbol{\epsilon})$, for any modality $m$, for $\boldsymbol{T} \in \mathcal{T}$ and $\boldsymbol{T} \notin \mathcal{P}(d(\mathbf{z}))$, we have

$$\sum_{b \in B, \tilde{b} \in E(b)} \left\| [\boldsymbol{T}^{-1}]_{(\tilde{b}), H(\tilde{b})} [\boldsymbol{G}]_{H(\tilde{b}),(b)} [\boldsymbol{T}]_{(b),(b)} \right\|_0 > \sum_{b \in B, \tilde{b} \in E(b)} \left\| [\boldsymbol{G}]_{(\tilde{b}),(b)} \right\|_0. \qquad (29)$$

Notice that at the absence of the shared block $(m \cap n)$, Condition C.8 recovers Condition 4.3 where $E(m) = B \setminus \{m\}$ and $H(m \setminus n) = m$, and $H(n \setminus m) = n$.

**Theorem C.9** (Generalized Component-wise Identifiability). *Let* $\boldsymbol{\theta}$ := $(\{g_{\mathbf{x}^{(m)}}, g_{\mathbf{z}^{(m)}}, p(\boldsymbol{\epsilon}^{(m)})\}_{m=1}^{M})$ *and* $\hat{\boldsymbol{\theta}}$ := $(\{\hat{g}_{\mathbf{x}^{(m)}}, \hat{g}_{\mathbf{z}^{(m)}}, \hat{p}(\boldsymbol{\epsilon}^{(m)})\}_{m=1}^{M})$ *be two specifications of the data-generating process in Eq. (1) and Eq. (2) with potentially shared variables* $\mathbf{z}^{(m \cap n)}$ *over any modalities* $m$ *and* $n$. *Suppose that they generate identical observational distributions (i.e.,* $p(\mathbf{x}) = \hat{p}(\mathbf{x})$) *and* $\boldsymbol{\theta}$ *satisfies Condition 4.1 and Condition C.8. If* $\hat{\boldsymbol{\theta}}$ *satisfies the following condition:*

$$\sum_{b \in B, \tilde{b} \in E(b)} \left\| [\hat{\boldsymbol{G}}]_{(\tilde{b}),(b)} \right\|_0 \leq \sum_{b \in B, \tilde{b} \in E(b)} \left\| [\boldsymbol{G}]_{(\tilde{b}),(b)} \right\|_0, \tag{30}$$

*each component* $z_i^{(m)}$ *and its counterpart* $\hat{z}_{\pi(i)}^{(m)}$ *are equivalent up to an invertible map* $h(\cdot)$, *i.e.,* $\hat{z}_{\pi(i)}^{(m)} = h(z_i^{(m)})$ *under a permutation* $\pi$ *over* $[d(\mathbf{z}^{(m)})]$.

*Proof.* This proof closely follows that of Theorem 4.4. We illustrate the key discrepancies as follows.

We start with only two modalities $\mathbf{z}^{(m)}$ and $\mathbf{z}^{(n)}$ for simplicity and then move on to general cases.

**The structure of the indeterminacy matrix** $\boldsymbol{T}_{\frac{\partial \hat{\mathbf{z}}}{\partial \mathbf{z}}}$**.** Identical to Equation 20, we have the relationship between Jacobian matrices:

$$\begin{bmatrix} \boldsymbol{G}_{\frac{\partial \hat{\mathbf{z}}}{\partial \hat{\mathbf{z}}}} & \boldsymbol{T}_{\frac{\partial \hat{\mathbf{z}}}{\partial \hat{\boldsymbol{\epsilon}}}} \end{bmatrix} \begin{bmatrix} \boldsymbol{T}_{\frac{\partial \hat{\mathbf{z}}}{\partial \mathbf{z}}} \\ \boldsymbol{T}_{\frac{\partial \hat{\boldsymbol{\epsilon}}}{\partial \mathbf{z}}} \end{bmatrix} = \boldsymbol{T}_{\frac{\partial \hat{\mathbf{z}}}{\partial \mathbf{z}}} \boldsymbol{G}_{\frac{\partial \mathbf{z}}{\partial \mathbf{z}}}. \tag{31}$$

The presence of the shared block $\mathbf{z}^{(m \cap n)}$ alters the indeterminacy matrix $\boldsymbol{T}_{\frac{\partial \hat{\mathbf{z}}}{\partial \mathbf{z}}}$ – instead of the disjoint diagonal-block shape, $\boldsymbol{T}_{\frac{\partial \hat{\mathbf{z}}}{\partial \mathbf{z}}}$, the columns belonging to the shared variables $\mathbf{z}^{(m \cap n)}$ (shared between two modalities) are possibly nonzero over rows belonging to $\mathbf{z}^{(m \cap n)}$. That is, the shared variables $\mathbf{z}^{(m \cap n)}$ can still mix in the estimates of the two individual parts $\hat{\mathbf{z}}^{(m \setminus n)}$ and $\hat{\mathbf{z}}^{(n \setminus m)}$. However, since we have identified the subspace of $\mathbf{z}^{(m \cap n)}$, its estimates would not contain information of the individual blocks $\mathbf{z}^{(m \setminus n)}$ and $\mathbf{z}^{(n \setminus m)}$, rendering the blocks $\frac{\partial \hat{\mathbf{z}}^{(m \cap n)}}{\partial \mathbf{z}^{(m \setminus n)}} = 0$ and $\frac{\partial \hat{\mathbf{z}}^{(m \cap n)}}{\partial \mathbf{z}^{(n \setminus m)}} = 0$.

**The sparse connection among modalities.** The reasoning in **Step 2** in the proof of Theorem 4.4 implies that the structure of the matrix $\boldsymbol{T}_{\frac{\partial \hat{\boldsymbol{\epsilon}}}{\partial \mathbf{z}}}$ is consistent with that of the matrix $\boldsymbol{T}_{\frac{\partial \hat{\mathbf{z}}}{\partial \mathbf{z}}}$. That is, they have zero block matrices at the same positions. In particular, since the subspace identifiability in Theorem C.7 implies that the estimated shared variable $\hat{\mathbf{z}}^{(m \cap n)}$ and the modality-specific variable $\hat{\mathbf{z}}^{(n \setminus m)}$ are not influenced by the other modality-specific variables $\mathbf{z}^{(m \setminus n)}$, the same applies to the estimated exogenous variable $\hat{\boldsymbol{\epsilon}}^{(m \cap n)}$ and $\hat{\boldsymbol{\epsilon}}^{(m \setminus n)}$. This structure permits us to disregard $\boldsymbol{T}_{\frac{\partial \hat{\mathbf{z}}}{\partial \hat{\boldsymbol{\epsilon}}}}$ (an identity matrix) and $\boldsymbol{T}_{\frac{\partial \hat{\boldsymbol{\epsilon}}}{\partial \mathbf{z}}}$ on the left-hand side of Eq. (31) when computing a sub-matrix of the right-hand side product:

$$\left[ \boldsymbol{G}_{\frac{\partial \hat{\mathbf{z}}}{\partial \hat{\mathbf{z}}}} \boldsymbol{T}_{\frac{\partial \hat{\mathbf{z}}}{\partial \mathbf{z}}} \right]_{(n),(m \setminus n)} = \left[ \boldsymbol{T}_{\frac{\partial \hat{\mathbf{z}}}{\partial \mathbf{z}}} \boldsymbol{G}_{\frac{\partial \mathbf{z}}{\partial \mathbf{z}}} \right]_{(n),(m \setminus n)}. \tag{32}$$

We further divide the block $[(n), (m \setminus n)]$ into two blocks along their rows: $[(m \cap n), (m \setminus n)]$ and $[(n \setminus m), (m \setminus n)]$ that represent the influence from $\mathbf{z}^{(m \setminus n)}$ to $\hat{\mathbf{z}}^{(m \cap n)}$ and $\hat{\mathbf{z}}^{(n \setminus m)}$.

Expressing the block $[(m \cap n), (m \setminus n)]$ on the left-hand side of Eq. (32) gives:

$$\left[ \boldsymbol{G}_{\frac{\partial \hat{\mathbf{z}}}{\partial \hat{\mathbf{z}}}} \boldsymbol{T}_{\frac{\partial \hat{\mathbf{z}}}{\partial \mathbf{z}}} \right]_{(m \cap n),(m \setminus n)} = \left[ \boldsymbol{G}_{\frac{\partial \hat{\mathbf{z}}}{\partial \hat{\mathbf{z}}}} \right]_{(m \cap n),:} \left[ \boldsymbol{T}_{\frac{\partial \hat{\mathbf{z}}}{\partial \mathbf{z}}} \right]_{:,(m \setminus n)} = \left[ \boldsymbol{G}_{\frac{\partial \hat{\mathbf{z}}}{\partial \hat{\mathbf{z}}}} \right]_{(m \cap n),(m \setminus n)} \left[ \boldsymbol{T}_{\frac{\partial \hat{\mathbf{z}}}{\partial \mathbf{z}}} \right]_{(m \setminus n),(m \setminus n)}. \tag{33}$$

Analogously, this block on the right-hand side of Eq. (32) can be expressed as:

$$\left[ \boldsymbol{T}_{\frac{\partial \hat{\mathbf{z}}}{\partial \mathbf{z}}} \boldsymbol{G}_{\frac{\partial \mathbf{z}}{\partial \mathbf{z}}} \right]_{(m \cap n),(m \setminus n)} = \left[ \boldsymbol{T}_{\frac{\partial \hat{\mathbf{z}}}{\partial \mathbf{z}}} \right]_{(m \cap n),:} \left[ \boldsymbol{G}_{\frac{\partial \mathbf{z}}{\partial \mathbf{z}}} \right]_{:,(m \setminus n)} = \left[ \boldsymbol{T}_{\frac{\partial \hat{\mathbf{z}}}{\partial \mathbf{z}}} \right]_{(m \cap n),(m \cap n)} \left[ \boldsymbol{G}_{\frac{\partial \mathbf{z}}{\partial \mathbf{z}}} \right]_{(m \cap n),(m \setminus n)}. \tag{34}$$

Thus, we have the equality for the block $[(m \cap n), (m \setminus n)]$:

$$\left[\boldsymbol{G}_{\frac{\partial \hat{\mathbf{z}}}{\partial \hat{\mathbf{z}}}}\right]_{(m \cap n),(m \setminus n)} \left[\boldsymbol{T}_{\frac{\partial \hat{\mathbf{z}}}{\partial \mathbf{z}}}\right]_{(m \setminus n),(m \setminus n)} = \left[\boldsymbol{T}_{\frac{\partial \hat{\mathbf{z}}}{\partial \mathbf{z}}}\right]_{(m \cap n),(m \cap n)} \left[\boldsymbol{G}_{\frac{\partial \mathbf{z}}{\partial \mathbf{z}}}\right]_{(m \cap n),(m \setminus n)}$$

$$\implies$$

$$\left[\boldsymbol{G}_{\frac{\partial \hat{\mathbf{z}}}{\partial \hat{\mathbf{z}}}}\right]_{(m \cap n),(m \setminus n)} = \left[\boldsymbol{T}_{\frac{\partial \hat{\mathbf{z}}}{\partial \mathbf{z}}}\right]_{(m \cap n),(m \cap n)} \left[\boldsymbol{G}_{\frac{\partial \mathbf{z}}{\partial \mathbf{z}}}\right]_{(m \cap n),(m \setminus n)} \left[\boldsymbol{T}_{\frac{\partial \hat{\mathbf{z}}}{\partial \mathbf{z}}}\right]_{(m \setminus n),(m \setminus n)}. \quad (35)$$

This graphical relation is identical to that in Eq. (25).

However, the relation for the block $[(n \setminus m), (m \setminus n)]$ between two modality-specific parts varies, due to the potential mixing of the shared part into these blocks, which may increase the inbound edges (not outbound edges), as we show below.

For the block $[(n \setminus m), (m \setminus n)]$ on the left-hand side of Eq. (32) gives:

$$\left[\boldsymbol{G}_{\frac{\partial \hat{\mathbf{z}}}{\partial \hat{\mathbf{z}}}} \boldsymbol{T}_{\frac{\partial \hat{\mathbf{z}}}{\partial \mathbf{z}}}\right]_{(n \setminus m),(m \setminus n)} = \left[\boldsymbol{G}_{\frac{\partial \hat{\mathbf{z}}}{\partial \hat{\mathbf{z}}}}\right]_{(n \setminus m),:} \left[\boldsymbol{T}_{\frac{\partial \hat{\mathbf{z}}}{\partial \mathbf{z}}}\right]_{:,(m \setminus n)} = \left[\boldsymbol{G}_{\frac{\partial \hat{\mathbf{z}}}{\partial \hat{\mathbf{z}}}}\right]_{(n \setminus m),(m \setminus n)} \left[\boldsymbol{T}_{\frac{\partial \hat{\mathbf{z}}}{\partial \mathbf{z}}}\right]_{(m \setminus n),(m \setminus n)}.$$
$$(36)$$

Unlike previous cases, the right-hand side of Eq. (32) for the block involves more than atomic blocks (i.e., it involves the entire modality $(n)$):

$$\left[\boldsymbol{T}_{\frac{\partial \hat{\mathbf{z}}}{\partial \mathbf{z}}} \boldsymbol{G}_{\frac{\partial \mathbf{z}}{\partial \mathbf{z}}}\right]_{(n \setminus m),(m \setminus n)} = \left[\boldsymbol{T}_{\frac{\partial \hat{\mathbf{z}}}{\partial \mathbf{z}}}\right]_{(n \setminus m),:} \left[\boldsymbol{G}_{\frac{\partial \mathbf{z}}{\partial \mathbf{z}}}\right]_{:,(m \setminus n)} = \left[\boldsymbol{T}_{\frac{\partial \hat{\mathbf{z}}}{\partial \mathbf{z}}}\right]_{(n \setminus m),(n)} \left[\boldsymbol{G}_{\frac{\partial \mathbf{z}}{\partial \mathbf{z}}}\right]_{(n),(m \setminus n)}. \quad (37)$$

Then, it follows from Eq. (36) and Eq. (37) that

$$\left[\boldsymbol{G}_{\frac{\partial \hat{\mathbf{z}}}{\partial \hat{\mathbf{z}}}}\right]_{(n \setminus m),(m \setminus n)} \left[\boldsymbol{T}_{\frac{\partial \hat{\mathbf{z}}}{\partial \mathbf{z}}}\right]_{(m \setminus n),(m \setminus n)} = \left[\boldsymbol{T}_{\frac{\partial \hat{\mathbf{z}}}{\partial \mathbf{z}}}\right]_{(n \setminus m),(n)} \left[\boldsymbol{G}_{\frac{\partial \mathbf{z}}{\partial \mathbf{z}}}\right]_{(n),(m \setminus n)}$$

$$\implies$$

$$\left[\boldsymbol{G}_{\frac{\partial \hat{\mathbf{z}}}{\partial \hat{\mathbf{z}}}}\right]_{(n \setminus m),(m \setminus n)} = \left[\boldsymbol{T}_{\frac{\partial \hat{\mathbf{z}}}{\partial \mathbf{z}}}\right]_{(n \setminus m),(n)} \left[\boldsymbol{G}_{\frac{\partial \mathbf{z}}{\partial \mathbf{z}}}\right]_{(n),(m \setminus n)} \left[\boldsymbol{T}_{\frac{\partial \mathbf{z}}{\partial \hat{\mathbf{z}}}}\right]_{(m \setminus n),(m \setminus n)}. \quad (38)$$

We can observe that the existence of the shared variables $\mathbf{z}^{(m \cap n)}$ divides the latent space into finer blocks $\mathbf{z}^{(m \setminus n)}$, $\mathbf{z}^{(n \setminus m)}$, and $\mathbf{z}^{(m \cap n)}$. Eq. (35) and Eq. (38) reveal that the bijective indeterminacy relation hold over these finer blocks, exception for the non-square transition matrix $\left[\boldsymbol{T}_{\frac{\partial \hat{\mathbf{z}}}{\partial \mathbf{z}}}\right]_{(n \setminus m),(n)}$ on the right-hand side of Eq. (38). This is because that the shared part $\mathbf{z}^{(m \cap n)}$ can potentially mix in $\hat{\mathbf{z}}^{(n \setminus m)}$, so $\hat{\mathbf{z}}^{(n \setminus m)}$ may receive edges inbound to $\mathbf{z}^{(m \cap n)}$.

**Interplay among multiple modalities.** In light of the graphical condition for the two-modality case (Eq. (35) and Eq. (38)), we can derive the conditions for the multi-modality case.

Specifically, we classify the blocks in the estimation graph $\hat{\boldsymbol{G}}_{\frac{\partial \hat{\mathbf{z}}}{\partial \hat{\mathbf{z}}}}$ into the following categories for two distinct atomic blocks $b_1$ and $b_2$.

Region 1 : Blocks $b_1$ and $b_2$ do not have nested memberships, i.e., $\mathcal{M}(\mathbf{z}^{(b_1)}) \not\subset \mathcal{M}(\mathbf{z}^{(b_2)})$ and $\mathcal{M}(\mathbf{z}^{(b_2)}) \not\subset \mathcal{M}(\mathbf{z}^{(b_1)})$;

Region 2 : Block $b_1$ has fewer memberships than block $b_2$: $\mathbf{z}^{(b_1)} \prec \mathbf{z}^{(b_2)}$;

Region 3 : Block $b_1$ has more memberships than block $b_2$: $\mathbf{z}^{(b_2)} \prec \mathbf{z}^{(b_1)}$.

Eq. (35) and Eq. (38) reveal that the sparsity for Region 1 and Region 2 is informative, whereas Region 3 is not. This is because in these the inherent indeterminacy from the subspace identifiability within each modality (Theorem 4.2) will engage the product $\boldsymbol{T}_{\frac{\partial \hat{\mathbf{z}}}{\partial \hat{\epsilon}}} \boldsymbol{T}_{\frac{\partial \hat{\epsilon}}{\partial \mathbf{z}}}$ in Eq. (31) in addition to the sparsity in the estimated graph $\boldsymbol{G}_{\frac{\partial \hat{\mathbf{z}}}{\partial \hat{\mathbf{z}}}}$.

**Overall conditions.** Consolidating all the considerations above, we re-define objects in Condition 4.3 as follows.

1. The indeterminacy matrix $\boldsymbol{T} := \boldsymbol{T}_{\text{on}} + \boldsymbol{T}_{\text{off}}$ is not strictly block-diagonal: The matrix $\boldsymbol{T}_{\text{on}}$ contains all the on-diagonal square invertible matrices $\boldsymbol{T}_{\text{on}} := \text{diag}(\boldsymbol{T}_{b_1}, \ldots, \boldsymbol{T}_{b_{|B|}})$ and $\boldsymbol{T}_{\text{off}}$ contains all the off-diagonal elements potentially nonzero in the regions $(b, H(b) \setminus b)$ for $b \in B$. Also, the matrix multiplication becomes $\left[\boldsymbol{T}_{\frac{\partial \hat{\mathbf{z}}}{\partial \mathbf{z}}}\right]_{(\tilde{b}), H(\tilde{b})} \left[\boldsymbol{G}_{\frac{\partial \mathbf{z}}{\partial \mathbf{z}}}\right]_{H(\tilde{b}),(b)} \left[\boldsymbol{T}_{\frac{\partial \mathbf{z}}{\partial \hat{\mathbf{z}}}}\right]_{(b),(b)}$ as a unified expression of Eq. (35) and Eq. (38).

2. The sub-matrices on which we impose the sparsity controls are exactly the union of Region 2 and Region 1, i.e., the complement of Region 3. We denote such a region as the function of the block index $(E(b), b)$ for each $b \in B$.

With these modifications, the rest of the proof follows exactly from that of Theorem 4.4.

$\square$

# D EXPERIMENTAL DETAILS

## D.1 NUMERICAL DATASET

We use six numerical datasets in this paper, including three multimodal datasets that satisfy our assumptions and three that slightly violate the sparsity assumptions in the proposed theorems.

**Multi-modality settings** We generate $n = 10000$ samples according to Eq. (1) and Eq. (2). Following prior work (Von Kügelgen et al., 2021; Yao et al., 2021; Zimmermann et al., 2021), we generate observations using a multi-layer perceptron (MLP). Specifically, the mixing function $g$ is modeled as a three-layer MLP with randomly initialized weights and leaky ReLU activations, enabling $g$ to represent a general nonparametric mixing function. The causal noise terms $\epsilon$ are independently and identically distributed (i.i.d.), and the exogenous variables are mutually independent. Sparse inter-modality causal dependencies are randomly generated, ensuring that each modality's latent variables maintain at least one causal connection with another modality.

**Ablation settings** For the ablation study, we generate two modality observations under different sparsity ratios. Each observation is generated from three causally related latent variables and one exogenous variable. The sample size for each dataset is set to $n = 10000$, and the dimensionality of the observations in each modality is $d(\mathbf{x}) = 20$. The sparsity ratio determines the extent of inter-modality connections among these latent variables. A higher sparsity ratio leads to a sparser causal structure, meaning fewer causal connections between latent variables. Conversely, a lower sparsity ratio yields a denser causal matrix with more causal dependencies. For example, a sparsity ratio of 0% indicates that all inter-modality latent variables are fully connected, whereas a sparsity ratio of 50% implies that half of the possible causal edges are removed.

## D.2 SYNTHETIC DATASET

**Variant MNIST** In real-world scenarios, the ground-truth latent processes are often unknown, making it challenging to evaluate model performance. To address this, we construct a synthetic dataset based on the real image dataset MNIST (LeCun, 1998) with known causal relationships, which supports the setting considered in our work. Our synthetic dataset consists of two modalities, each with latent variables that exhibit causal relationships. The design is flexible. The modalities could correspond to different MNIST variants, such as colored MNIST (Arjovsky et al., 2019) or fashion MNIST (Xiao et al., 2017). The causally related latent variables could be, for example, digit identity, image color, clothing category, image rotation, etc.

In order to make the synthetic setting more intuitive, we introduce an alternative setting: object position acts as a latent variable that influences the appearance of MNIST images. Across different modalities, such causal influence may vary. Furthermore, position in modality 1 may causally influence position in modality 2, which aligns with the data generation process in our work. For

example, the horizontal position of a digit — such as the six — directly influences the transparency of the MNIST image. This horizontal position then serves as a causal factor for the vertical position of shoes in the fashion MNIST, which in turn affects the grayscale intensity of the shoe image. To systematically evaluate the performance of our algorithm under different observational conditions, we consider three variations in colored MNIST, where the digits are assigned one of three colors: red, green, or blue. These relationships are visually illustrated in Figure 7 (a) for clarity.

## D.3  REAL-WORLD DATASET

In this paper, we consider three types of datasets, including image, time series, and tabular data. Visualizations of the image and time-series datasets are shown in Figure 7 (b-c).

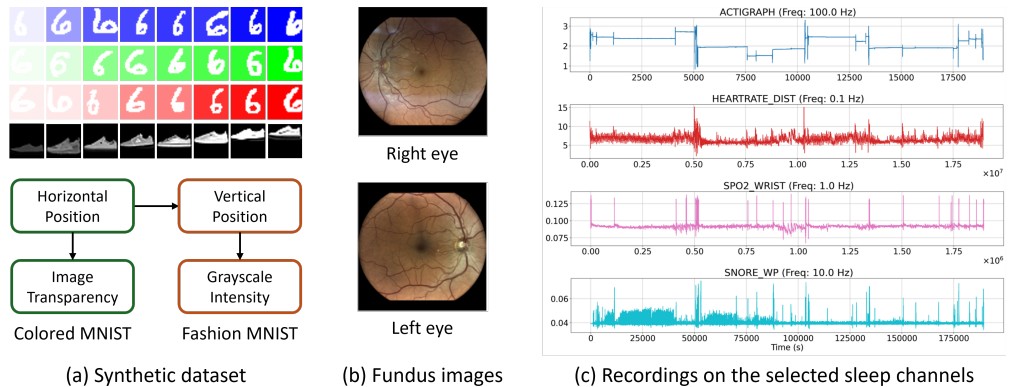

Figure 7: Visualization on the datasets: (a) Synthetic dataset: Variant MNIST. (b) Real-world dataset: Fundus imaging shows the interior surface of the eyes. (c) Real-world dataset: Sleep monitoring shows the time-series recording of sleep-related metrics overnight.

**Fundus imaging** is the visualization of the interior surface of the fundus, which includes structures such as the optic disc, retina, and retinal microvasculature. High-resolution images of the back of the eye are essential for the diagnosis and monitoring of a variety of eye diseases and conditions.

For example, the retinal microvasculature, which consists of small blood vessels that supply blood to the retina, provides valuable information about eye health. Moreover, fundus imaging can improve understanding of the underlying mechanisms of various eye diseases. It serves as a non-invasive tool to assess the overall health of the microvascular circulation health and provides a direct view of part of the central nervous system.

**Sleep monitoring** is a time-series dataset collected over three consecutive nights that records various metrics including sleep stage, body position, respiratory events, heart rate, oxygen saturation, and snoring. This dataset focuses on obstructive sleep apnea (OSA), a sleep disorder in which a person's breathing is interrupted during sleep due to the relaxation of throat muscles, causing upper airway obstruction. These interruptions often lead to loud snoring, reduced blood oxygen levels, stress responses, awakenings, and fragmented sleep.

This dataset is collected from a Home Sleep Apnea Test (HSAT), a non-invasive diagnostic method for sleep apnea. Patients wear a portable device overnight to monitor their breathing patterns, heart rate, oxygen levels, snoring, and other sleep patterns. The dataset includes multiple channels, such as ACTIGRAPH for movement, HEARTRATE_DIST for heart rate, SPO2_WRIST for blood oxygen saturation, and SBORE_WP for snoring, capturing key aspects of physical activity and sleep patterns during the HSAT. The device calculates apnea-related indices, including the Apnea/Hypopnea Index (AHI), Respiratory Disturbance Index (RDI), and Oxygen Desaturation Index (ODI), as well as indices for diagnosing conditions such as atrial fibrillation.

## D.4  EVALUATION METRICS

**MCC: Mean Correlation Coefficient**    MCC is a standard metric used to evaluate the recovery of latent factors in causal representation learning. It measures the alignment between the ground-truth

factors and the estimated latent variables. Specifically, MCC first computes the absolute values of the correlation coefficients between each ground-truth factor and each estimated latent variable. To account for possible permutations of the latent variables, the metric solves a linear sum assignment problem on the computed correlation matrix in polynomial time, ensuring optimal matching between the factors and their corresponding latent representations.

**R2: Coefficient of Determination**   R2 is a standard metric used to evaluate the goodness of fit in regression models. It measures the proportion of variance in the dependent variable that is explained by the independent variables in the model. Specifically, R2 compares the residual sum of squares of the model with the total sum of squares and returns a value between 0 and 1. A higher R2 indicates that the model explains a larger portion of the variance in the data, with 1 representing a perfect fit and 0 indicating that the model explains none of the variability.

**SHD: Structural Hamming Distance**   SHD is a widely used metric for evaluating the accuracy of graph structure recovery in causal discovery. It quantifies the difference between the true causal graph and the estimated graph. Specifically, SHD counts the number of edge modifications—additions, deletions, or reversals—required to transform the estimated graph into the ground-truth graph. This metric provides a simple yet effective measure of structural similarity, with a lower SHD indicating a closer alignment between the estimated and true causal structures.

### D.5   DETAILED DISCUSSION ON HUMAN PHENOTYPE

Without learning such latent variables, we cannot provide a causal explanation between different modalities. The estimated model shows all causal influences involved, suggests the existence of hidden causal variables, and illustrates their relationships with each other and with observable data. Asymptotically, the learned adjacency matrix $A$ corresponds to a graph within the Markov equivalence class given by the PC algorithm.

To interpret the learned hidden variables, we primarily refer to the existing medical literature, which supports their alignment with background knowledge, thereby adding validity to our results. For example, the latent variable FRight3 relates handgrip strength to fundus imaging, consistent with findings showing that handgrip strength correlates with intraocular pressure (IOP) (Pérez-Castilla et al., 2021). In addition, the association between the cataract and changes in IOP (Slabaugh et al., 2013) is consistent with the findings of the model. These connections underline the physiological relevance of the learned hidden variable. Similarly, FRight1 and FLeft1, associated with fundus imaging and age estimation, are consistent with studies demonstrating age-related changes in fundus image color content (Ege et al., 2002). Another latent variable Sleep1 associated with oxygen saturation and sleep metrics aligns with findings that oxygen saturation is a strong predictor of obstructive sleep apnea (OSA) severity (Wali et al., 2020). This indicates that the model's latent variable effectively captures critical factors related to sleep disorders.

## E   EXTENDED EXPERIMENT

To further assess the robustness, scalability, and applicability of our proposed method, we conducted a series of extended experiments under more complex scenarios. These experiments aim to evaluate the performance under diverse latent variable configurations, varying sample sizes, and different structural assumptions, including non-DAG settings and shared latent variables.

**Performance in complex scenarios.**   To evaluate the scalability and generalizability of our method to complex causal structures, we conducted additional experiments on higher-dimensional simulated tasks with diverse configurations of latent variables and modalities. These setups introduce significantly more complex causal relationships between variables. Specifically, we consider three extended scenarios: (1) *Five-mods*, with 30-dimensional observations from five modalities with two latent variables and one exogenous variable per modality. (2) *Six-mods*, with 30-dimensional observations from six modalities with two latent variables and one exogenous variable per modality. (3) *Eight-mods*, involving 30-dimensional observations from eight modalities with two latent variables and one exogenous variable per modality. The results, summarized in Table 4, show that our method consistently delivers robust performance under these challenging conditions.

**Impact of the number of latent variables.** In real-world applications, the true number of latent variables is typically unknown, and arbitrarily predefining this number may introduce bias and degrade model performance. In this section, we discuss how our method can eliminate the redundant effect of the latent variables, and introduce a cross-validation-based method to determine the appropriate number of latent nodes. By manually setting a range for the number of latent variables and selecting the one with the lowest validation loss, we ensure a principled approach that is both simple and widely applicable (Khemakhem et al., 2020b). Here we conduct synthetic experiments to validate its effectiveness. We followed the data generation process in Section D.1, where the ground-truth number of latent variables is two per modality. The results, as shown in Figure 8(a), demonstrate that our approach accurately recovers the correct number of latent variables.

**Impact of sample size.** To investigate the impact of sample size on model performance, we conducted an additional experiment evaluating the MCC as the number of data samples increased. Following the data generation process described in Section D.1, where the ground-truth number of latent variables is two for two modalities. We systematically increased the sample size from 10,000 to 40,000 and measured MCC and R2 accordingly. The results, presented in Figure 8(b), show a consistent improvement in MCC as the sample size increases. This finding confirms the hypothesis that greater data availability enhances the model's ability to recover the underlying causal structure.

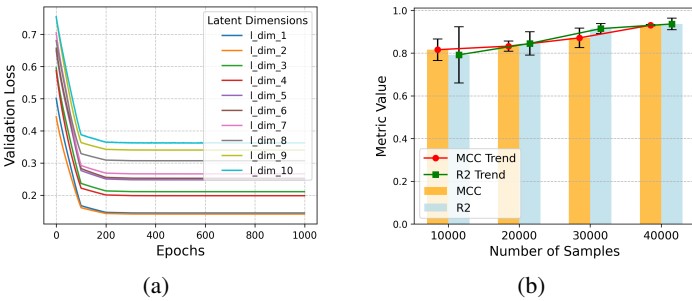

(a)                                         (b)

Figure 8: (a) Comparison of loss across different latent dimensions. (b) The effect of sample size.

**Evaluation under non-DAG assumptions.** The theoretical results in this paper do not strictly require the assumption of Directed Acyclic Graphs (DAGs) for latent variable structures within or across modalities. To evaluate our method under non-DAG settings, we conducted synthetic experiments where cycles were introduced within and across modalities. Specifically, we followed the data generation process in Section D.1, and considered: (1) cyclic influence within modality; and (2) cyclic influence across modalities. Empirical results in Table 4 demonstrate that the presence of cycles does not hinder the identification of latent variables.

**Discussion on the shared latent variables.** We present how to preprocess the current framework to accommodate shared variables across modalities and provide empirical results. The extended framework incorporates an additional mechanism to estimate the shared latent variable. Inspired by previous works (Yao et al., 2023; Daunhawer et al., 2023; Von Kügelgen et al., 2021), we incorporate an additional contrastive loss to enforce similarity in the shared latent representations. To evaluate the effectiveness of this extension, we modify the data generation process in Section D.1 and allow for the existence of a shared variable across modalities. The results, summarized in Table 4, show that our method accurately recovers both shared and modality-specific latent variables across different scenarios, confirming the theoretical guarantees of the extended framework.

## F  IMPLEMENTATION DETAILS

In this section, we provide details of the network architecture, including the optimization scheme and hyperparameter setting.

| Metric | Complex Scenarios | | | Non-DAG Settings | | Shared Latent Variables | |
|---|---|---|---|---|---|---|---|
| | Five mods | Six mods | Eight mods | Cyclic within | Cyclic across | Two mods | Three mods |
| R2 | $0.89_{\pm\ 1e\text{-}4}$ | $0.97_{\pm\ 8e\text{-}7}$ | $0.83_{\pm\ 1e\text{-}3}$ | $0.95_{\pm\ 1e\text{-}5}$ | $0.94_{\pm\ 2e\text{-}4}$ | $0.86_{\pm\ 5e\text{-}4}$ | $0.90_{\pm\ 1e\text{-}4}$ |
| MCC | $0.84_{\pm\ 3e\text{-}4}$ | $0.82_{\pm\ 4e\text{-}4}$ | $0.91_{\pm\ 5e\text{-}4}$ | $0.89_{\pm\ 2e\text{-}4}$ | $0.92_{\pm\ 1e\text{-}5}$ | $0.83_{\pm\pm\ 7e\text{-}4}$ | $0.83_{\pm\ 4e\text{-}6}$ |

Table 4: Extended experiment results across different experimental settings.

## F.1 NETWORK ARCHITECTURE

We summarize our network architecture below and describe it in detail in Table 5.

- **(1,2) Encoder and Decoder**: The encoder transforms raw observations into latent representations, while the decoder reconstructs the inputs from the latent variables. The encoder-decoder design varies depending on the downstream task. For synthetic data, MLPs with leaky ReLU activation were used. For image data, CNN was used as the encoder, and ConvTranspose2D as the decoder. LSTMs were used for time series data. Based on the universal approximation theorem, the model is theoretically able to approximating the underlying mixing function.
- **(3) Learnable Adjacency Matrix**: The causal relationships are embedded in the learned adjacency matrix, where the binary elements indicate whether specific pairs of vertices contribute to the generation of components. It initializes a learnable matrix that captures these dependencies. During the forward pass, the matrix is processed to ensure a directional structure where only certain connections are allowed based on a threshold. This allows the model to learn sparse, meaningful relationships between the latent variables.
- **(4) Flow-based Transformation**: The flow-based transformation is implemented using an MLP to process the latent variable and a flow model for the transformation. The MLP first extracts features from the latent variable, which are then used as input to the flow model, which applies an invertible transformation to the latent space, allowing the model to estimate the noise distribution.

## F.2 TRAINING DETAILS

**Optimization Scheme.** The estimation framework was trained using the Adam optimizer on GPU, and the StepLR scheduler was used to reduce the learning rate periodically. The training process ran for a maximum of 10000 epochs, with early stopping applied if the validation loss does not improve for 20 consecutive epochs. Random seeds were used to ensure reproducibility, and results were averaged across experiments, with variance reported.

The training loss combines multiple components.

- Reconstruction loss: Mean squared error between reconstructed inputs and original data.
- KL divergence loss: Encourages estimated variables to follow a standard normal prior.
- Sparsity loss: An L1-norm penalty is applied to the adjacency matrix to enforce sparsity.

**Hyperparameter.** The hyperparameters $\alpha = [\alpha_{\text{Ind}}, \alpha_{\text{Sp}}, \alpha_{\text{Recon}}]$ represent the weights assigned to each term in the composite objective function. For each dataset, they were tuned within appropriate logarithmic intervals, ensuring a balance between independence, sparsity, and reconstruction. For the experiments, the following settings were applied: $\alpha = [1e-1, 1e-2, 1]$ for the synthetic dataset, $\alpha = [1e-2, 1e-3, 2]$ for the MNIST dataset, and $\alpha = [1e-1, 1e-2, 1]$ for the phenotype dataset.

## G ALGORITHM PSEUDOCODE

The pseudocode for the proposed algorithm is presented in Algorithm 1.

| Configuration | Description | Output |
|---|---|---|
| **1.1 MLP-Encoder** | Encoder for numerical data | |
| Input | Multi-modality observations | BS × d_x |
| Dense | h_dim neurons, LeakyReLU | BS × h_dim |
| Dense | h_dim neurons, LeakyReLU | BS × h_dim |
| Dense | Latent embeddings | BS × l_dim |
| **2.1 MLP-Decoder** | Decoder for numerical data | |
| Input | Latent embeddings | BS × l_dim |
| Dense | h_dim neurons, LeakyReLU | BS × h_dim |
| Dense | h_dim neurons, LeakyReLU | BS × h_dim |
| Dense | Reconstructed observations | BS × d_x |
| **1.2 Image-Encoder** | Encoder for image data | |
| Input | Image input | BS × 3 × H × W |
| ResNet18 | ResNet backbone, LeakyReLU | BS × h_dim |
| Dense | Latent embeddings | BS × l_dim |
| **2.2 Image-Decoder** | Decoder for image data | |
| Input | Latent embeddings | BS × l_dim |
| Dense | h_dim neurons | BS × h_dim × H' × W' |
| ConvTranspose2D | Reconstructed observations | BS × 3 × H × W |
| **1.3 Time-series Encoder** | Encoder for time-series data | |
| Input | Multi-channel time-series data | BS × seq_len × n_channel |
| LSTM | Sequences into hidden representations | BS × h_dim |
| Output | Latent representation | BS × l_dim |
| **2.3 Time-series Decoder** | Decoder for time-series data | |
| Input | Latent representation | BS × l_dim |
| LSTM | Sequence into output features | BS × seq_len × h_dim |
| Output | Reconstructed time-series data | BS × seq_len × n_channel |
| **3. Adjacency Matrix** | Sparsity regularization | |
| Input | Latent variables from encoders | BS × z_all |
| Masking | Lower triangular mask | z_all × z_all |
| Thresholding | Retain entries exceeding threshold | z_all × z_all |
| Output | Learned causal adjacency matrix | z_all × z_all |
| **4. Flow Transformation** | Estimate the noise term | |
| Input | Latent variables across modalities | BS × z_all |
| Condition Input | Apply adjacency matrix to latent | BS × z_all × z_all |
| Flow Transformation | Apply transformation to latent | BS × z_all |
| Output | Estimated noise variables | BS × z_all |

Table 5: Architecture details. BS: batch size, d_x: input dimension, l_dim: latent dimension in each modality, z_all: latent dimensions across all modalities, h_dim: hidden dimension, H/W: height/width of the input image, seq_len: sequence length, n_channel: number of channels.

---

**Algorithm 1** Pseudocode for the proposed algorithm.

---

1: **Input:** Grouped observations $\{\mathbf{x}^{(m)}\}_{m=1}^{M}$
2: **Output:** Estimated latent variables $\{\hat{\mathbf{z}}^{(m)}\}_{m=1}^{M}$
3:
4: *# Random Initialization*
5: Initialize encoders $\{\text{En}^{(m)}\}_{m=1}^{M}$ and decoders $\{\text{De}^{(m)}\}_{m=1}^{M}$ for each group
6:
7: *# Encoder*
8: **Input:** Grouped observations $\{\mathbf{x}^{(m)}\}_{m=1}^{M}$
9: **Output:** Estimated latent variables $\hat{\mathbf{z}}^{(m)}$ for each group $m$
10: **for each group** $m = 1$ **to** $M$ **do**
11:     Encode the current group latent and exogenous variables: $\hat{\mathbf{z}}^{(m)}, \hat{\eta}^{(m)} = \text{En}^{(m)}(\mathbf{x}^{(m)})$
12: **end for**
13: Concatenate latent representations: $\{\hat{\mathbf{z}}^{(m)}\}_{m=1}^{M} = \hat{\mathbf{z}}^{(1)} \oplus \hat{\mathbf{z}}^{(2)} \oplus \ldots \oplus \hat{\mathbf{z}}^{(M)}$
14: **return** Estimated latent variables and exogenous variables $\{\hat{\mathbf{z}}^{(m)}, \hat{\eta}^{(m)}\}_{m=1}^{M}$
15:
16: *# Flow-based Noise Estimation*
17: **Input:** Estimated latent variables for each group $\{\hat{\mathbf{z}}^{(m)}\}_{m=1}^{M}$
18: **Output:** Estimated noise term $\hat{\epsilon}_{i=1}^{d(\mathbf{z})}$
19: Initialize adjacency matrix $\hat{\mathbf{A}}$
20: Select the parents of latent variable based on the adjacency matrix
21: Pass through flow model to obtain estimated residuals $\hat{\epsilon}_i$
22: Update the estimated causal graph based on the adjacency matrix with threshold
23: Compute sparsity loss based on $L_1$ norm
24: Compute the KL divergence between $[\{\hat{\eta}^{(m)}\}_{m=1}^{M}, \hat{\epsilon}_{i=1}^{d(\mathbf{z})}]$ and Gaussian prior
25: **return** Estimated noise term $\hat{\epsilon}_{i=1}^{d(\mathbf{z})}$
26:
27: *# Decoder*
28: **Input:** Estimated latent and exogenous variables in each group $\{\hat{\mathbf{z}}^{(m)}, \hat{\eta}^{(m)}\}_{m=1}^{M}$
29: **Output:** Reconstructed grouped features $\{\hat{\mathbf{x}}^{(m)}\}_{m=1}^{M}$
30: **for each group** $m = 1$ **to** $M$ **do**
31:     Decode $(\hat{\mathbf{z}}^{(m)}, \hat{\eta}^{(m)})$ to reconstruct features $\hat{\mathbf{x}}^{(m)}$: $\hat{\mathbf{x}}^{(m)} = \text{De}^{(m)}(\hat{\mathbf{z}}^{(m)}, \hat{\eta}^{(m)})$
32:     Compute reconstruction loss using MSE: $\mathcal{L}_{\text{Recon}}^{(m)} = \text{MSE}(\hat{\mathbf{x}}^{(m)}, \mathbf{x}^{(m)})$
33: **end for**

---

