# OpenReview forum: "Causal Representation Learning from Multimodal Biomedical Observations"
_ICLR.cc/2025/Conference — ICLR 2025 Poster_

### Official Review · Reviewer_91p1 · 2024-10-28

**Soundness:** 2
**Presentation:** 2
**Contribution:** 2
**Rating:** 3
**Confidence:** 4

**Summary:**

The paper addresses the identification of hidden causes underlying observed multimodal data. The authors propose a model that avoids previous limitations, specifically not requiring shared latent information across multiple modalities or assuming an exponential family distribution for hidden causes. The paper provides a theoretical identifiability analysis and presents experiments, including one on biological data.

**Strengths:**

The paper presents meaningful relaxations of assumptions from prior work.

It is relatively well-written, considering the complexity of the theory and notation involved.

**Weaknesses:**

There are issues with the main theoretical claims and their proofs.

The practicality of the current assumptions and their real-world applicability remains unclear.

The experiment using the human phenotype data does not justify the title or the highlights in the abstract.

**Questions:**

### Theory

1. *Identification vs. Estimation*: In Theorems 4.2 and 4.4, identification and estimation are confused by statements like “We estimate the generating process…”. Identification is a model property and should be established before estimation is addressed. For a model $\mathcal{P}:=\\{p_{\phi}:\phi \in \Phi\\}$, we simply show that $\forall \phi_1, \phi_2 \in \Phi (p_{\phi_1}=p_{\phi_2} \implies \phi_1=\phi_2)$ to demonstrate identification.

2. *"Identification under Regularization"*: Particularly, Theorem 4.4 is problematic in stating that identification is achieved through regularization, which pertains to estimation. If regularization is essential, the model should be reformulated—like for ridge regression, the regularization can be treated as a sparse prior in a probabilistic model.

3. *Proof of Theorem 4.2*: The step to equation (14) seems to rely on $x=\hat{x}$. However, equal distributions do not imply equal values of random variables. If (14) is correct, the derivation should clarify how equal distribution leads to (14).

4. *Definition of $\tilde{g}_x$*: Strictly speaking, there is one $\tilde{g}_x$ for each $m$, and it’s unclear if these $m$ functions can be shown as the same function. If not, we are in fact making assumptions about $m$ functions.

5. *Proof of Theorem 4.4*: 1) Condition 4.3 mentions *undefined matrix $A$*. In the proof, equation (28) involves more than a single matrix,  as denoted by $A$ in Condition 4.3, and the definition of $d^*(U)$ does not seem to align with $d^*$ in Condition 4.3 (this questionable reasoning later is extended to $D$). This proof section is unclear both in notation and logic flow; we can not even see $A$ as a placeholder for the various matrices used in equation (28)! I suggest sorting out the involved matrixes (I believe, if the proof is not simply wrong, there should be several similar conditions for different matrixes), and a clear connection between the proof and assumptions should be seen. 2) *the regularization again*: the reasoning following equation (29) requires satisfaction of (29), but regularization (5) can only encourage, not guarantee this condition. This suggests the idea of “identification under regularization” itself does not work, not only an incorrect formulation as mentioned in the 2nd point.

### Implications
One core aspect of the causal model is that two modalities cannot share any direct causes, which is questionable. While $z^{(m)}$ can, and indeed is required to, affect $x^{(-m)}$ indirectly through $z^{(-m)}$, this seems to require hidden causes $z$ to be lower-level features. For example, “age” is a high-level feature affecting many modalities, like “sleep” and “eyes.” Thus, if the "sleep" modality includes "age" in $z^{sleep}$, then, to satisfy the "indirect" causes requirement, lower-level features $z^{eyes}$ would be needed to mediate the causal effects from “age” to “eyes", i.e., $d(z^{eyes})$ should be large enough to block all the causal effects from “age” to “eyes.” And, when we have many modalities and many hidden causes, this eventually requires most hidden causes to be low-level.

While the theoretical relaxations and real-world examples are appreciated, it’s unclear why, in the “sleep patterns” and “brain imaging” examples, your assumptions could be satisfied, though I can see previous ones are limited.


### Biological Application
The demonstration of the biological application lacks depth, and the claimed “strong interpretability” is overstated. For example, in the human phenotype dataset, how should we interpret the learned hidden variables? Does the method enhance our understanding of the data or its underlying mechanisms? How does the number of latent variables affect the results (possibly with very different causal graphs)?

Additional specific questions are: Why are causal relationships involving FRight and FLeft not symmetric? For example, why does FRight affect Cataract but not FLeft?
Why are there no Sleep variables that influence “Sleep efficiency”?
Shouldn’t ocular quality impact hand grip strength rather than the reverse?

More experiments should be conducted, both on the model side (e.g., testing various numbers of latent variables and other hyperparameters) and the data side (e.g., trying many different modalities).


Overall, the authors could clean up theoretical developments, clarify the practical implications, and leave the biological application to future work.

---

> ### Author Response · Authors · 2024-11-22
> **Rebuttal by Authors [Part 1]**
>
> Thank you so much for the constructive and insightful feedback. We really appreciate your valuable time and efforts dedicated to reviewing our work.
>
> **W1: There are issues with the main theoretical claims and their proofs.**
>
> We appreciate your valuable comments on the technical language. We hoped that our contribution could be appreciated by the community, so we followed recent developments in the field to formulate the identifiability results [1-5]. However, in light of your suggestion, we have refined the presentation of the identifiability result as indicated in our individual responses, which aligns with the classical identifiability definition.
>
> We hope that this can address your concerns. Any further feedback on these matters will be highly appreciated – thank you!
>
> **W2: The practicality of the current assumptions and their real-world applicability remains unclear.**
>
> Thank you for the comments. We address your specific concern on the practicality of our assumptions (i.e., shared variables) in the question section below both theoretically and empirically.
>
> We totally agree that there would exist real-world cases that our assumptions may not cover. Nevertheless, please let us add that since science is established step by step, we believe that this contribution will shed light on the advancement of this field, despite its limitations.
>
> **W3: The experiment using the human phenotype data does not justify the title or the highlights in the abstract.**
>
> Thank you for your thoughtful comment. Our goal is to facilitate scientific discovery by automatically discovering hidden variables and uncovering causal relations. Without learning such latent variables, it would be challenging to provide a causal explanation across different modalities. Our model can capture the causal influences involved, suggest the existence of hidden causal variables, and illustrate their relationships with one another as well as with observable data.
>
> To address your concern, in the revised manuscript, we have provided a detailed discussion on the phenotype result. Specifically, we interpret the interactions among the learned variables and highlight how some of these findings align with existing medical literature. Additionally, for hidden variables that may not yet be interpretable within the current domain, we believe that our results can inspire domain experts to explore these variables further, potentially leading to new insights and a deeper understanding of the problem. We hope these revisions and clarifications could address your feedback. Thank you again for your constructive suggestions.
>
> **(Theory) Q1: Identification vs. Estimation**
>
> **A1**: Thank you for the valuable feedback. Following your suggestion, we have revised these statements. For instance, in lines 191-193, we have updated the identification goal in words:
> > For two specifications $ \theta:=\\{ g \_{\mathbf{x}^{(m)}}, g\_{\mathbf{z}^{(m)}}, p( \epsilon^{(m)} ) \\} \_{m=1}^{M}$ and $ \hat{\theta} := \\{ \hat{g} \_{\mathbf{x}^{(m)}}, \hat{g} \_{\mathbf{z}^{(m)}}, \hat{p}( \epsilon^{(m)} ) \\} \_{m=1}^{M} $ of the data-generating process Eq. (1) and Eq. (2) that fit the marginal distribution $ p( \mathbf{x} ) $, we would like to show that, *given the same $ \mathbf{x} $ value*, each latent component $ \hat{z}\_{i}^{(m)} $ is equivalent to its counterpart $ z^{(m)}\_{i} $ up to an invertible map $ h\_{i}^{(m)} $, i.e., $ \hat{z}\_{i}^{(m)} = h\_{i}^{(m)} ( z^{(m)}\_{i} )$.
> In addition, we have rephrased Theorem 4.2 and 4.4 to reflect this change.
>
> We hope our current phrasing is clearer and would appreciate any feedback from you – thanks!
>
>
> **(Theory) Q2: Identification under Regularization**
>
> **A2**: Thank you for the constructive comments. In light of your suggestion, we have reformulated this regularization as a constraint of the alternative specification $ \hat{\theta} $:
> > $ \hat{\boldsymbol\theta} $ satisfies the following condition: $ \sum\_{m\neq n \in[M]} ||  [\mathbf{J} \_{ \hat{ g }\_{ \mathbf{z} } }] \_{ (m), (n) } ||\_{0} \leq \sum\_{m\neq n \in[M]} ||  [\mathbf{J} \_{ g_{ \mathbf{z} } } ] \_{ (m), (n) } || \_{0} $.
>
> Please refer to Theorem 4.4 in the revised manuscript and we hope this is clearer. Thank you!

---

> ### Author Response · Authors · 2024-11-22
> **Rebuttal by Authors [Part 2]**
>
> **(Theory) Q3: Proof of Theorem 4.2**
>
> **A3**: Thank you for the comment. There seems to be a misunderstanding and please let us clarify it. Equation (14) is part of the definition of identification, rather than a consequence of equal distribution.
> Specifically, following the classic definition of identifiability, we define two specifications $ \theta  =\\{ g _{ x^{(m)} }, g _{ z^{(m)} }, p ( \epsilon^{ (m) }  ) \\} _{ m=1 }^{M}  $ and $ \hat{\theta} := \\{ \hat{g} _{ x^{(m)} }, \hat{g} _{ z^{(m)} }, \hat{p} ( \epsilon^{ (m) }  ) \\} _{ m=1 }^{M}  $ that fit the observation distribution $ p(\mathbf{x}) $. To show the identifiability in terms of the functions in $ \theta$ and $ \hat{\theta} $, we show *given the same $\mathbf{x}^{(m)} value$*, the identifiability between $ \mathbf{z}^{(m)} $ and $ \hat{ \mathbf{z}}^{(m)} $.
> Thanks to your suggestion, we have included the above text in lines 191-194. We hope this clarifies your concerns and please kindly let us know if this is clear.
>
>
> **(Theory) Q4: Definition of $\tilde{g}_x$**
>
> **A4**: Thank you for the helpful comment. You are correct that there should be one $ \tilde{g}\_{x} $ for each modality $m$. We have revised the notation from $ \tilde{g}\_{x} $ to $ \tilde{g}^{(m)} $ to clearly indicate this. We appreciate you for your feedback!
>
> **(Theory) Q5: Proof of Theorem 4.4**
>
> **Q5.1 Condition 4.3 mentions undefined matrix.**
>
> **A5.1**: Thank you for the constructive feedback. The matrix $\mathbf{A}$ is a placeholder for sub-matrices $ [\mathbf{T} \mathbf{G} ]_{(-m), I(C^{(m)}) }$ and $ ([\mathbf{G} \mathbf{T}^{-1}] _{I(R^{(m)}), (-m)})^{\top} $. Given your comment, we have reformulated the statement in Condition 4.3 to clearly define $A$ and $d^{*}$, included the following text to explain the matrices in main text (lines 323-325), and refined the proof section (lines 1197-1262) to more explicitly illustrate its connection to this assumption.
>
> > $\mathbf{G}$ denotes the graphical connectivity in the model $\boldsymbol\theta$ and $\mathbf{T}$ denotes potential mixings of latent variables in the model $ \hat{\theta} $. Thus, sub-matrices $ [\mathbf{T} \mathbf{G}] _{(-m), I(C^{(m)}) }$ and $ ([\mathbf{G} \mathbf{T}^{-1}] _{I(R^{(m)}), (-m)})^{\top} $ represent the cross-modality connectivity between modality $m$ and other modalities $-m$ in the model $ \hat{\boldsymbol\theta} $. Condition 4.3 effectively imposes a sparsity constraint on these edges.
>
> Your comments have greatly improved our work. We really appreciate them!
>
> **Q5.2 The regularization question.**
>
> **A5.2**: Thank you for the comment. We believe this is a confusion resulting from the formulation difference – please note in the initial manuscript (29) is not a consequence of the objective (5). Rather, we are discussing the implications of $ |U| \ge 1 $ and $ |U| = 0 $ on the relative relation between the cross-modality edge densities for the two specifications $ || [ \mathbf{G}\_{ \frac{ \partial z }{ \partial z } } ] \_{ (-m), \hat{i} }  || \_{0} $ and $ || [ \mathbf{G}\_{ \frac{ \partial \hat{z} }{ \partial \hat{ z }} } ] \_{ (-m), \hat{i} }  || \_{0} $: Equation (28) indicates that this density for specification $ \hat{\theta} $ is strictly larger than that of specification $ \hat{\theta}  $ when $ |U| \ge 1 $. Equation (29) states the fact that the two densities are equal when $ | U | = 0 $. Since the objective (5) minimizes this term for $\hat{\theta} $, and the minimizer is attained at $ | U | = 0 $ according to (29). In the formulation you suggest, this minimization objective (5) amounts to a sparsity condition/constraint on the specification $\hat{\theta} $.
>
> We have revised this part of the proof under the classic formulation (lines 1197-1262). We hope this is clearer to you and please let us know if you have any questions – many thanks!

---

> ### Author Response · Authors · 2024-11-22
> **Rebuttal by Authors [Part 3]**
>
> **(Implications) Q6: One core aspect of the causal model is that two modalities cannot share any direct causes, which is questionable. It’s unclear why. in the “sleep patterns” and “brain imaging” examples, your assumptions could be satisfied.**
>
> **A6**: Thank you for the insightful question. We had considered the existence of shared latent variables as a special case of our existing setting that needs some proper post-processing to deal with. (While due to the page limit, we did not discuss this case separately in the initial draft.)
>
> Specifically, we can check each pair of adjacent cross-modality latent variables whether they are nearly deterministically related and merge them as a shared variable if so. We compute the regression $R^{2}$ score to detect nearly deterministic relations. The following $R^{2}$ results are based on our current framework. In this experiment, the ground truth latent variables include two adjacent relations, with one latent variable being shared. Our estimation yields three adjacent relations, and one pair of latent variables is nearly deterministically related ($R^{2}=1.00$) so we can merge them together.
>
> | Adjacent Pairs | 1 | 2 | 3 |
> |--------|-------|--------|--------|
> | $R^2$ | 1.00 | 0.36 | 0.24 |
>
> Moreover, thanks to your question that suggests the significance of this setting, we have extended the proposed framework to deal with the scenario in a principled way. Theoretical results (section 4.2 in the revision) and the corresponding synthetic data experiments (appendix E5 in the revision) demonstrate the generality of our framework.
>
> Specifically, in Section 4.2, we discuss the generality of the extended theorem and provide detailed proofs in Appendix C.3 and C.4. For these proofs, we invoke a theorem from Yao et al. 2022, whose technical language is the same as that in our original draft. Due to time constraints, we have not been able to update the whole terminologies to align with the classical identifiability definition. We will update you when it is ready. Thank you for your patience.
>
> > Theorem 4.2 and Theorem 4.4 can be straightforwardly extended to identify causal models with directly shared latent variables as those in [3-5], thus strictly more general than prior work. In particular, we can identify such shared latent variables block-wise through incorporating contrastive learning objectives into Theorem 4.1. Therefore, we can treat such blocks of shared latent variables as separate modalities and directly apply Theorem 4.4 to attain the component-wise identifiability.
>
> In Appendix E5, we extend our current framework and introduce an additional contrastive loss, following methodologies inspired by [4-5]. This contrastive loss ensures the similarity of shared latent representations, facilitating the identification of shared causal structures across modalities. The results in the table below demonstrate the successful recovery of both shared and modality-specific latent variables in different scenarios (Two modalities and Three modalities).
>
> | Metric | Two mods | Three mods |
> |--------|---------------|---------------|
> | R2 | 0.86±5e−4 | 0.90±1e−4 |
> | MCC | 0.83±7e−4 | 0.83±4e−6 |
>
> Thank you again for this great suggestion – it greatly strengthens our work and we would like to acknowledge the contribution from the anonymous reviewer in the updated version.

---

> ### Author Response · Authors · 2024-11-22
> **Rebuttal by Authors [Part 4]**
>
> **(Biological Application) Q7: The demonstration of the biological application lacks depth, and the claimed “strong interpretability” is overstated.**
>
> **A7**: Thank you for your valuable feedback. We appreciate the opportunity to clarify and address each of these important points.
>
> **(1) How should we interpret the learned hidden variables?**
>
> To interpret the learned hidden variables, we primarily refer to existing medical literature, which supports their alignment with background knowledge, thus adding validity to our results.
>
> For example, our model identified a latent variable, FRight3, associated with hand grip strength and fundus image, which aligns with the finding that handgrip strength is significantly associated with intraocular pressure (IOP) [6]. Additionally, the association between the cataract and changes in IOP [7] aligns with the model's discovery. These connections underline the physiological relevance of the learned hidden variable.
>
> Moreover, the learned variables linked to fundus imaging (FRight1 & FLeft1) and age estimation are consistent with literature showing that fundus image color content correlates with age in healthy subjects [8]. This suggests that latent variables encoding fundus image features can serve as biomarkers for aging.
>
> Another latent variable Sleep1 associated with oxygen saturation and sleep metrics aligns with findings that oxygen saturation is a strong predictor of obstructive sleep apnea (OSA) severity [9]. This indicates that the model's latent variable effectively captures critical factors related to sleep disorders.
>
> At the same time, it is possible that some latent variables don't have a clear understanding because of limited knowledge at the current stage. However, we hope that the learned latent variable together with their relations can provide plausible hypotheses for domain experts to look into the problem and understand what is there, and even design plausible interventions involved.
>
> **(2) Does the method enhance our understanding of the data or its underlying mechanisms?**
>
> We believe our work can facilitate scientific discovery by automatically discovering hidden variables and causal relations. Given these interpretations in terms of how they interact with each other, as you can see in the response in the previous comment, some of the results are consistent with the findings in the medical area. We hope domain experts can be engaged to understand other hidden variables and increase their domain knowledge about some specific problems.
>
>
> **(3) How does the number of latent variables affect the results (possibly with very different causal graphs)?**
>
> Thank you for raising this issue. If the initial value of the latent variables is too high, with the sparsity constraint, our method can automatically reduce the number of latent variables. At the same time, we can follow the principle of previous research [10] to capture the maximum amount of information with a minimum number of latent variables to avoid redundancy. Specifically, we can manually set the range of latent variable numbers and use cross-validation to select the one with the lowest validation loss.
>
> In light of your question, we conducted synthetic experiments to validate the effectiveness of cross-validation. The ground-truth number of latent variables is two (Num=2) for each modality. The validation losses under different numbers of latent variables (Num=1,2,..,10) are summarized in the table below. It shows that cross-validation can accurately help identify the correct number of latent variables. More detailed experimental results have been added in the revised manuscript, and we have added a discussion section on the number of latent variables in Appendix E2.
>
>
> Num=1 | Num=2 | Num=3 | Num=4 | Num=5 | Num=6 | Num=7 | Num=8 | Num=9 | Num=10 |
> |---------|---------|---------|---------|---------|---------|---------|---------|---------|----------|
> | 0.144 | **0.141** | 0.211 | 0.198 | 0.247 | 0.252 | 0.266 | 0.307 | 0.340 | 0.362 |

---

> ### Author Response · Authors · 2024-11-22
> **Rebuttal by Authors [Part 5]**
>
> **(Biological Application) Q8: Why are causal relationships involving FRight and FLeft not symmetric?**
>
> **A8**: Thank you for the question. We appreciate your attention to detail and will address each point individually.
>
> **(1) Why are causal relationships involving FRight and FLeft not symmetric?**
>
> The asymmetry observed between FRight and FLeft aligns with well-documented phenomena in biological systems. Research on CNS asymmetry [11] highlights that such disparities often arise due to functional specialization, developmental differences, or systemic influences. These factors play a critical role in shaping asymmetric relationships across sensory systems, including vision [12]. A detailed explanation of the fundus data is provided below.
>
> **(2) Why does FRight affect Cataract but not FLeft?**
>
> Asymmetries in ocular characteristics, such as the cup-to-disc ratio or refractive errors, are often linked to specific pathologies. Studies have shown that systemic factors, measurement variability, or disease processes can contribute to differences between the eyes [13]. If FRight affects cataract while FLeft does not, this may indicate an underlying systemic or pathological cause specific to FRight.
>
> The dominant eye typically experiences greater environmental exposure, particularly to light, and undergoes more frequent lens shape adjustments during accommodation. These factors can accelerate cataract formation. Moreover, cataracts introduce asymmetry in lens opacity, resulting in unequal visual inputs and long-term disparity in the brain's visual processing [14]. This finding emphasizes that differences in FRight and FLeft reflect both functional and environmental influences on the dominant eye.
>
> **(3) Why are there no Sleep variables that influence “Sleep efficiency”?**
>
> Sleep efficiency is not directly influenced by the measured variables in the sleep monitoring modality because these variables primarily focus on obstructive sleep apnea (OSA)-related indices [15]. Patients wear a portable device overnight to monitor their OSA-related indices such as breathing patterns, heart rate, oxygen levels, and other sleep patterns. While these variables provide insights into sleep architecture and other aspects like heart rate variability and arrhythmia, they are not significant predictors of sleep efficiency.
>
>
> **(4) Shouldn’t ocular quality impact hand grip strength rather than the reverse?**
>
> A study on intraocular pressure (IOP) responses during maximal isometric handgrip efforts revealed that hand dominance influences the IOP rise during exertion [6], with the dominant hand exhibiting a greater increase compared to the non-dominant hand. This finding suggests a strong physiological link between handgrip strength and the ocular system. This implies that hand dominance may modulate the ocular response under strain, potentially exacerbating eye condition changes during high-effort tasks.
>
> We appreciate the reviewer’s comments on the biology result and have added a detailed discussion on the phenotype dataset in Appendix D5 in the revised manuscript.
>
>
> **Q9: More experiments should be conducted, both on the model side and the data side.**
>
> **A9**: Thank you for your feedback. Previously, we followed the existing literature’s experimental setups/scales [16-17] to verify our theoretical contribution. In light of your concern, we have verified our framework on higher-dimensional simulation tasks under different numbers of latent variables and different modalities, where the causal relationships between variables are more complex.
>
> Specifically, we consider three cases: (1) Two mods: Observations are 30-dimensional, generated from two modalities, with a total of eight latent variables and two exogenous variables. (2) Five mods: Observations are 30-dimensional, generated from five modalities, with a total of ten latent variables and five exogenous variables. (3) Six mods: Observations are 30-dimensional, generated from six modalities, with a total of twelve latent variables and six exogenous variables.
>
> The results are summarized in the table below. Metrics such as MCC and R2 show that our method continues to perform well under these more challenging conditions. We have updated the manuscript and summarized the results in Table 4 in Appendix E1.
>
> Please let us know whether these additional experiments have addressed your concerns. We would appreciate any feedback on this – thank you in advance!
>
> | Metric | Two mods | Five mods | Six mods |
> |--------|---------------|---------------|----------------|
> | R2 | 0.89±1e−4 | 0.89±1e−4 | 0.97±8e−7 |
> | MCC | 0.83±4e−6 | 0.84±3e−4 | 0.82±4e−4 |

---

> ### Author Response · Authors · 2024-11-22
> **Rebuttal by Authors [Part 6]**
>
> **Reference**
>
> [1] Hyvarinen, Aapo, and Hiroshi Morioka. "Unsupervised feature extraction by time-contrastive learning and nonlinear ica." Advances in neural information processing systems 29 (2016).
>
> [2] Hyvarinen, Aapo, Hiroaki Sasaki, and Richard Turner. "Nonlinear ICA using auxiliary variables and generalized contrastive learning." The 22nd International Conference on Artificial Intelligence and Statistics. PMLR, 2019.
>
> [3] Von Kügelgen, Julius, et al. "Self-supervised learning with data augmentations provably isolates content from style." Advances in neural information processing systems 34 (2021): 16451-16467.
>
> [4] Daunhawer, Imant, et al. "Identifiability results for multimodal contrastive learning." arXiv preprint arXiv:2303.09166 (2023).
>
> [5] Yao, Dingling, et al. "Multi-view causal representation learning with partial observability." arXiv preprint arXiv:2311.04056 (2023).
>
> [6] Pérez-Castilla A, García-Ramos A, Redondo B, Andrés FR, Jiménez R, Vera J. Determinant factors of intraocular pressure responses to a maximal isometric handgrip test: hand dominance, handgrip strength and sex. Current Eye Research. 2021 Jan 2;46(1):64-70.
>
> [7] Slabaugh M, Chen P, Smit B, et al. Cataract surgery and IOP[J]. Glaucoma Today, 2013: 17-8.
>
> [8] Ege BM, Hejlesen OK, Larsen OV, Bek T. The relationship between age and colour content in fundus images. Acta Ophthalmol Scand. 2002 Oct;80(5):485-9. doi: 10.1034/j.1600-0420.2002.800505.x. PMID: 12390158.
>
> [9] Wali S O, Abaalkhail B, AlQassas I, et al. The correlation between oxygen saturation indices and the standard obstructive sleep apnea severity[J]. Annals of thoracic medicine, 2020, 15(2): 70-75.
>
> [10] Khemakhem, Ilyes, et al. "Ice-beem: Identifiable conditional energy-based deep models based on nonlinear ica." Advances in Neural Information Processing Systems 33 (2020): 12768-12778.
>
> [11] Andrew RJ. Origins of asymmetry in the CNS. InSeminars in cell & developmental biology 2009 Jun 1 (Vol. 20, No. 4, pp. 485-490). Academic Press.
>
> [12] Soydan A, Kürtül İ, Ray G, et al. Examining the concordance between dominant eye and hand preference in healthy adults[J]. Northwestern Medical Journal, 2024, 4(3): 176-180.
>
> [13] Maguire MG. Assessing intereye symmetry and its implications for study design. Investigative Ophthalmology & Visual Science. 2020 Jun 3;61(6):27-.
>
> [14] Song T, Duan X. Ocular dominance in cataract surgery: research status and progress. Graefe's Archive for Clinical and Experimental Ophthalmology. 2024 Jan;262(1):33-41.
>
> [15] https://knowledgebase.pheno.ai/datasets/009-sleep.html
>
> [16] Von Kügelgen J, Sharma Y, Gresele L, et al. Self-supervised learning with data augmentations provably isolates content from style[J]. Advances in neural information processing systems, 2021, 34: 16451-16467.
>
> [17] Zhang K, Xie S, Ng I, et al. Causal representation learning from multiple distributions: A general setting[J]. arXiv preprint arXiv:2402.05052, 2024.

---

> ### Author Response · Authors · 2024-11-25
> **Could you please let us know whether our responses properly addressed your concern?**
>
> Dear Reviewer 91p1,
>
> Thank you again for your valuable comments. Your suggestions on the theorem and the implications were very helpful to us, and we are eager to learn whether our biological explanations and responses have properly addressed your concerns.
>
> Due to the limited time for discussion, we look forward to your feedback and hope to have the opportunity to respond to any further questions you may have.
>
> Yours sincerely,
>
> Authors of Submission 11612

---

> > ### Comment · Reviewer_91p1 · 2024-11-26
> > **Thank you for the rebuttal**
> >
> > I'd say this is an exceptionally detailed rebuttal and I appreciate it a lot.
> >
> > The most critical concern remaining is on the proof of Th 4.2. You said "To show the identifiability..., we show *given the same $x^{(m)}$ values*, the identifiability between $z^{(m)}$ and $\hat{z}^{(m)}$". This is exactly what concerned me. Again, standard identifiability is what I wrote as $\forall \phi_1, \phi_2 \in \Phi (p_{\phi_1}=p_{\phi_2} \implies \phi_1=\phi_2)$. *The starting point should be equal distributions* (indexed by different parameters), but not equal obervational ($x^{(m)}$ ) values. In addition, even if we proved the identifiability of parameters, we could identify $z^{(m)}$ only when the $\epsilon$ noises are zero.

---

> > > ### Author Response · Authors · 2024-11-29
> > > **Could you please kindly let us know if our further response answers your questions?**
> > >
> > > Dear Reviewer 91p1,
> > >
> > > We are very grateful for your careful review and engaged discussion. We have provided further clarification regarding identifiability in the proof of Theorem 4.2 in our response and the revised submission, and we hope that you find the concerns properly addressed. We understand that you have a busy schedule and your further feedback would be highly appreciated. We hope for the opportunity to respond to it.
> > >
> > > Yours sincerely,
> > >
> > > Authors of Submission 11612

---

> ### Author Response · Authors · 2024-11-26
>
> Thank you so much for acknowledging our rebuttal and your comments! We humbly believe that the identifiability adopted in this paper is consistent with the classical definition. Let us elaborate below. In case you don’t find it natural, please kindly let us know and we would appreciate the opportunity to respond.
>
> It might be helpful to first address your second point “even if we proved the identifiability of parameters, we could identify $z^{(m)}$ only when the $\epsilon$ noises are zero.”
>
> Following literature on ICA [1,2,6] and causal representation learning [3,4,5], we assume the generating function $ g _{ \mathbf{x}^{(m)} }  $ (Eq. (2)) is an invertible map from $ (\mathbf{ z }^{\mathbf{(m)}}, \mathbf{ \epsilon }^{ (m) })  $ to $ \mathbf{ x }^{ (m) } $ (Condition 4.1).
> Under this invertibility assumption, given the value of $ \mathbf{x }^{ (m) } $, one can perfectly determine the value of $ \mathbf{ z}^{ (m) } $, which is essentially the posterior $ p ( \mathbf{ z} | \mathbf{ x } ) $ (a point mass here). Therefore, this type of result is a special case of the general case you allude to. Here, $\mathbf{z} ^{ (m) }$ is a function of $\mathbf{x} ^{ (m) }$, and the identifiability of $g$ gives rise to the result that $\mathbf{z} ^{ (m) }$ values can be identified from $\mathbf{x} ^{ (m) }$.
>
> Back to your main question:
> Given the above discussion, we can see the subtle difference between the two proof formulations.
> As you’ve kindly written down, usually, we start with equal distributions $  p_{\phi_1}=p_{\phi_2} $ to derive the equivalence of parameters $ \phi_1=\phi_2 $.
> However, in our case, since functions $g _{ \mathbf{x}^{(m)} }$, $ \hat{g} _{ \mathbf{x}^{(m)} } $ are invertible, one can reason about the relation between the two specifications $g _{ \mathbf{x}^{(m)} }$ and $ \hat{g} _{ \mathbf{x}^{(m)} } $ through a composition $ h: = \hat{g}^{-1} _{ \mathbf{x}^{(m)} } \circ g _{ \mathbf{x}^{(m)} }$. For instance, $ h $ is the identity when the two specifications are identical.
>
> Equivalently, this approach starts with equal $ \mathbf{x} $ values to establish the relation between $ \mathbf{z} $ and $ \hat{\mathbf{z}} $, as we have followed in our proof.
> This technique is also employed in the proof of Corollary 13 in the classical ICA work [6] and recent work on nonlinear ICA [1,2] and causal representation learning [3,4,5].
>
> Thanks to your comments, we have included a discussion on the notions of identifiability in Appendix A and refer readers to it in the main text line 215.
>
> Please kindly let us know your feedback on this point; we look forward to the opportunity to respond.
> Again, thank you for your valuable questions, which have undoubtedly improved our work!
>
> **Reference**
>
>
> [1] Hyvarinen, Aapo, and Hiroshi Morioka. "Unsupervised feature extraction by time-contrastive learning and nonlinear ica." Advances in neural information processing systems 29 (2016).
>
> [2] Hyvarinen, Aapo, Hiroaki Sasaki, and Richard Turner. "Nonlinear ICA using auxiliary variables and generalized contrastive learning." The 22nd International Conference on Artificial Intelligence and Statistics. PMLR, 2019.
>
> [3] Von Kügelgen, Julius, et al. "Self-supervised learning with data augmentations provably isolates content from style." Advances in neural information processing systems 34 (2021): 16451-16467.
>
> [4] Daunhawer, Imant, et al. "Identifiability results for multimodal contrastive learning." arXiv preprint arXiv:2303.09166 (2023).
>
> [5] Yao, Dingling, et al. "Multi-view causal representation learning with partial observability." arXiv preprint arXiv:2311.04056 (2023).
>
> [6] Comon, Pierre. "Independent component analysis, a new concept?." Signal processing 36.3 (1994): 287-314.

---

> ### Author Response · Authors · 2024-12-02
>
> Dear Reviewer 91p1,
>
> ​​Once again, we are grateful for your time and efforts. Since the discussion period will end in less than 24 hours, we are very eager to get your feedback on our response and will be online waiting to respond. We understand that you are very busy, but we would highly appreciate the chance to address your concerns properly during the discussion.
>
> Yours sincerely,
>
> Authors of Submission 11612

---

> ### Author Response · Authors · 2024-12-03
> **Can the response solve your concerns?**
>
> Dear Reviewer 91p1,
>
> Thank you very much for your time and effort in reviewing our manuscript. We are fully committed to addressing all your concerns and have made comprehensive revisions. We believe these changes have significantly improved our work and hope they meet your expectations.
>
> We would greatly appreciate it if the reviewer could reconsider the rating score based on our revised manuscript. The main revisions are as follows.
>
> ### Theory
> - **Refinement of definitions and constraints**: Following the reviewer’s feedback, we have revised our statement of the identification goal `(Q1)`, notations `(Q4)`, and reformulated the regularization as a constraint on alternative specifications `(Q2)`, and a reformulation of Condition 4.3 `(Q5)`.
> - **Clarification of the theorems**: We have refined the technical presentation of our identifiability results to be consistent with the suggested framework and addressed a potential misunderstanding in Theorem 4.2 `(Q3, Follow-up Question)`.
>
> ### Implications
> - **Treatment of shared latent variables**: We have extended our framework to account for cases where two modalities may share latent variables. This extension is validated both theoretically and experimentally `(Q6)`.
>
> ### Biological Application
> - **Improved biological interpretability**: We have clarified how the learned latent variables may correspond to known biological phenomena by referring to the existing literature `(Q7 (1-2), Q8)`.
> - **Evaluation of latent variable numbers**: We have provided a guideline on how to choose the number of latent variables and conducted additional synthetic experiments to validate the effectiveness of cross-validation `(Q7 (3))`.
> - **More experimental evaluation**: To address concerns about limited experiments, we have verified our framework on higher-dimensional simulation tasks in more complex scenarios `(Q9)`.
>
> We believe that our detailed responses have addressed your concerns. We understand that you have a busy schedule, and would appreciate it so much if you could take our further responses into consideration and update your comments accordingly while discussing our submission with the ACs. We thank you again for your valuable time, effort, and expertise!
>
> Yours sincerely,
>
> Authors of Submission 11612

---

### Official Review · Reviewer_mBq3 · 2024-11-02

**Soundness:** 3
**Presentation:** 3
**Contribution:** 3
**Rating:** 6
**Confidence:** 2

**Summary:**

This paper develops a theory for identifying the underlying structure of multi-modal data involving latent causal variables, based on the assumption of smooth invertible generating functions. It further investigates the sparsity of this causal structure and proposes an estimation method. Experiments conducted on both synthetic and real human phenotype data demonstrate the effectiveness of the proposed approach.

**Strengths:**

- The paper is generally well-written.
- It includes an adequate literature review.

**Weaknesses:**

- On real impact. As mentioned by the authors, knowing the number of underlying latent variables is unrealistic.

- Missing implementation details, e.g., more details on data generation, network structure, optimization scheme, hyper-parameters, etc. see questions as well.

**Questions:**

1. In my view, the example provided in Section 3 does not align well with the proposed model. What is your rationale for considering heart sizes and bone structures as latent variables?
2. In Appendix D.1, could you provide further details on the MLP architecture used for the numerical data generation process? Have you considered implementing an invertible neural network structure in your simulations? If you increase the number of data samples, do you expect the MCC to approach 1? I would appreciate seeing results on this, particularly illustrated with four variables (two latent and two observed).
3. The arrow from $z$ to the adjacency matrix in Figure 4 is unclear. Could you clarify this relationship?
4. How did you compute the KL divergence? Do you assume that $p(\gamma)$ follows a Gaussian distribution, with the encoder output providing the mean and variance?
5. When calculating the SHD, given that the indices of latent variables can be permuted, could this pose an issue?
6. In the context of the MNIST dataset, what causal relationship exists between the colored MNIST and fashion MNIST data?
7. How does the learned adjacency matrix A relate to the one derived from the PC algorithm?

Comment:
For the real-world application, I encourage the authors to explicitly present evidence from the literature demonstrating the success of their algorithm.

---

> ### Author Response · Authors · 2024-11-22
> **Rebuttal by Authors [Part 1]**
>
> We are grateful for the time and effort you dedicated to reviewing our paper and for providing insightful comments and valuable feedback. Our point-by-point responses are provided below.
>
> **W1: On real impact. As mentioned by the authors, knowing the number of underlying latent variables is unrealistic.**
>
> Thank you for your good question. We considered this issue and here we used a cross-validation-based method for determining the appropriate number of latent nodes. Specifically, we can manually set the range of latent variable numbers and use cross-validation to select the one with the lowest validation loss.
>
> In light of your question, we have included the following experiments to validate the effectiveness of cross-validation. The ground-truth number of latent variables is two (Num=2) for each modality. The validation losses under different numbers of latent variables (Num=1,2,..,10) are summarized in the table below. It shows that cross-validation can accurately help identify the correct number of latent variables. More detailed experimental results have been added in the revised manuscript, and we have added a discussion section on the number of latent variables in Appendix E2.
>
> Num=1 | Num=2 | Num=3 | Num=4 | Num=5 | Num=6 | Num=7 | Num=8 | Num=9 | Num=10 |
> |---------|---------|---------|---------|---------|---------|---------|---------|---------|----------|
> | 0.144 | **0.141** | 0.211 | 0.198 | 0.247 | 0.252 | 0.266 | 0.307 | 0.340 | 0.362 |
>
> **W2: Missing implementation details, e.g., more details on data generation, network structure, optimization scheme, hyper-parameters, etc. see questions as well.**
>
> Thank you for the comment. In the revised manuscript, we have provided details on the network architecture, training settings, and hyperparameter settings in Appendix F. Furthermore, we provided more details on the data generation process in Appendix D1 and D2.
>
> Please see the revised manuscript for details and kindly let us know if there is any specific information we can include – thank you!
>
> **Q1: In my view, the example provided in Section 3 does not align well with the proposed model. What is your rationale for considering heart sizes and bone structures as latent variables?**
>
> **A1**: Thank you for the thoughtful question. Heart sizes and bone structures are considered latent variables because they are not directly measured in a given dataset. Instead, their information is reflected in the measured observations, which are the X-ray images in the dataset. Naturally, they can be considered as functions of the measured variables, which is consistent with the proposed data generation process.
>
>
> **Q2: In Appendix D.1, could you provide further details on the MLP architecture used for the numerical data generation process? Have you considered implementing an invertible neural network structure in your simulations?**
>
> **A2**: Thank you for the question. We used a three-layer MLP with randomly sampled well-conditioned weights and leaky ReLU activations. The hidden layer size is set to 8. Our current implementation follows established practices in the nonlinear ICA literature [1-3], which encourage invertibility through Leaky ReLU and well-conditioned weights. Moreover, the reconstruction loss term enforces that the reconstructed output closely approximates the original input, thereby encouraging the encoder to act as an approximate inverse of the decoder.
>
> In light of your question, we have added more details about the data generation process in Appendix D1. Please let us know if there are further questions – thank you!
>
> **Q3: If you increase the number of data samples, do you expect the MCC to approach 1? I would appreciate seeing results on this, particularly illustrated with four variables (two latent and two observed).**
>
> **A3**: Thank you for the insightful question. Yes, we would expect MCC to approach 1, as the number of data samples increases. To verify this, we have included new experiments in Appendix E3 on progressively larger sample sizes as you suggested. The results show that as the sample size increases, the MCC steadily approaches 1, confirming the effectiveness of our approach.

---

> ### Author Response · Authors · 2024-11-22
> **Rebuttal by Authors [Part 2]**
>
> **Q4: The arrow from to the adjacency matrix in Figure 4 is unclear. Could you clarify this relationship?**
>
> **A4**: Thank you for the question. The inputs (inbound arrow) to the adjacency matrix $\hat{\mathbf{A}}$ are all the causal components $ \hat{z}^{(m)} _{i} $ and the outputs (outbound arrow) are passed to the flow model to recover the corresponding exogenous variables $ \hat{ \epsilon }^{(m)} _{i} $. Thus, the adjacent matrix $\hat{\mathbf{A}}$ and the flow model jointly represent (the inverse of) the latent causal relation in Eq (1) and the binary elements of $\hat{\mathbf{A}}$ indicate the specific connectivity between each pair of components.
>
> Given your valuable feedback, we have included a detailed explanation of $\hat{\mathbf{A}}$ and its role in capturing these relationships in Appendix F1. Please let us know if there are further questions – thank you!
>
> **Q5: How did you compute the KL divergence? Do you assume that follows a Gaussian distribution, with the encoder output providing the mean and variance?**
>
> **A5**: Yes, we assume that the prior distribution follows a standard Gaussian. The KL divergence is computed between the posterior distribution, which is inferred from the encoder's output, and the standard Gaussian prior.
>
> **Q6: When calculating the SHD, given that the indices of latent variables can be permuted, could this pose an issue?**
>
> **A6**: Thank you for the nice question.  This permutation does not pose an issue – following existing literature (e.g.,  [4-5]), we systematically evaluate all possible permutations of the estimated latent variables to find the permutation with the best match (the highest SHD), thus eliminating this permutation indeterminacy.
>
> **Q7: In the context of the MNIST dataset, what causal relationship exists between the colored MNIST and fashion MNIST data?**
>
> **A7**: Thank you for your question.  Between the two modalities, the class labels from the colored MNIST serve as the cause for the class labels in the fashion MNIST in a non-deterministic manner. We have added Figure 7 in Appendix D2 to illustrate the causal relationships in the MNIST dataset.
>
> **Q9: How does the learned adjacency matrix A relate to the one derived from the PC algorithm?**
>
> **A9**: Thank you for the interesting question. Asymptotically, the learned adjacency matrix $A$ corresponds to a graph within the Markov equivalence class given by the PC algorithm (e.g., see [6] for discussion on the linear Gaussian case). In response to your question, we have incorporated this statement in Appendix D5.
>
>
> **Q10: For the real-world application, I encourage the authors to explicitly present evidence from the literature demonstrating the success of their algorithm.**
>
> **A10**: Thank you for your advice! We have added a discussion section in Appendix D5, and provide more explicit evidence from the medical literature to demonstrate that our results are consistent with the findings in the medical area.
>
> For example, our model identified a latent variable, FRight3, associated with hand grip strength and fundus image, which aligns with the finding that handgrip strength is significantly associated with intraocular pressure (IOP) [7]. Additionally, the association between the cataract and changes in IOP [8] aligns with the model's discovery. These connections underline the physiological relevance of the learned hidden variable.
>
> Moreover, the learned variables linked to fundus imaging (FRight1 & FLeft1) and age estimation are consistent with literature showing that fundus image color content correlates with age in healthy subjects [9]. This suggests that latent variables encoding fundus image features can serve as biomarkers for aging.
>
> The latent variable Sleep1 associated with oxygen saturation and sleep metrics aligns with findings that oxygen saturation is a strong predictor of obstructive sleep apnea (OSA) severity [10]. This indicates that the model's latent variable effectively captures critical factors related to sleep disorders.

---

> ### Author Response · Authors · 2024-11-22
> **Rebuttal by Authors [Part 3]**
>
> **Reference**
>
> [1] Von Kügelgen J, Sharma Y, Gresele L, et al. Self-supervised learning with data augmentations provably isolates content from style. Advances in neural information processing systems, 2021.
>
> [2] Yao W, Sun Y, Ho A, et al. Learning Temporally Causal Latent Processes from General Temporal Data. International Conference on Learning Representations, 2022.
>
> [3] Zimmermann, Roland S., et al. "Contrastive learning inverts the data generating process." International Conference on Machine Learning. PMLR, 2021.
>
> [4] Kivva, Bohdan, et al. "Learning latent causal graphs via mixture oracles." Advances in Neural Information Processing Systems 34 (2021): 18087-18101.
>
> [5] Yao W, Sun Y, Ho A, et al. Learning temporally causal latent processes from general temporal data[J]. arXiv preprint arXiv:2110.05428, 2021.
>
> [6] Shimizu, S., Hoyer, P. O., Hyvärinen, A., Kerminen, A., & Jordan, M. (2006). A linear non-Gaussian acyclic model for causal discovery. Journal of Machine Learning Research, 7(10).
>
> [7] Pérez-Castilla A, García-Ramos A, Redondo B, Andrés FR, Jiménez R, Vera J. Determinant factors of intraocular pressure responses to a maximal isometric handgrip test: hand dominance, handgrip strength and sex. Current Eye Research. 2021 Jan 2;46(1):64-70.
>
> [8] Slabaugh M, Chen P, Smit B, et al. Cataract surgery and IOP[J]. Glaucoma Today, 2013: 17-8.
>
> [9] Ege BM, Hejlesen OK, Larsen OV, Bek T. The relationship between age and colour content in fundus images. Acta Ophthalmol Scand. 2002 Oct;80(5):485-9. doi: 10.1034/j.1600-0420.2002.800505.x. PMID: 12390158.
>
> [10] Wali S O, Abaalkhail B, AlQassas I, et al. The correlation between oxygen saturation indices and the standard obstructive sleep apnea severity[J]. Annals of thoracic medicine, 2020, 15(2): 70-75.

---

> ### Author Response · Authors · 2024-11-25
> **Could you please let us know whether our responses properly addressed your concern?**
>
> Dear Reviewer mBq3,
>
> Thank you very much for your time and effort in reviewing our manuscript. We particularly appreciate your concerns regarding the real-world extension and implementation details, and have revised our manuscript accordingly.
>
> As the discussion deadline approaches, please let us know if you have any further concerns and we will be happy to address them.
>
> Yours sincerely,
>
> Authors of Submission 11612

---

> ### Comment · Reviewer_mBq3 · 2024-11-25
>
> Thank you for your feedback. However, points A4 and A7 lack the necessary detail.
> In A4, how to get A from z?
> Could the authors please provide further clarification on these points?

---

> > ### Author Response · Authors · 2024-11-25
> > **Responses to the follow-up questions**
> >
> > Dear Reviewer mBq3,
> >
> > Thank you so much for your time and feedback! Please let us answer your questions below.
> >
> > **Follow-Up Q4: How to get A from z?**
> >
> > **Follow-Up A4:** Thank you for the question! There might be a misunderstanding – the adjacency matrix $\hat{\mathbf{A}}$ is a set of *trainable model parameters*, rather than outputs derived from $\mathbf{z}$. It is trained along the entire architecture in an end-to-end fashion, with sparsity loss imposed $ \mathcal{L} _{ \mathrm{Sp}  } $ on it (Eq. (6)). The edges that run into/out $\hat{\mathbf{A}}$ indicate the implementation of the causal connectives among all components $\mathbf{z}$.
> >
> > Specifically, according to the latent causal relation in Eq (1): $z_ i = g_ i (\text{Pa}(z_ i), \epsilon_ i)$. The causal relationships, represented as $\text{Pa}(z_ i)$, are embedded in the adjacency matrix $\hat{\mathbf{A}}$. The binary elements of $\hat{\mathbf{A}}$ indicate whether specific pairs of vertices contribute to the generation of latent components. Based on the parent variables specified by $\hat{\mathbf{A}}$, we generate the input for the flow model and apply a flow transformation to $z_ i$, mapping it back to $\epsilon _{i}$.
> >
> > **Follow-Up Q7: In the context of the MNIST dataset, what causal relationship exists between the colored MNIST and fashion MNIST data?**
> >
> > **Follow-Up A7:** Thanks for your further question! Let’s denote the colored MNIST as modality 1 and the fashion MNIST as modality 2. The colored MNIST dataset involves two latent components: the digit class $ z^{(1)} _{1} $ and the image color $ z^{(1)} _{2} $. The fashion MNIST dataset also involves two latent components: the clothing class  $ z^{(2)} _{1} $ and the image rotation angle $ z^{(2)} _{2} $.
> >
> > Between two modalities, we implement the digit class $ z^{(1)} _{1} $ in the colored MNIST as the cause for the clothing labels $ z^{(2)} _{1} $ in the fashion MNIST through a function $ z^{(2)} _{1} = z^{(1)} _{1} + \epsilon $, where $ \epsilon $ denotes the exogenous variable in this causal relation.
> >
> > Please let us know if there are any further concerns, and we are more than happy to address them.
> >
> > Sincerely,
> >
> > Authors of Submission 11612

---

> ### Author Response · Authors · 2024-12-02
> **Could you please kindly let us know if our further response answers your questions?**
>
> Dear Reviewer mBq3,
>
> Thank you for your valuable time in reviewing our paper and for your helpful comments. Your feedback on the real impact and implementation details was extremely helpful in improving our paper. We hope that our further responses regarding A4 and A7 have addressed your concerns.
>
> Due to the limited time for rebuttal discussion, we look forward to your feedback at your earliest convenience and the opportunity to respond to it.
>
> Sincerely,
>
> Authors of Submission 11612

---

### Official Review · Reviewer_uhwZ · 2024-11-03

**Soundness:** 3
**Presentation:** 3
**Contribution:** 3
**Rating:** 6
**Confidence:** 3

**Summary:**

Authors develop new methods to identify patterns in complex multi-modal biological datasets. It improves upon previous work in two key ways: First, it uses a flexible mathematical framework that doesn't make rigid assumptions about how the underlying biological factors (latent factors) are distributed. Second, these factors can influence each other across different modalities. Simulation study and real-world data support authors's claim.

**Strengths:**

1. The paper is well-written and motivated.
2. Authors provide identifiability guarantees for each latent component. This is helpful to characterize the interactions among all latent
components across modalities for the biological applications.
3. The assumptions of theoretical results look reasonable to me.
4. Real-world dataset analysis is provided, demonstrating its usefulness.

**Weaknesses:**

1. A1 indicated the neural network needs to be invertible. How do authors achieve this? Authors use normalizing flow, an invertible generative model, as a part of their network, what about others? Also what's the computation efficiency?
2. In real world, user-defined number of latent variables can be biased. Have authors analyzed it?

**Questions:**

See above

---

> ### Author Response · Authors · 2024-11-22
> **Rebuttal by Authors**
>
> We sincerely appreciate the time you took to review our paper, as well as your insightful comments and valuable feedback. Our detailed responses are provided below.
>
> **Q1: A1 indicated the neural network needs to be invertible. How do authors achieve this? Authors use normalizing flow, an invertible generative model, as a part of their network, what about others?**
>
> **A1:** Thank you for the question. Following established practices in the nonlinear ICA literature [1-3], we adopt LeakyReLU activations and well-conditioned weights for model initializations to facilitate invertibility.
> In addition, the reconstruction loss enforces that the reconstructed output closely approximates the original input, thereby encouraging the encoder to act as an approximate inverse of the decoder.
>
>
> **Q2: What's the computation efficiency?**
>
> **A2**: To evaluate computation efficiency, we compared our method with other baselines on a synthetic dataset in terms of training time, testing time, and model parameter size. Below is a detailed comparison:
>
> | **Model** | **Para. Size** | **Training Time (s)** | **Avg. Training Time/Epoch (s)** | **Testing Time (s)** |
> |-----------------|--------------------|-----------------------------|----------------------------------|----------------------------|
> | **CausalVAE** | 9437 | 210.82 | 3.51 | 1.21 |
> | **BetaVAE** | 2196 | 201.65 | 0.13 | 0.50 |
> | **MCL** | 2502 | 943.97 | 0.47 | 44.63 |
> | **Our Method** | 3169 | 156.98 | 0.52 | 1.00 |
>
> **Q3: In real world, user-defined number of latent variables can be biased. Have authors analyzed it?**
>
> **A3**: Thank you for the insightful question. We considered this issue and here we used a cross-validation-based method for determining the appropriate number of latent nodes. Specifically, we can manually set the range of latent variable numbers and use cross-validation to select the one with the lowest validation loss.
>
> In light of your question, we have included the following experiments to illustrate the effectiveness of cross-validation. The ground-truth number of latent variables is two (Num=2) for each modality. The validation losses under different numbers of latent variables (Num=1,2,..,10) are summarized in the table below. It shows that cross-validation can accurately help identify the correct number of latent variables. More detailed experimental results have been added in the revised manuscript, and we have added a discussion section on the number of latent variables in Appendix E2.
>
> Num=1 | Num=2 | Num=3 | Num=4 | Num=5 | Num=6 | Num=7 | Num=8 | Num=9 | Num=10 |
> |---------|---------|---------|---------|---------|---------|---------|---------|---------|----------|
> | 0.144 | **0.141** | 0.211 | 0.198 | 0.247 | 0.252 | 0.266 | 0.307 | 0.340 | 0.362 |
>
>
> **Reference**
>
> [1] Von Kügelgen J, Sharma Y, Gresele L, et al. Self-supervised learning with data augmentations provably isolates content from style. Advances in neural information processing systems..
>
> [2] Yao W, Sun Y, Ho A, et al. Learning Temporally Causal Latent Processes from General Temporal Data. International Conference on Learning Representations, 2022.
>
> [3] Zimmermann, Roland S., et al. "Contrastive learning inverts the data generating process." International Conference on Machine Learning. PMLR, 2021.

---

> ### Author Response · Authors · 2024-11-25
> **Could you please let us know whether our responses properly addressed your concern?**
>
> Dear Reviewer uhwZ,
>
> Thank you for your valuable time in reviewing our paper and for your helpful suggestions. Your feedback on the implementation and real-world extension were very helpful in improving our paper. We hope that our responses have addressed your concerns.
>
> Due to the limited time for rebuttal discussion, we look forward to receiving your feedback at your earliest convenience and the opportunity to respond to it.
>
> Yours sincerely,
>
> Authors of Submission 11612

---

### Official Review · Reviewer_JxDh · 2024-11-04

**Soundness:** 3
**Presentation:** 4
**Contribution:** 3
**Rating:** 8
**Confidence:** 3

**Summary:**

This paper provides a method that identifies the latent variables up to invertible component-wise transformations from multi-view data under the weak assumption that these latent variables exert partial influence on every modality and strong influence on one modality. The proposed approach relies on weaker assumptions compared to the existing works and yet provides strong results. The primary motivation for this approach is on multimodal health care data but the experiments are conducted on both synthetic and image datasets in addition to health care data.

**Strengths:**

1.   The contribution is very clear and useful for future works on identifiability. More strengths included in the summary.
2.   The writing is lucid. The examples are cleverly used to contrast the proposed method against the existing works.

**Weaknesses:**

I did not find any significant weaknesses. However, I do have a few questions for the sake of clarity. Questions in the following section.

**Questions:**

1. It is assumed that the information of $z^{(m)}$ is preserved in its corresponding observation $x^{(m)}$ and it exerts sufficient influence on other modalities' observations $x^{(-m)}$. Now, consider the observation from another modality $x^{(k)}$ that has a similar behavior. How does $z^{(k)}$ not interfere in the identifiability of the subspace of $z^{(m)}$?
2. Is the considered setting a non-linear version of [10]? Can the authors add text comparing/contrasting the proposed approach with [10]?
3. Does "fully share" mean "retrieve without errors" in line 212? That's what I understood from the example that follows this statement in line 215.
4. Can the authors explain how the method is different from [9] apart from the latent variables being independent in [9]?Specifically, what does "part of the variables are directly observed and causal directions are given by default" mean in lines 335-338?
5. Can the authors add an intuitive description for Condition 4.3 as its implications are not clear from the main text? It is enough that this intuitive explanation is included in the appendix.


Typos and minor writing mistakes:

1. In line 78, what is $n$? Is it $z_i^{(m)}\rightarrow z_j^{(n)}$?
2. Is the co-domain of $G(z, \epsilon)$ correct in line 303?
3. Some issues with the references. E.g., "Non-parametric Identifiability of causal representations from unknown interventions" appeared in NeurIPS 2023. Sturma et al., (2023) cited twice in lines 126-127.


[1] Sturma et al., 2023, "Unpaired Multi-domain Causal Representation Learning", NeurIPS 2023

[2] Zheng et al., 2022, "On the Identifiability of Nonlinear ICA: Sparsity and Beyond", NeurIPS 2022

---

> ### Author Response · Authors · 2024-11-22
> **Rebuttal by Authors [Part 1]**
>
> We sincerely thank you for the time dedicated to reviewing our paper and encouraging feedback. Please find the response to your comments and questions below.
>
> **Q1: Consider the observation from another modality $x^{(k)}$ that has a similar behavior. How does $z^{(k)}$ not interfere in the identifiability of the subspace of $z^{(m)}$?**
>
> **A1:** Thank you for the thoughtful question. This is because of the conditional independence condition $\mathbf{x}^{(m)} \perp  \mathbf{z}^{(k)} | \mathbf{z}^{(m)} $. That is, all the influence from other modalities $\mathbf{z}^{(k)}$ to $\mathbf{x}^{(m)}$ is fully captured in $\mathbf{z}^{(m)}$.
> In particular, unlike the exogenous information $ \eta^{(m)} $, $ \mathbf{z}^{(k)} $ would not add information to $ \mathbf{x}^{(m)} $ *in addition to* $ \mathbf{z}^{(m)} $. It follows that $ \mathbf{z}^{(k)} $ would not add information to $ \hat{\mathbf{z}}^{(m)} $, as $ \hat{ \mathbf{z} }^{(m)}  $ is defined as a function of $ \mathbf{x}^{(m)} $ (i.e., $ \hat{ \mathbf{z} }^{(m)}, \hat{ \eta }^{ (m) } := \hat{g}^{-1} _{ \mathbf{x}^{(m)} } ( \mathbf{x}^{(m)} ) $).
>
> Please let us know if this has clarified your question – thank you!
>
> **Q2: Is the considered setting a non-linear version of [10]? Can the authors add text comparing/contrasting the proposed approach with [10]?**
>
> **A2:** Thank you for the question. We assume that you refer to Sturma et al., please kindly let us know if this is not the case.
>
> In addition to the linear/nonlinear distinction you mention, Sturma et al. consider a *unpaired* multimodal setting, in which the cross-correspondences of samples are unknown. Suppose that we have two modalities: an EEG and a heartbeat record. In the *unpaired* setting, one has no knowledge of whether a specific EEG and a heartbeat record belong to the same patient. In our work and baseline works (e.g., Yao et al.), this relation is assumed known. As a trade-off, Sturma et al. resort to linearity and non-Gaussianity to overcome this difficulty, whereas we can permit more flexible nonlinear functions as you’ve keenly pointed out.
>
>
> **Q3: Does "fully share" mean "retrieve without errors" in line 212? That's what I understood from the example that follows this statement in line 215.**
>
> **A3:** You are absolutely right – it means that those latent variables can be retrieved without errors from any one of the modalities under their influence.
>
>
> **Q4: Can the authors explain how the method is different from [9] apart from the latent variables being independent in [9]? Specifically, what does "part of the variables are directly observed and causal directions are given by default" mean in lines 335-338?**
>
> **A4:** Thank you for the insightful question. We assume you are referring to Zheng et al. 2022; please kindly let us know if otherwise.
> Lines 335-338 (in the initial manuscript) allude to the fact Zheng et al. 2022 assume the sparsity of the causal connections between the latent variables and the observed variables; therefore, the directions (from the latent to the observed variables) are given and the children are directly observed. In contrast, we allow for general (not necessarily sparse) nonlinear transformations from the latent variables to the observed variables and we assume sparsity over the causal connections over the latent variables themselves. Thus, there is no prior knowledge of the directions of these edges, and variables on both ends are latent, posing a greater challenge for identification.
>
> Thanks to your feedback, we have included the following explanation in the revision (line 341).
>
> >In contrast, Zheng et al. (2022) assume the sparsity of the causal connections between the latent variables and the observed variables -- the directions (from the latent to the observed variables) are given and the children are directly observed.
>
> Please let us know if this is clear – thank you!

---

> ### Author Response · Authors · 2024-11-22
> **Rebuttal by Authors [Part 2]**
>
> **Q5: Can the authors add an intuitive description for Condition 4.3 as its implications are not clear from the main text? It is enough that this intuitive explanation is included in the appendix.**
>
> **A5:** Thank you for the helpful comments. Given your suggestion, we have included the following text in our appendix C2 and refer to it in main text line 333.
>
> >Condition 4.3 stipulates sparse cross-modality causal connections among latent components $ \mathbf{z} $. Under this condition, when one latent component $ \hat{z}^{(m)} _{i} $ is a function of two components $ \hat{z}^{(m)} _{j} $ and $\hat{z}^{(m)} _{k}  $ (when component-wise identification breaks), the cross-modality causal connections in $ \hat{\mathbf{G}} $ are guaranteed to be denser than those in $ \mathbf{G} $. Therefore, the sparsity control enforces us to select the sparest estimated models, in which one latent component $ \hat{z}^{(m)} _{i} $ is a function of a unique component $ z^{(m)} _{j}$, yielding the desired component-wise identifiability.
>
> We hope this helps and would be happy to further discuss!
>
> **Q6: Typos and minor writing mistakes.**
>
> **A6:** Thank you so much for these catches – they greatly improve the presentation of our work.
>
> **- In line 78, what is $n$? Is it $z_i^{(m)}\rightarrow z_j^{(n)}$?**
>
> Yes, you are absolutely right. We have corrected this typo in the revision.
>
> **- Is the co-domain of $G(z, \epsilon)$ correct in line 303?**
>
> Thanks for the question. Yes, this should be correct. The matrix $ \mathbf{G} $ encodes the influences among all the latent variables, so it is a square matrix with dimensions of $ d(\mathbf{z}) $.
>
> **- Some issues with the references.**
>
> We have revised them in the manuscript. Thank you so much for bringing these to our attention.

---

> > ### Comment · Reviewer_JxDh · 2024-11-24
> > **Response to authors' rebuttal**
> >
> > Thank you for the time you took to answer my questions. I apologize for the confusion in the reference numbers. Your answers pointed to the correct references. Your rebuttal has answered my concerns and I don't have any additional questions about the manuscript.

---

> > > ### Author Response · Authors · 2024-11-24
> > >
> > > Thank you again for your dedicated time and effort that have helped us improve our manuscript!

---

### Official Review · Reviewer_5aPk · 2024-11-04

**Soundness:** 3
**Presentation:** 4
**Contribution:** 3
**Rating:** 6
**Confidence:** 4

**Summary:**

This paper studies the causal graph discovery problem from multiview data. Given observed multimodal data, the goal is to estimate the causal relationship of latent features between modalities. The authors study the data generation process where the observed multimodal data are independent given the latent variables. Under this assumption, the authors employ the encoder-decoder framework to decompose the latent factors and nuisance factors, and a sparsity regularization function to impose the sparse relationship between modalities in the latent spaces. They compare the proposed method with several baselines on simulated tasks and the results show improved mean correlation coefficient. Furthermore, the proposed method is applied to human phenotype dataset.

**Strengths:**

This paper tackles an important and yet not well-addressed problem. It is well-motivated by biomedical applications. The proposed framework is more general compared to the prior work (as shown in Table 1).

**Weaknesses:**

In general, I find several parts that need further clarification. (see the question parts)

My main concern is that the simulated tasks focus on low-dimensional data with simple sparse causal structures. It is not clear whether the method is scalable and can be generalized to more complex causal structures.

**Questions:**

a. the paper assumes that the latent variables within the same modality form a DAG, could the author clarify why this assumption is necessary? Secondly, if this is needed, the loss function (6) does not enforce the latent variables within the same modality to be a DAG. How can we ensure such structures will be satisfied in the learning phase?

b. does the causal relationship between different modalities need to be DAG? If there are cycles, are we still being able to identify the latent variables?

c. I find the definition of condition 4.3 confusing. What are the connections between matrices A, G, T? and A seems to only denote the relationship between modality 1 and 2?

d. In Assumption A.1, what is smooth inverse?

e. Could the authors provide a guideline for practitioners on how to systematically choose the number of latent nodes?

f. In Theorem 4.4, does it require the regularization term to be scaled by some coefficient?

---

> ### Author Response · Authors · 2024-11-22
> **Rebuttal by Authors [Part 1]**
>
> Thank you for the time dedicated to reviewing our paper, the insightful comments, and valuable feedback. Please see our point-by-point responses below.
>
> **W1: I find several parts that need further clarification.**
>
> We appreciate your valuable comments and would be more than happy to address your concerns regarding assumptions, definitions, requirements, and how to choose the number of latent nodes. Any further feedback on these matters will be highly appreciated – thank you!
>
> **W2: My main concern is that the simulated tasks focus on low-dimensional data with simple sparse causal structures. It is not clear whether the method is scalable and can be generalized to more complex causal structures.**
>
> Thank you for your feedback. Previously, we followed the existing literature’s experimental scale [1-2] to verify our theoretical contribution. In light of your concern, we have verified our framework on higher-dimensional simulation tasks under different numbers of latent variables and different modalities, where the causal relationships between variables are more complex.
>
> Specifically, we consider three cases: (1) Two mods: Observations are 30-dimensional, generated from two modalities, with a total of eight latent variables and two exogenous variables. (2) Five mods: Observations are 30-dimensional, generated from five modalities, with a total of ten latent variables and five exogenous variables. (3) Six mods: Observations are 30-dimensional, generated from six modalities, with a total of twelve latent variables and six exogenous variables.
>
> The results are summarized in the table below. Metrics such as MCC and R2 show that our method continues to perform well under these more challenging conditions. We have updated the manuscript and summarized the results in Table 4 in Appendix E1.
>
> Please let us know whether these additional experiments have addressed your concerns. We would appreciate any feedback on this – thank you in advance!
>
> | Metric | Two mods | Five mods | Six mods |
> |--------|---------------|---------------|----------------|
> | R2 | 0.89±1e−4 | 0.89±1e−4 | 0.97±8e−7 |
> | MCC | 0.83±4e−6 | 0.84±3e−4 | 0.82±4e−4 |
>
>
> **Q1 (a, b): Could the author clarify why DAG assumption is necessary? If this is needed, how can we ensure such structures will be satisfied? Does the causal relationship between different modalities need to be DAG? If there are cycles, are we still being able to identify the latent variables?**
>
> **A1:** Thank you for the great questions. In fact, the DAG assumption is not necessary for our main theorems. In particular, the proof of Theorem 4.2 does not leverage any latent graphical structures (Condition 4.1) and the proof of Theorem 4.4 only necessitates the cross-modality graphical structures to be sparse, with no requirements on the DAG aspect.
>
> Please let us note that Theorems 4.2 and 4.4 do not claim any graphical structure identifiability. Rather, they focus on the subspace/component-wise identifiability of the latent variables. The graphical learning task can be performed as a post-processing step – one can apply existing graph learning algorithms to the identified components (treating them as observed variables), as we outlined in lines 196 - 198. To verify our claim, we have included in the revision synthetic experiments over non-DAG models in Appendix E4 as follows. We can observe that empirically the non-DAGs do not pose trouble for *latent variable identification*, as we have discussed.
>
> | Metric | Cyclic within Modality | Cyclic across Modalities |
> |--------|---------------|---------------|
> | R2 | 0.95±1e−5 | 0.94±2e−4 |
> | MCC | 0.89±2e−4 | 0.92±1e−5 |
>
>
> Initially, we assumed the DAG structure because of the convenience of illustration and also DAGs are widely used. If there are cycles over the latent variables, the identification of such graphs may be much more complicated and does not yield unique results in general (for instance, see this one with conditional independence-based methods [3] and see this one [4] for the linear non-Gaussian model). However, in practice, we completely agree that the DAG assumption can be relaxed.
>
> Thanks to your valuable questions, we have removed “directed acyclic” and added “one can choose structural learning algorithms suitable to the assumed graph class (e.g., potentially non-DAGs) and this step is orthogonal to our contribution” to lines 200 to clarify this point.
>
> Please let us know whether this has addressed your concerns, thank you!

---

> ### Author Response · Authors · 2024-11-22
> **Rebuttal by Authors [Part 2]**
>
> **Q2 (c): I find the definition of condition 4.3 confusing. What are the connections between matrices A, G, T? and A seems to only denote the relationship between modality 1 and 2?**
>
> A2: Thank you for the valuable feedback. $\mathbf{G}$ denotes the graphical connectivity in the true model (line 303) and $\mathbf{T}$ denotes potential mixings of latent variables in the \textit{estimated} model. Thus, the product $ \mathbf{T} \mathbf{G} $ represents the graphical structure in the \textit{estimated} model.
> $\mathbf{A}$ is a placeholder of submatrices of $ \mathbf{T} \mathbf{G} $ that correspond to cross-modality edges between modality $m$ and and any other modalities $-m$, which Condition 4.3 stipulates to be sparse.
> In light of your comment, we have clearly defined $\mathbf{A}$ in the revision Condition 4.3 and included  the explanation above (lines 308) to make their relations clearer.
>
> We would appreciate any feedback from you!
>
> **Q3 (d): In Assumption A.1, what is smooth inverse?**
>
> A3: Thank you for the question. *Smooth inverses* refers to smooth inverse functions. We have revised this to make it clearer. Thank you.
>
> **Q6 (e): Could the authors provide a guideline for practitioners on how to systematically choose the number of latent nodes?**
>
> A6: Thank you for the insightful question. We can manually set the range of latent variable numbers and use cross-validation to select the one with the lowest validation loss. In light of your question, we conducted synthetic experiments to validate the effectiveness of cross-validation. The ground-truth number of latent variables is two (Num=2) for each modality. The validation losses under different numbers of latent variables (Num=1,2,..,10) are summarized in the table below. It shows that cross-validation can accurately help identify the correct number of latent variables. More detailed experimental results have been added in the revised manuscript, and we have added a discussion section on the number of latent variables in Appendix E2.
>
> Num=1 | Num=2 | Num=3 | Num=4 | Num=5 | Num=6 | Num=7 | Num=8 | Num=9 | Num=10 |
> |---------|---------|---------|---------|---------|---------|---------|---------|---------|----------|
> | 0.144 | **0.141** | 0.211 | 0.198 | 0.247 | 0.252 | 0.266 | 0.307 | 0.340 | 0.362 |
>
>
>
> **Q7 (f): In Theorem 4.4, does it require the regularization term to be scaled by some coefficient?**
>
> Thank you for the question. Scaling is not needed for the theoretical development – in theory, we optimize this term under the constraint that the model $\\{\hat{g}\_{\mathbf{x}^{(m)}}, \hat{g}\_{\mathbf{z}^{(m)}}, \hat{p} (\epsilon^{(m)})\\}_{m=1}^{M}$ matches the observed distribution _$p(\mathbf{x})$_. In practice, we convert this constrained optimization problem to an unconstrained one (its Lagrangian), in which the objective becomes a weighted sum of the sparsity regularization (i.e., $\mathcal{L}\_{\mathrm{Sp}}$ in Equation (6)) and distribution matching (i.e., $\mathcal{L}\_{\mathrm{Ind}}$ and $\mathcal{L}\_{\mathrm{Recon}}$). The relative scaling of these terms is indeed needed.
>
>
>
> **Reference**
>
> [1] Von Kügelgen J, Sharma Y, Gresele L, et al. Self-supervised learning with data augmentations provably isolates content from style[J]. Advances in neural information processing systems, 2021, 34: 16451-16467.
>
> [2] Zhang K, Xie S, Ng I, et al. Causal representation learning from multiple distributions: A general setting[J]. arXiv preprint arXiv:2402.05052, 2024.
>
> [3] Richardson, Thomas S. Discovering cyclic causal structure. Carnegie Mellon [Department of Philosophy], 1996.
>
> [4] Lacerda, Gustavo, et al. "Discovering cyclic causal models by independent components analysis." arXiv preprint arXiv:1206.3273 (2012).

---

> ### Author Response · Authors · 2024-11-25
> **Could you please let us know whether our responses properly addressed your concern?**
>
> Dear Reviewer 5aPk,
>
> Thank you again for your valuable time dedicated to reviewing our paper and for your helpful suggestions. We particularly appreciate your questions regarding the assumption, theorem, and scalability. So we are eager to see whether our responses properly addressed your concerns and would be grateful for your feedback.
>
> With best wishes,
>
> Authors of Submission 11612

---

### Author Response · Authors · 2024-12-04
**Global Response**

We are thankful for the reviewers' thoughtful feedback and insightful comments. To provide further clarity, we'd like to share a summary of the discussion phase.

All five reviewers acknowledged the strengths of our paper:
- **Importance of problem**: `Reviewer 5aPk` highlighted that we address an important yet underexplored problem, and our framework is more general compared to prior work.
- **Theoretical contributions**: `Reviewer JxDh` recognized our clear and impactful contributions on identifiability. The examples effectively contrast our method with existing works. `Reviewer uhwZ` recognized the proposed identifiability is crucial in biological applications, and assumptions were deemed reasonable.
- **Relaxations of assumptions**: `Reviewer 91p1` appreciated the meaningful relaxations of assumptions from prior work.
- **Real-world relevance**: `Reviewer uhwZ` noted that the real-world analysis demonstrated practical utility.
- **Good writing quality**: `Reviewer mBq3` noted that the paper is generally well-written. `Reviewer JxDh` appreciated lucid writing. `Reviewer 91p1` noted that the paper is relatively well-written given the complexity of the theory and notation.

While the reviewers recognized the above strengths, they also raised some concerns:
- **Scalability and generalization**: `Reviewer 5aPk` noted that the simulated tasks focus on low-dimensional data, raising concerns about the scalability and applicability of the method to more complex causal structures. `Reviewer uhwZ` and `Reviewer mBq3` raised concerns about the real impact of the user-defined numbers of latent variables.
- **Implementation details**: `Reviewer uhwZ` questioned how to achieve invertibility and what is computational efficiency. `Reviewer mBq3` pointed out missing implementation details.
- **Theoretical claims**: `Reviewer 91p1` raised concerns about theoretical presentation and formulation.
- **Real-world applicability**: `Reviewer 91p1` questioned the practicality of the assumptions and their relevance to real-world scenarios. The analysis of human phenotype results was unclear to justify the paper's claims.

During the discussion phase, we carefully addressed each reviewer’s concerns. Below is a summary of the revised manuscript:
### Regarding Theory
- **Refinement of definitions and formulations** (`Reviewer 91p1`): We have revised our statement of the identification goal and notations, reformulated the regularization as a constraint on alternative specifications, and reformulated Condition 4.3.
- **Clarification of the theorems** (`Reviewer 91p1`): We have refined the technical presentation of our identifiability results to be consistent with the suggested framework and addressed a misunderstanding of the proof for Theorem 4.2.

### Regarding Experiment
- **More experimental evaluation** (`Reviewer 5aPk` and `Reviewer 91p1`): To address concerns about limited experiments, we have verified our framework on higher-dimensional simulation tasks in more complex scenarios.
- **Evaluation of latent variable numbers** (`Reviewer uhwZ`, `Reviewer mBq3` and `Reviewer 91p1`): We have provided a guideline on how to choose the number of latent variables and conducted additional synthetic experiments to validate the effectiveness.
- **Detailed implementation details** (`Reviewer uhwZ` and `Reviewer mBq3`): We have provided details on the data generation process, network architecture, training settings, and hyperparameter settings. We detailed how we achieved invertibility and empirically provided computational efficiency.

### Regarding Implications
- **Treatment of shared latent variables** (`Reviewer 91p1`): We have extended our framework to account for shared latent variables. This extension is validated both theoretically and experimentally.

### Regarding Biological Application
- **Improved biological interpretability** (`Reviewer 91p1`): We have clarified how the learned latent variables may correspond to known biological phenomena by referring to the existing literature.

Eventually:
- **Reviewer 5aPk** raised no further questions and raised the rating.
- **Reviewer JxDh** replied that our response has answered the reviewer’s questions, and maintained a positive rating.
- **Reviewer uhwZ** might have been occupied during the discussion phase and was unable to engage in discussions.
- **Reviewer mBq3** raised further concerns regarding implementation details and we have provided further details accordingly.
- **Reviewer 91p1** appreciated our detailed response, and raised questions regarding the theorem.  We believe that it is a language barrier and the further concern is a misunderstanding of the proof framework we follow from the field of causal representation and nonlinear ICA. We hope our further response can resolve this question effectively.

We thank the reviewers and the ACs’ efforts in encouraging discussions. We hope this summary will assist all reviewers and ACs in making the final decision.

---

### Meta-Review · Area_Chair_7cgm · 2024-12-21

**Metareview:**

This paper proposes a theoretical framework for identifiability conditions for causal representation learning from multimodal data. The reviews are largely positive, and agree this is a solid technical contribution to the community. I urge the authors to address a minor clarification required as noted by the reviewer 91p1.

**Additional Comments On Reviewer Discussion:**

The reviewers noted the positive aspects of the paper, including importance and impact of the problem being addressed, as well as the of the theoretical contritbutions of the paper. There were some concerns about scalability, some missing implementation details + complexity analysis, and the main point of contention on the definition of identifiability as raised by the reviewer 91p1 who seemed unconvinced till the end. I feel all the points were adeqautely addressed by the authors, as also noted by all the reviewers, except the identifiability issue raised by the reviewer 91p1. I think the issue is minor nomenclature/language problem which can be easily clarified during the camera ready edit, but even without this clarification, the paper in its current form is important enough to be accepted.

---

### Decision · Program_Chairs · 2025-01-22

Accept (Poster)